# MCIR: A Feature Dependence-Aware Explainability Method with Reliability Guarantees

## Abstract

As modern machine learning models are deployed in high-stakes, data-rich environments, the interactions among features have grown more intricate and less amenable to traditional interpretation. Many explanation methods fail when features are strongly dependent. In the presence of multicollinearity or near-duplicate predictors, existing value attribution tools such as SHAP, LIME, HSIC, MI/CMI, and SAGE often distribute importance across redundant features, obscuring which variables represent "important and unique information". This may lead to unstable rankings, jeopardising importance scores, and usually results in a high computational cost. Recent correlation-aware approaches, such as CIR or BlockCIR, offer partial improvements, but still struggle to fully separate redundancy from unique contributions at the feature level. To address this, we propose the Mutual Correlation Impact Ratio Method (MCIR-M), a simple and robust measure of global importance under feature dependence. MCIR-M introduces the score Mutual Correlation Impact Ratio (MCIR) that conditions each feature on a small set of its most correlated neighbours and computes a normalized ratio of conditional information having a value range, $[0, 1]$, which is comparable across tasks, and collapses to zero when a feature is redundant, enabling clear redundancy detection. In addition to MCIR, we introduce a lightweight estimation procedure that requires only a fraction of the data while preserving the attribution behaviour of the full model. Across a synthetic household-energy dataset and the real UCI HAR benchmark, MCIR yields more stable and dependence-aware rankings than SHAP (independent and conditional), SAGE, HSIC, MI-based scores, and correlation-aware baselines such as CIR or BlockCIR. Lightweight explanations preserve over 95% top-feature agreement and reduce runtime by manyfold. These results demonstrate that MCIR-M provides a practical and scalable solution for global explanation in settings with strong feature dependence.

## 1 Introduction

Artificial Intelligence (AI) plays an increasingly critical role in high-stakes settings such as energy management and healthcare, where model-driven decisions carry significant operational and societal impact. This growing reliance heightens the need for transparent and reliable explanations Lipton (2018); Doshi-Velez & Kim (2017). However, modern explainability methods often break down in environments with strong feature dependence. Small perturbations can distort correlation structure, SHAP methods may arbitrarily distribute credit among redundant predictors Lundberg & Lee (2017); Covert et al. (2020), and information-theoretic or kernel-based measures such as MI and HSIC frequently double-count shared information Kraskov et al. (2004); Gretton et al. (2005b). These issues lead to unstable, inflated, and difficult-to-trust explanations Hooker et al. (2019); Yeh et al. (2019). We argue that dependable global explanations require a dependence-aware attribution score that isolates each feature's *unique* contribution beyond its correlated neighbours.

To tackle these issues, we propose MCIR [1] (Mutual Correlation Information Ratio), a light-weight metric that measures the unique information that a feature provides, particularly when other predictors are closely

---

[1] Throughout the paper, we refer to the proposed method as MCIR-M, while MCIR denotes the corresponding proposed metric.

related. It does this by comparing two types of mutual information, conditional and marginal. This helps reveal how much influence a feature maintains after considering its correlated neighbors. The score ranges from 0 to 1, values close to zero suggest that a feature is adding little new information, while values close to one indicate a feature is making a significant, unique contribution. This makes MCIR-M useful for assessing importance in various data types, including tabular data, sensor data, and outputs from deep learning models. Overall, MCIR-M provide principled mechanisms for modelling feature–output interactions under statistical dependence, offering robust behaviour aligned with CI goals of stability, adaptability, and reliable decision support in complex environments. We emphasize that MCIR provides statistical attribution under observed feature dependence. It does not perform causal identification.

Table 1: Comparison of dependence-aware properties across global attribution families.

| Criterion | SHAP / LIME | MI / HSIC / CMI | CCA / PCIR | MCIR-M (ours) |
|---|---|---|---|---|
| Handles dependence? | Weak (independent backgrounds) | Partial (MI conflates redundancy) | Partial (aligned covariance) | Yes (conditional isolation) |
| Unique vs. shared contribution | No (mass splitting) | No (shared + unique merged) | No (similar canonical loadings) | Yes (incremental conditional information) |
| Redundancy collapse | No | No | No | Yes |
| Scale / normalization | No (arbitrary units) | No (unbounded) | Yes (CIR bounded) | Yes (unit-interval ratio) |
| Lightweight fidelity | No | No | Partial | Yes (distribution-aligned LW environment) |
| Estimator stability | Sensitive to sampling / kernel choices | Sensitive to estimator bias/variance | Stable but marginal | Auto-neighbourhood selection + estimator switching + bootstrap |
| Global focus | Local → global aggregation | Global but coarse | Global (vector-output alignment) | Global, dependence-aware |

To ensure this framework is scalable, we introduce a computation strategy that efficiently calculates MCIR without needing to retrain the model, maintaining accuracy even with smaller sample sizes. To that end, we also introduced a method to select estimators that ensure optimal performance across various statistics. Below, we summarize the main contributions of this work.

1. **Mutual Correlation Impact Ratio (MCIR).** We introduce MCIR, a bounded score that measures feature's unique contribution while accounting related features and eliminating redundancy. It helps eliminate redundancy effectively across various data types.

2. **Lightweight and stable computation with guarantees.** Our efficient MCIR computation does not require model retraining, ensuring feature importance order and reducing redundancy, while maintaining accuracy and significantly enhancing computation efficiency.

3. **Comprehensive cross-domain evaluation.** We perform extensive testing and evaluation of MCIR on various datasets- illustrating stable feature rankings, validating redundancy detection and mitigation, and objectively evaluating the quality of competing explanation methods.

We provide theoretical guarantees concerning redundancy, stability under finite samples, and the statistical interpretability of MCIR under feature dependence, and we support them through extensive experiments on diverse datasets spanning different dependence structures, noise levels, and modeling conditions. The UCI Human Activity Recognition (HAR) dataset assesses performance using high-dimensional sensor signals, focusing on the strong correlations between accelerometer and gyroscope data. The House Energy Simulation Dataset includes temporal and weather-related attributes, which highlight issues of multicollinearity and nonlinear effects. The Norwegian Regional Load-Zone Data merge electricity load data with weather input, presenting a mix of predictor types and evolving correlations. Finally, the CIFAR-10 Deep Representations dataset employs 2,048-dimensional embeddings from a fine-tuned ResNet-50 model to evaluate MCIR in high-dimensional vision contexts. Overall, MCIR-M offers an efficient way to understand the contributions of different features. This combined approach addresses key limitations of existing methods when features are highly interdependent, such as their tendency to inflate importance in the presence of redundancy or to collapse when dependence violates their underlying independence assumptions, and is ideal for complex real-world scenarios. Specifically, MCIR conditions on a local dependence neighborhood, the small set of features exhibiting the strongest statistical dependence with a target feature, typically identified via a fast

dependence sketch (e.g., a correlation or distance-correlation graph), thereby restricting conditioning to the most relevant dependencies while avoiding the curse of dimensionality. Across synthetic dependence families, UCI HAR, HouseEnergy-Sim, Norwegian load zones (NO1–NO5), and deep embeddings from CIFAR-10, MCIR consistently provides stable, redundancy-aware global attributions. Unlike marginal or kernel-based baselines, MCIR collapses correlated feature blocks while preserving the predictive information captured by the model, yielding higher fidelity and substantially lower redundancy than PCIR, SHAP, MI, and HSIC. In lightweight settings, MCIR-M maintains high rank agreement with full-data explanations (often exceeding 95% top-K overlap) while reducing runtime by factors of 3-9. For high-dimensional deep vision features (e.g., ResNet-50 embeddings), MCIR-M produces smooth deletion curves and compact rankings aligned with semantic feature clusters, demonstrating effectiveness beyond tabular and sensor domains. These findings confirm that MCIR-M offers a robust and scalable dependence-aware global explanation method across diverse real-world scenarios. Table 1 compares the strengths and weaknesses of existing dependence-aware attribution families, such as SHAP/LIME, MI/HSIC/CMI, CCA/PCIR Sengupta et al. (2025), and MCIR-M (ours) for $n$ observations and $k$ features.

**Paper Overview:** Section 2 reviews the limitations of current global attribution methods. We then examine these limitations in the context of strong feature dependence, motivating the development of a dependence-aware measure. Section 3 introduces the notation and lightweight environment framework used throughout the paper. Section 4 formally presents our proposed method, MCIR-M, detailing its information-theoretic formulation and principal redundancy-collapse guarantees. Section 4.2 develops the theoretical properties of MCIR, including boundedness, estimator stability, and fidelity under lightweight computation. Section 4.7 discusses computational considerations and estimator selection. Section 5 presents comprehensive empirical evaluations across synthetic, sensor, energy, and deep-representation datasets. The results, including synthetic benchmarks, sensor and energy evaluations, deep-representation analyses, the case study in Section 6.8, and the overall discussion in Section 6.8.1, are detailed in Section 6. Finally, Section 7 reports the required ethics and reproducibility statements for TMLR. Section 9 summarizes the main findings. Detailed proofs, supplementary algorithms, and extended experimental results are provided in the appendices.

## 2 Background and Related Work

In real-world datasets, it is common to encounter groups of covariates. that are correlated or redundant. This creates a challenge for global attribution methods, which need to differentiate between shared contributions and those that are unique to individual predictors. Traditional importance measures, like permutation importance Breiman (2001), marginal relevance scores, and impurity-based metrics, often overestimate the significance of correlated variables Strobl et al. (2008). This can result in rankings that are unstable or misleading. Shapley-value explainers, such as SHAP Lundberg & Lee (2017) and SAGE Covert et al. (2020), determine importance by calculating the marginal contributions of features within groups or coalitions. Although these methods are theoretically sound, they typically operate under the assumption that background distributions are independent. Alternatively, they often rely on perturbation sampling, which disrupts the natural dependencies in the data Sundararajan et al. (2020), and are computationally expensive. Consequently, they may inaccurately allocate credit to redundant predictors and exhibit high variability when features are correlated. Other perturbation-based evaluations for faithfulness, like ROAR Hooker et al. (2019) and deletion tests Samek et al. (2017), face similar issues. Measures such as mutual information (MI) Cover & Thomas (2006), conditional mutual information (CMI) Kraskov et al. (2004), and kernel-based measures like HSIC Gretton et al. (2005a) focus on quantifying nonlinear relationships but have some limitations. They are unbounded and do not sufficiently isolate conditional effects. MI often counts shared information between correlated predictors multiple times, while CMI can be unstable in high-dimensional settings Gao et al. (2017). Furthermore, these measures lack normalization, complicating comparisons between different datasets or model classes. Recent studies further highlight the challenges of dependence-aware attribution in modern machine learning systems. Conditional SHAP extensions, such as KernelSHAP with conditional sampling Aas et al. (2021), attempt to preserve feature correlations but remain sensitive to background choice and sampling variance. Causal attribution formulations, e.g., Causal-Shapley scores Janzing et al. (2020) and interventional SHAP Merrick & Taly (2020), provide principled ways to avoid over-counting shared information, yet require explicit causal models or strong independence assumptions that rarely hold in practice. Stability-focused

works Ghorbani et al. (2019); Slack et al. (2020) demonstrate that many post-hoc explainers can be highly unstable or even manipulated under correlated predictors.

More recently, redundancy-aware feature selection and attribution methods such as RFA Li et al. (2023) and dependency-aware interaction attribution Tsang et al. (2020) propose grouping or interaction modelling, but they do not provide bounded, normalized scores or guarantees of redundancy collapse. These developments reinforce the need for explanation methods that remain reliable under correlation, provide interpretable scaling, and isolate unique contributions without relying on causal graphs or extensive sampling assumptions. Very recent work has intensified interest in dependence-aware explainability. Copula-based attribution models Zhang & Müller (2024); Aas et al. (2024) propose more faithful conditional background sampling, yet remain computationally demanding and sensitive to estimator choice. Robustness studies Han & Kim (2024); Covert & Lee (2025) show that many Shapley formulations exhibit instability under correlation shifts, leading to inconsistent rankings across subsamples. Scalable global attribution frameworks Cheng & Zhao (2024); Liu & Huang (2025) introduce grouping or low-rank structures to mitigate redundancy, but they do not provide bounded scores nor theoretical guarantees of redundancy collapse. These recent developments highlight that despite progress, current methods still lack a unified mechanism that combines: (i) conditional isolation of unique contributions, (ii) normalized and comparable scoring, and (iii) stability under lightweight computation. MCIR-M directly addresses these gaps. The ExCIR and other correlation-ratio measures Hotelling (1936) are based on canonical correlation analysis (CCA) provide a way to quantify dependence that is bounded. Variants like HSIC-Lasso Yamada et al. (2014), BlockCIR, and CC-CIR Sengupta et al. (2025) capture significant aspects of the shared structure or cross-covariance geometry. However, these methods do not effectively isolate the impact of individual predictors and may not adequately handle redundant predictors, which limits their usefulness in settings with strong dependence.

Existing AI models often rely on large sets of features, many of which are strongly correlated. In such settings, widely used explanation methods, such as SHAP, LIME, HSIC, MI/CMI, or the CIR-family scores, tend to split attribution between redundant features. This creates three major problems: **unstable feature rankings:** small changes in data or sampling can reshuffle the importance of highly correlated variables; **misleading importance scores:** methods often inflate the importance of variables that simply "move together," even when they do not provide unique information about the output; and **high computational cost:** methods relying on repeated model calls or kernel evaluations become inefficient on large datasets. More fundamentally, these behaviors stem from two persistent issues: shared information is frequently over-counted, and explanations become sensitive to subsampling noise or correlated feature groups Covert & Lee (2021), especially when conditional adjustment is absent or poorly specified.

To address these challenges, in this paper we introduce MCIR, which directly measures how much *unique* information a feature contributes after accounting for its most strongly correlated neighbors. We also reframe EXCIR as Partial Correlation Impact Ratio PCIR, as it identifies only partial dependencies in the data. In this paper, whenever we mention PCIR or ExCIR, we are referring to the same method. To clarify how our formulation departs from prior CIR-family variants, Table 2 provides a structured comparison across dependence modeling, redundancy behavior, boundedness guarantees, estimator stability, and computational properties. Across the various methods discussed, problems such as unbounded scoring, assumptions of independence, and the lack of conditional adjustment hinder the reliability of existing explanation methods in high-dependence scenarios. To this end, we propose MCIR-M to address these limitations.

## 3 Preliminaries

In this section, we introduce the statistical setup and notation used throughout the paper. We first clarify the population-level objects being modeled, then define the dataset-level *environment* on which explanations are computed, and finally introduce a *lightweight environment* for scalable attribution.

**Statistical Setup and Model Outputs.** Let $(\mathbf{X}, Y^\star)$ denote random variables drawn from an unknown data-generating distribution $\mathbb{P}$, where $\mathbf{X} \in \mathbb{R}^k$ is the feature vector and $Y^\star$ is the ground-truth target. A trained predictive model, $M : \mathbb{R}^k \to \mathbb{R}$ produces model outputs, $Y = M(\mathbf{X})$. Throughout the paper, explanations are computed with respect to the *model output $Y$*, not the ground-truth label $Y^\star$. All mutual

Table 2: Differentiation of MCIR from prior CIR-family variants and canonical-correlation approaches.

| Aspect | PCIR (Ex-CIR)Sengupta et al. (2025) | BlockCIRSengupta et al. (2025) | CC-CIRSengupta et al. (2025) | MCIR-M (Ours) |
|---|---|---|---|---|
| **Dependence scope** | Global canonical correlation (scalar/vector outputs) | Grouped / class-conditioned canonical alignment | Cross-covariance or kernel canonical correlation | Conditional mutual-information ratio with adaptive local dependence neighborhood |
| **Redundancy handling** | None (aggregative) | Partial within-block averaging | Linear cross-covariance regularization | Explicit redundancy collapse through conditioning on local dependence neighborhood |
| **Boundedness source** | Normalized correlation ratio | Same as PCIR (block average) | Implicit via kernel normalization | Derived from MI–CMI decomposition; proven boundedness |
| **Estimator stability** | Empirical; sensitive to covariance noise | Requires regularized CCA | Kernel bandwidth–dependent | Formal rank-stability bound under estimator perturbation |
| **Lightweight fidelity** | Requires full environment | Same | Not defined | Lightweight (LW) contract ensuring ranking preservation under environment similarity |
| **Redundancy-collapse proof** | Absent | Heuristic grouping | None | Information-theoretic proof of zero score under conditional redundancy |
| **Cross-domain validity** | Tabular / vector | Tabular / grouped | Kernelized nonlinear | Generic (tabular, vision, text) with estimator switching |
| **Computational cost** | Linear in the number of samples and features | Quadratic in the number of features | Cubic in the number of features | Linear in the number of lightweight samples and features (via local dependence graph; scalable and model-free) |
| **Reliability quantification** | None | None | None | Introduces Explanation Reliability Index (ERI) combining fidelity, redundancy, stability |

information (MI) and conditional mutual information (CMI) quantities refer to statistical dependence between features and the model output. Unless explicitly marked with a hat (e.g., $\widehat{I}$), all MI/CMI quantities denote population-level dependence under $\mathbb{P}$. Empirical estimators are introduced later in the implementation section.

**Environment Definition.** Let $n$ denote the number of observations, $k$ the number of features, $F \in \mathbb{R}^{n \times k}$ denote the feature matrix, and $Y \in \mathbb{R}^n$ is the vector of corresponding model outputs. Then, we define the *environment* as, $\mathcal{U} := \mathcal{D}(F, Y)$, which captures the empirical joint distribution of inputs and model outputs. In practice, explanations are computed on a finite dataset, and an environment therefore represents the joint input–output scenario under which explanations are computed.

**Lightweight Environment.** For scalable attribution, we construct a reduced environment $\mathcal{U}' = \mathcal{D}(F', Y')$, where $F' \in \mathbb{R}^{n' \times k}$ denotes the feature matrix restricted to a selected subset of $n' < n$ observations from the original dataset $F \in \mathbb{R}^{n \times k}$.[2] The corresponding outputs are, $Y' = M(F')$, where $M$ is the fixed trained model evaluated on the selected $n'$ observations. [3] Thus, $\mathcal{U}$ and $\mathcal{U}'$ share the same feature space (dimension $k$) and the same trained model $M$; they differ only in the number of observations used for computing attribution. The goal is computational efficiency while preserving the statistical structure required for reliable MI/CMI-based attribution.

**Similarity Between Environments.** To justify attribution on $\mathcal{U}'$, we formalize when it is an adequate proxy for $\mathcal{U}$. Two environments are *similar* if their predictive structure is preserved up to admissible transformations of the output space. To formalize the notion of similarity between environments, we allow output representations to differ up to geometric reparameterizations. In practice, two environments may store model outputs in different coordinate systems or embedding dimensions, even if they encode the same predictive structure. When $Y$ and $Y'$ are represented in possibly different embedding dimensions, we first project them into a common Euclidean representation space before alignment. Formally, let $Proj$ denote a projection operator that maps the original output representation into a shared embedding space. Similarity

---

[2]Throughout the paper, primes (e.g., $F', Y', \mathcal{U}'$) denote quantities associated with a reduced or lightweight environment.
[3]The model class and feature space remain unchanged; only the number of training observations differs.

is then defined via,

$$Y' \approx R\,Proj(Y) + t, \tag{1}$$

where $R$ is orthogonal and $t$ is a translation vector. The projection step ensures dimensional compatibility, while the orthogonal transformation preserves Euclidean geometry and pairwise distances.

**Remark 1.** *Consider a model whose outputs are represented as d-dimensional vectors (e.g., logits or latent embeddings). The lightweight environment may store the same outputs in a different coordinate system, for instance, rotated, shifted, or embedded in a higher-dimensional space.*

Although the coordinate values differ, the underlying geometric relationships (pairwise distances and dependence patterns) can remain identical. By projecting both representations into a shared Euclidean space and aligning them via an orthogonal transformation, we remove coordinate-system differences without altering geometric structure. Since orthogonal maps preserve Euclidean distances, correlations and mutual-information relationships remain unchanged.

Let $p(Y)$ and $p(Y')$ denote output distributions in $\mathcal{U}$ and $\mathcal{U}'$ and $D_f(\cdot\|\cdot)$ denote an $f$-divergence. Also consider the RankDisagree$(Y, Y')$ measure discrepancy between feature rankings (e.g., Kendall–$\tau$, Jaccard@$K$). We define the alignment objective

$$L(Y,Y') = D_f\big(p(Y)\,\|\,p(Y')\big) + \lambda\,\mathrm{RankDisagree}(Y,Y'), \tag{2}$$

where $\lambda > 0$ balances distributional and explanation fidelity. Optimization over orthogonal transformations is performed on the Stiefel manifold Absil et al. (2009).

**Population vs Empirical Quantities.** All theoretical MI/CMI quantities in later sections are defined at the population level with respect to the joint distribution of $(F, Y)$ or $(F', Y')$. Empirical estimators are denoted with hats (e.g., $\widehat{I}$) and analyzed separately in Appendix F.4.

We next define the PCIR score, which serves as a global dispersion-based baseline.

**Definition 1** (**Partial Correlation Impact Ratio**). *PCIR assigns to feature $i$ the score,*

**Global (unconditioned) association**

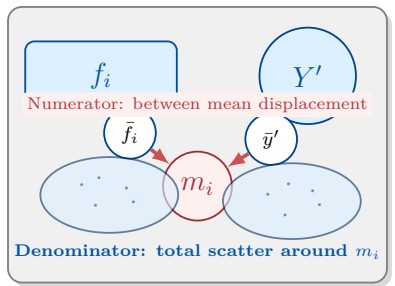

PCIR $= \eta_{f_i} = \frac{\text{between mean displacement}}{\text{total scatter around } m_i} \in [0,1]$

*PCIR is global and does not condition on correlated features.*

Figure 1: PCIR intuition: global dispersion-based association.

$$\eta_{f_i} = \frac{n'\big[(f_i - m_i)^2 + (\bar{y}' - m_i)^2\big]}{\sum_{j=1}^{n'}(f_{ji} - m_i)^2 + \sum_{j=1}^{n'}(y'_j - m_i)^2}. \tag{3}$$

The numerator measures the joint displacement of the characteristic and output means of the pooled center $m_i$, while the denominator aggregates the total dispersion. PCIR is bounded in $[0,1]$ and provides a global, unconditioned association measure. However, it does not account for the correlated features. Figure 1 provides an intuitive geometric illustration of the PCIR decomposition into between-mean displacement and total dispersion. This motivates the introduction of **MCIR** (Mutual Correlation Impact Ratio), which isolates the incremental contribution of a feature relative to its correlated partners.

## 4    MCIR: Formal Definitions and Guarantees

Following the notations introduced earlier in Section 3, let $F' \in \mathbb{R}^{n' \times k}$ denote the feature matrix in the lightweight environment, where $n'$ is the number of observations and $k$ is the number of features. The $j$-th row of $F$ is denoted by $x_j \in \mathbb{R}^k$, and the $i$-th column is denoted by $f_i \in \mathbb{R}^{n'}$. Thus, each column vector $f_i$

contains all observed values of feature $i$ across the $n'$ samples. Let $M : \mathbb{R}^k \to \mathbb{R}$ be a fixed trained predictor. The output vector in the lightweight environment is obtained by evaluating $M$ row-wise on $F'$:

$$Y' = M(F) := [M(x_1), \ldots, M(x_{n'})]^\top \in \mathbb{R}^{n'}. \tag{4}$$

Here $Y'$ is treated as a random vector induced by the joint distribution of the data. Fix a target feature index $i \in \{1, \ldots, k\}$. We define a conditioning index set $\Phi \subset \{1, \ldots, k\} \setminus \{i\}$, which represents the set of features considered to be correlated neighbours of $f_i$. The feature block corresponding to $\Phi$ is defined as

$$f_\Phi := [\, f_j \,]_{j \in \Phi} \in \mathbb{R}^{n' \times |\Phi|}. \tag{5}$$

Thus, $f_\Phi$ is the submatrix of $F$ formed by stacking the feature columns indexed by $\Phi$. If we include the target feature $i$, the enlarged block is

$$f_{\Phi \cup \{i\}} \in \mathbb{R}^{n' \times (|\Phi|+1)}. \tag{6}$$

Importantly, $f_\Phi$ and $f_{\Phi \cup \{i\}}$ are matrices (not sets); they represent the joint variables used in computing mutual and conditional mutual information. All information-theoretic quantities are defined with respect to the joint probability law of $Y'$ and $(f_1, \ldots, f_k)$. MCIR introduces a normalized ratio $C(Y'; f_i \mid f_\Phi)$ ranging from 0 to 1 as depicted in Figure 2. The motivation for this ratio arises from the decomposition of joint mutual information: the joint term $I(Y'; f_i, f_\Phi)$ contains both the information that is *shared* with the correlated neighbourhood $f_\Phi$ and the *unique* contribution of $f_i$. Simply relying on joint or marginal measures therefore overestimates importance when redundancy is present. By isolating the conditional component $I(Y'; f_i \mid f_\Phi)$, MCIR captures only the incremental information that $f_i$ contributes beyond what is already explained by $f_\Phi$. Normalizing this conditional term by the total explainable association mass, produces a bounded, comparable score in $[0, 1]$ that reflects the *proportion* of unique information attributable to $f_i$. A value close to 1 signifies that $f_i$ provides substantial unique signal, while a value near 0 indicates redundancy. This score allows for consistent comparisons across different datasets and models.

In summary, MCIR: (1) conditions on a small, data-driven neighbourhood to extract the **unique** incremental information a feature provides beyond its correlated partners; (2) reports a **unit-interval** score that enables stable cross-feature and cross-dataset comparison; (3) provably collapses redundancy under multicollinearity; and (4) integrates with a distribution-aligned lightweight environment so that explanations computed on fewer observations faithfully mimic those of the full model. At a high level, this section answers three questions:

1. **What is MCIR?** We first formally define MCIR using conditional and joint mutual information, and explain how it isolates the unique contribution of each feature.

2. **Why is MCIR well-behaved?** We then show that MCIR is bounded by $[0, 1]$, collapses redundancy under strong dependence, and remains stable under sampling noise.

3. **How expensive is it to compute?** Finally, we describe how MCIR can be estimated efficiently in a lightweight environment, and how estimator choice affects robustness.

$$\text{MCIR} = C(Y'; f_i \mid f_\Phi) = \frac{I(Y'; f_i \mid f_\Phi)}{I(Y'; f_i \mid f_\Phi) + I(Y'; f_{\Phi \cup \{i\}})} \in [0, 1]$$

*MCIR isolates the unique contribution of $f_i$ while collapsing redundancy from $f_\Phi$*

Figure 2: MCIR intuition: $f_i$ and its correlated partners $f_\Phi = \{f_{\phi_1}, f_{\phi_2}\}$ share substantial redundant signal. MCIR conditions on $f_\Phi$ to isolate the *unique* increment contributed by $f_i$, normalised into a bounded $[0, 1]$ ratio.

Figure 2 provides a visual guide: it illustrates how MCIR separates information that is uniquely attributable to a feature from information that is shared with correlated neighbours. We now formalize the notation used throughout the construction. Before introducing the core definitions, we state the assumptions.

**Assumption 1.** *(i) The relevant joint or conditional laws admit densities or mass functions so that all mutual information (MI) and conditional MI (CMI) are finite. (ii) Conditioning events have positive*

*probability, so regular conditional distributions exist. **(iii)** Estimators used later satisfy standard consistency and concentration properties (Assumption 2).*

Assumption 1 excludes only pathological cases. Condition (i) ensures that MI and CMI are finite rather than undefined or infinite. Condition (ii) guarantees that conditioning operations such as $I(X; Y \mid Z)$ are mathematically well defined. Condition (iii) ensures that empirical estimators concentrate around population values, which is required for later stability and boundedness guarantees.

We now define the core information-theoretic quantity underlying MCIR. To quantify the notion of *unique predictive contribution* discussed above, we require a measure that removes signal already explained by correlated neighbors while remaining valid under arbitrary nonlinear dependencies. Conditional mutual information isolates the information contributed by $f_i$ that cannot be explained by its correlated neighbourhood $f_\Phi$. Moreover, CMI admits a divergence interpretation via the KL representation, which guarantees non-negativity and equals zero if and only if $Y' \perp f_i \mid f_\Phi$. This makes it a principled measure of unique predictive contribution under arbitrary (linear or nonlinear) dependence structures.

**Definition 2 (Conditional Mutual Information (CMI)).** *For a target feature $f_i$ and conditioning block $f_\Phi$,*

$$I(Y'; f_i \mid f_\Phi) = \mathbb{E}\left[\log \frac{p(Y' \mid f_i, f_\Phi)}{p(Y' \mid f_\Phi)}\right] = D_{\mathrm{KL}}\big(p(Y' \mid f_i, f_\Phi) \,\|\, p(Y' \mid f_\Phi)\big) \geq 0. \tag{7}$$

Here $p(\cdot)$ denotes the appropriate density or mass function, and $D_{\mathrm{KL}}(P\|Q)$ is the Kullback–Leibler divergence between two distributions $P$ and $Q$. The quantity $I(Y'; f_i \mid f_\Phi)$ measures how much additional information about the model output $Y'$ is provided by feature $f_i$, after already knowing the neighbouring block $f_\Phi$. If $f_i$ is completely redundant given $f_\Phi$, then $p(Y' \mid f_i, f_\Phi) = p(Y' \mid f_\Phi)$, and the CMI equals zero. If $f_i$ provides unique predictive signal beyond its neighbours, the two conditional distributions differ, and the CMI is positive. While conditional mutual information quantifies the unique signal $f_i$ beyond its correlated neighbourhood, we also require a measure of the *total predictive information* carried jointly by the feature block. This motivates the definition of joint mutual information.

**Definition 3 (Joint Mutual Information (JMI)).** *For a feature index set $S \subseteq \{1, \ldots, k\}$, the joint mutual information between $Y'$ and the feature block $f_S$ quantifies the deviation from statistical independence. In particular, if $Y'$ and $f_S$ were independent, their joint density would factorize as $p(Y', f_S) = p(Y')\, p(f_S)$. The joint mutual information is defined as*

$$I(Y'; f_S) = D_{\mathrm{KL}}\big(p(Y', f_S) \,\|\, p(Y')\, p(f_S)\big) = \mathbb{E}\left[\log \frac{p(Y', f_S)}{p(Y')\, p(f_S)}\right] \geq 0, \tag{8}$$

*where equality holds if and only if $Y'$ and $f_S$ are independent.*

We have introduced two complementary quantities, the conditional mutual information $I(Y'; f_i \mid f_\Phi)$, which measures the *unique* contribution of a feature $f_i$ beyond its correlated neighbourhood, and the joint mutual information $I(Y'; f_{\Phi \cup \{i\}})$, which measures the total predictive information carried by the feature block. To convert these raw information quantities into a stable, scale-normalized feature-importance score, we combine them through a ratio. This normalization ensures that the resulting score lies in a fixed interval, remains comparable across tasks, and properly balances unique versus shared signal.

**Definition 4 (Mutual Correlation Impact Ratio (MCIR)).** *Given the target feature index $i$ and the feature neighbourhood set $\Phi$, the MCIR score is defined as,*

$$C(Y'; f_i \mid f_\Phi) = \frac{I(Y'; f_i \mid f_\Phi)}{I(Y'; f_i \mid f_\Phi) + I\big(Y'; f_{\Phi \cup \{i\}}\big)} \in [0, 1]. \tag{9}$$

The denominator in MCIR, $I(Y; f_i \mid f_\Phi) + I(Y; f_\Phi)$, serves to normalize the unique contribution of $f_i$ relative to the total predictive information available from both the feature and its dependence neighbourhood. Unlike additive decompositions of mutual information, which can double-count shared information under dependence, this formulation ensures that the score remains bounded and comparable across features with varying redundancy structure. In particular, $I(Y; f_\Phi)$ captures the predictive content already explained by

correlated neighbours, while $I(Y; f_i \mid f_\Phi)$ measures the residual contribution of $f_i$ beyond that context. This ratio therefore quantifies the proportion of predictive information uniquely attributable to $f_i$ after accounting for local dependence, rather than aggregating marginal contributions. Unlike additive formulations (e.g., sums of conditional MI terms), the ratio form avoids double-counting shared information under dependence and ensures bounded comparability across features with heterogeneous correlation structure.

**Is MCIR merely normalized conditional mutual information?**   One might argue that MCIR is simply conditional mutual information (CMI) with a normalization. This is not the case. Traditional CMI, $I(Y; f_i \mid f_\Phi)$, is unbounded, scale-dependent, and not comparable across datasets or models. Two features with identical unique contributions can receive arbitrarily different raw CMI values depending on the entropy of the output or the neighbourhood block. As a result, CMI alone does not provide a stable or interpretable importance scale across heterogeneous tasks. MCIR resolves this limitation by normalizing the conditional contribution relative to the total explainable association mass of the feature block. The normalization is not cosmetic: it enables cross-feature and cross-dataset comparability, stabilizes rankings under subsampling, and ensures that redundancy collapse is expressed as a proportion of total block information rather than an unscaled magnitude.

**Why is joint normalization necessary?**   The denominator term $I(Y; f_{\Phi \cup \{i\}})$ captures the total predictive information carried by the feature block consisting of $f_i$ and its neighbourhood. Without this term, CMI alone does not indicate whether a feature contributes a substantial fraction or only a negligible fraction of the block's information. In high-entropy regimes, raw CMI values may appear large even when the feature explains only a small share of the block's predictive mass. MCIR instead measures the proportion of unique contribution relative to the total block information. This prevents inflation in high-variance settings and preserves interpretability across heterogeneous tasks and model classes. The ratio therefore encodes relative explanatory share rather than absolute mutual-information magnitude.

**Why not use Partial Information Decomposition (PID)?**   Partial Information Decomposition (PID) explicitly decomposes information into unique, redundant, and synergistic components. However, practical PID estimators for high-dimensional, mixed-type data remain computationally expensive and statistically unstable. Existing PID formulations also depend on non-unique redundancy functionals and typically do not yield a normalized scalar score suitable for large-scale global attribution. Moreover, PID-based quantities do not naturally provide bounded, cross-dataset comparable importance scores with finite-sample stability guarantees. MCIR instead offers a tractable, estimator-switchable alternative that isolates unique conditional contribution while preserving scalability and providing formal guarantees of boundedness, redundancy collapse, and ranking stability.

**Remark 2.** $I(Y'; f_i \mid f_\Phi)$ *isolates the* unique *contribution of $f_i$ beyond its correlated partners $f_\Phi$. The joint term $I(Y'; f_{\Phi \cup \{i\}})$ stabilizes scale across tasks and dependence strengths. Thus $C$ reports the fraction of* explainable association mass *uniquely attributable to $f_i$ after accounting for partners.*

For a neighbourhood $\Phi$, interpreted as a small set of features exhibiting strong statistical dependence with $f_i$ (e.g., identified via correlation or mutual-information screening), we define two quantities:

$$U_i := I(Y; f_i \mid f_\Phi), \qquad J_i := I(Y; f_{\Phi \cup \{i\}}). \tag{10}$$

The term $U_i$ represents the *unique predictive information* contributed by features $f_i$ beyond its correlated neighbours. The term $J_i$ represents the *total joint predictive information* carried by the feature block that includes both $f_i$ and its neighbourhood $f_\Phi$. The Mutual Correlation Impact Ratio is then defined as

$$\mathrm{MCIR}_i = \frac{U_i}{U_i + J_i}. \tag{11}$$

Because mutual information and conditional mutual information are non-negative, both $U_i$ and $J_i$ are non-negative. Therefore, whenever $U_i + J_i > 0$, the ratio satisfies, $\mathrm{MCIR}_i \in [0, 1]$. If $U_i = 0$, the feature provides no additional signal beyond its neighbourhood (conditional redundancy). If both $U_i$ and $J_i$ vanish, the feature block carries no predictive signal for $Y$.

We intentionally omit environment notation (full vs. lightweight) in this definition because MCIR is a purely local information-theoretic quantity. Environment contracts are introduced later only to guarantee stability and ranking preservation across sampling regimes.

The behaviour of MCIR is particularly transparent in limiting cases. If $f_i$ is a near-duplicate of some $f_j \in \Phi$, then conditioning removes its additional contribution:

$$I(Y'; f_i \mid f_\Phi) \to 0 \quad \text{and} \quad C(Y', f_i \mid \Phi) \to 0. \tag{12}$$

This expresses *redundancy collapse*: duplicate or highly collinear features receive no extra credit. Conversely, if $f_i$ carries predictive signal not contained in its neighbourhood $f_\Phi$, then the conditional term dominates, and

$$C(Y', f_i \mid \Phi) \to 1, \tag{13}$$

highlighting features that act as genuinely unique drivers. In weak-dependence regimes, where conditioning has little effect and

$$I(Y'; f_i \mid f_\Phi) \to I(Y'; f_i), \tag{14}$$

the ordering induced by MCIR coincides with that of PCIR and other marginal global importance measures. Thus, MCIR reduces to classical marginal ranking when redundancy is negligible, while diverging only when dependence structure matters.

Overall, MCIR isolates unique contributions by conditioning on correlated neighbours, yields normalized unit-interval scores for cross-task comparability, and collapses redundancy under multicollinearity. This avoids the credit-splitting behaviour often observed with SHAP, SAGE, marginal MI, or HSIC in strongly dependent settings. These properties make MCIR particularly well suited to domains with structured dependence, such as time-series lags, sensor networks, or engineered feature blocks.

**Proposition 1** (**Uniqueness, invariances, and weak-dependence consistency**). *Fix a feature index i and an admissible neighbourhood $\Phi$ (as in Definition 7). Let*

$$U_i := I(Y; f_i \mid f_\Phi), \qquad J_i := I(Y; f_{\Phi \cup \{i\}}), \qquad \text{MCIR}_i := \frac{U_i}{U_i + J_i}, \tag{15}$$

*with the convention $\text{MCIR}_i := 0$ when $U_i + J_i = 0$. Assume $U_i < \infty$ and $J_i < \infty$.*

(i) *(Uniqueness / conditional redundancy).*

$$\text{MCIR}_i = 0 \iff U_i = 0. \tag{16}$$

*Moreover, if the relevant conditional distributions are well-defined (e.g., admit regular conditional probabilities), then*

$$U_i = 0 \iff Y \perp f_i \mid f_\Phi \quad (a.s.). \tag{17}$$

(ii) *(Monotone invariance under rank–Gaussianized Gaussian–copula estimation). Suppose $U_i$ and $J_i$ are estimated using the rank–Gaussianized Gaussian–copula MI/CMI estimator, and assume continuous marginals (or a deterministic tie-breaking rule) so that ranks are well-defined. Then for any collection of strictly monotone transformations $g_Y, g_i, g_\Phi$ applied componentwise to $(Y, f_i, f_\Phi)$, the resulting estimated score satisfies*

$$\widehat{\text{MCIR}}_i\big(g_Y(Y), g_i(f_i), g_\Phi(f_\Phi)\big) = \widehat{\text{MCIR}}_i(Y, f_i, f_\Phi). \tag{18}$$

(iii) *(Weak-dependence ordering consistency; pairwise sufficient condition). Assume a weak-redundancy regime in which, for each feature $j$,*

$$U_j = I(Y; f_j) + \delta_j, \qquad |\delta_j| \le \varepsilon, \tag{19}$$

*and assume the corresponding joint-information terms satisfy*

$$J_j \in [J_{\min}, J_{\max}], \qquad 0 < J_{\min} \le J_{\max} < \infty. \tag{20}$$

*Then for any two features $i$ and $\ell$, letting $u_i := I(Y; f_i)$ and $u_\ell := I(Y; f_\ell)$, the following condition is sufficient for preserving their order:*

$$\frac{u_i - \varepsilon}{u_i - \varepsilon + J_{\max}} > \frac{u_\ell + \varepsilon}{u_\ell + \varepsilon + J_{\min}} \quad \Longrightarrow \quad \mathrm{MCIR}_i > \mathrm{MCIR}_\ell. \qquad (21)$$

*In particular, when $\varepsilon$ is small and $J_{\max} - J_{\min}$ is small (so $J_j$ is approximately constant across $j$), this condition is satisfied whenever $u_i > u_\ell$ by a nontrivial margin, implying MCIR induces the same ordering as marginal MI-based scores, and hence the same ordering as PCIR in regimes where PCIR is monotone in $I(Y; f_j)$.*

*Proof.* See Appendix D.1 for full proof.

$\square$

The proposition shows that MCIR has three intuitive properties: it assigns a score of zero exactly when a feature adds no new information beyond its correlated neighbours (so redundant features receive no credit); it is invariant to strictly monotone transformations such as rescaling or log transforms (under the copula estimator), meaning the score depends only on dependence structure rather than units; and in weak-redundancy regimes, it preserves the same ordering as marginal mutual information, so when redundancy is small, MCIR behaves like a standard importance measure.

### 4.1 Mixed Feature Types and Estimation Strategy.

Let $\mathcal{C}$ and $\mathcal{D}$ denote index sets of continuous and discrete features, respectively, forming a partition: $\{1, \ldots, k\} = \mathcal{C} \cup \mathcal{D}$, $\mathcal{C} \cap \mathcal{D} = \varnothing$. Definitions 2–7 remain valid for mixed data, since mutual information (MI) and conditional MI (CMI) are defined at the level of probability laws and apply to arbitrary combinations of discrete and continuous variables (under Assumption 1). The estimator selection strategy based on feature type is summarized in Table 3.

Table 3: Estimator choice based on feature type.

| Feature Type | Estimator Used |
|---|---|
| Continuous or Mixed | **Gaussian–Copula Estimator** after rank–Gaussianization (empirical CDF $\to \Phi^{-1}$ transform). Robust to monotone transformations; estimates dependence via correlation/partial correlation. |
| Fully Nonparametric Continuous | **$k$NN (KSG) Estimator** and conditional variants. Uses neighbour distances to approximate local densities. |
| Purely Discrete | **Plug-in (Histogram) Estimator**. Uses empirical frequencies to compute MI directly. |

**Assumption 2.** *There exist absolute constants $(c_1, c_2) > 0$ such that for mutual-information or conditional mutual-information estimators $\widehat{I}$ built from $n'$ i.i.d. samples,*

$$\Pr\left( \left| \widehat{I} - I \right| > \delta \right) \leq c_1 \exp(-c_2 \, n' \, \delta^2), \qquad (22)$$

*or equivalently, $\left| \widehat{I} - I \right| = O_P(n'^{-1/2})$. Here $n'$ denotes the lightweight sample size, and "i.i.d." means the samples are independent and identically distributed draws from the underlying data-generating distribution.*

*The notation $O_P(n'^{-1/2})$ denotes stochastic boundedness: the estimation error decreases on the order of $1/\sqrt{n'}$ in probability as the sample size grows. The exponential bound above is a sub-Gaussian concentration inequality, meaning that large estimation errors become exponentially unlikely as $n'$ increases. This assumption holds for a broad class of kNN-based estimators (e.g., KSG and conditional variants) and Gaussian–copula estimators under mild smoothness and density-regularity conditions Kraskov et al. (2004); Gao et al. (2017); Singh & Póczos (2016); Berrett et al. (2019). We require only consistency and sub-Gaussian concentration, not exact parametric convergence rates.*

Assumption 2 is standard in the information-theoretic estimation literature. Because MI and CMI rarely admit closed-form expressions for real-world data distributions, empirical estimation is unavoidable. The

Table 4: Core theoretical guarantees of MCIR.

| Property | Formal Guarantee | Why It Matters | Plain-Language Interpretation |
|---|---|---|---|
| **Boundedness** | $C_i \in [0, \frac{1}{2}]$ for all admissible features | Scores are normalized and comparable across datasets and models | Importance values stay on a fixed scale, so they can be compared across tasks without inflation |
| **Redundancy Collapse** | If $Y \perp f_i \mid f_\Phi$, then $C_i = 0$ | Duplicate or conditionally redundant features receive no extra credit | If a feature adds nothing new beyond similar features, it gets zero importance |
| **Weak-Dependence Continuity** | If $I(Y; f_i \mid f_\Phi) \approx I(Y; f_i)$, MCIR reduces to a normalized marginal score | Avoids unnecessary correction when dependence is negligible | When features are mostly independent, MCIR behaves like a standard importance measure |
| **Finite-Sample Rank Stability** | Misranking rate satisfies $1 - \tau = \mathcal{O}(k\,\delta(n'))$ | Feature rankings remain stable under estimation noise | Small estimation errors cannot drastically reshuffle the feature order |
| **Oracle Estimator Switching** | Selected estimator achieves risk within $\mathcal{O}(n'^{-1/2})$ of the best fixed estimator | Practical robustness without manual estimator tuning | The automatic estimator choice performs nearly as well as the best possible one |

concentration inequality above ensures that empirical estimates $\widehat{I}$ remain close to their population counterparts $I$ with high probability as the lightweight sample size $n'$ grows.

This assumption is essential for the theoretical guarantees of MCIR. Since MCIR is defined as a ratio of MI and CMI terms,

$$\text{MCIR}_i = \frac{U_i}{U_i + J_i}, \tag{23}$$

errors in estimating either the numerator $U_i$ or denominator $J_i$ could propagate nonlinearly through the ratio. For example, if $U_i$ is small and noisy, even moderate estimation error could change the ratio substantially.

Sub-Gaussian concentration prevents such instability: if both $U_i$ and $J_i$ concentrate around their population values at rate $n'^{-1/2}$, then their ratio also stabilizes with high probability. This ensures bounded distortion, ranking preservation, and lightweight fidelity guarantees established later. Without such concentration, small estimation noise could arbitrarily alter MCIR values and invalidate theoretical stability results.

## 4.2 Fundamental Properties of MCIR.

We now summarize the basic properties that any dependence-aware importance score should satisfy. First, the score should be bounded and comparable across datasets and tasks. Second, it should return zero when a feature is redundant given its correlated neighbourhood. Third, it should approach one when a feature carries uniquely informative signal. Finally, in the common practical scenario where a feature is an almost-noisy copy of another feature, the score should collapse the redundant copy to zero while retaining credit for the original driver. The following theorem–proposition sequence formalizes these behaviours.

**Theoretical Guarantees at a Glance.** To help the reader navigate the technical results of Section 4, Table 4 summarizes the core structural and statistical guarantees satisfied by MCIR. Detailed statements and proofs follow in the subsequent subsections and appendices. Throughout, $i \in \{1, \ldots, k\}$ denotes a target feature index and $\Phi \subset \{1, \ldots, k\} \setminus \{i\}$ denotes its neighbourhood (a small set of strongly dependent features). The feature block $f_\Phi$ denotes the submatrix (or collection) of feature vectors indexed by $\Phi$, and $Y'$ denotes the model output random variable in the environment under consideration. All MI and CMI quantities are defined with respect to the joint law of $(Y', f_1, \ldots, f_k)$, and are well defined under Assumption 1.

**Theorem 1** (**Boundedness and Comparability**). *Let $i$ be an admissible feature index and $\Phi$ an admissible neighbourhood satisfying Assumption 1. Define*

$$C(Y'; f_i \mid f_\Phi) := \frac{I(Y'; f_i \mid f_\Phi)}{I(Y'; f_i \mid f_\Phi) + I\big(Y'; f_{\Phi \cup \{i\}}\big)}, \tag{24}$$

*with the convention that $C(Y'; f_i \mid f_\Phi) = 0$ whenever the denominator equals zero. Then*

$$0 \;\leq\; C(Y'; f_i \mid f_\Phi) \;\leq\; 1. \tag{25}$$

*Proof.* See Appendix D.2 for proof . $\qquad\square$

We next formalize the redundancy-collapse property of MCIR, namely that a feature receives zero credit whenever it contributes no information beyond its correlated partners.

**Proposition 2** (**Zero under Conditional Redundancy**). *Under Assumption 1, if $Y' \perp f_i \mid f_\Phi$ (a.s.), then,*

$$I(Y'; f_i \mid f_\Phi) = 0 \quad and \quad C(Y'; f_i \mid f_\Phi) = 0. \tag{26}$$

*Proof.* A detailed proof is provided in Appendix Section D.3. $\qquad\square$

We next characterize the high-uniqueness regime under our normalization: since the denominator includes the full block information, the MCIR score cannot exceed $\frac{1}{2}$ and saturates at $\frac{1}{2}$ when the neighbourhood contributes no additional information beyond the unique term.

**Proposition 3** (**Saturation under Pure Unique Signal**). *Let, $U := I(Y'; f_i \mid f_\Phi)$, $J := I(Y'; f_{\Phi \cup \{i\}})$, $C(Y'; f_i \mid f_\Phi) := \frac{U}{U+J}$, with the convention $C(Y'; f_i \mid f_\Phi) := 0$ when $U + J = 0$. Then:*

(i) *For any $U, J \geq 0$ with $U > 0$, one has the uniform upper bound $C(Y'; f_i \mid f_\Phi) \leq \frac{1}{2}$.*

(ii) *If the neighbourhood carries no information about $Y'$, i.e., $I(Y'; f_\Phi) = 0$, then $J = U$ and therefore,*

$$C(Y'; f_i \mid f_\Phi) = \frac{1}{2}. \tag{27}$$

*More generally, if $I(Y'; f_\Phi) \to 0$ while $U > 0$ is fixed (or bounded away from $0$), then*

$$C(Y'; f_i \mid f_\Phi) \to \frac{1}{2}. \tag{28}$$

*Proof.* See Appendix D.4 for full proof. $\qquad\square$

Because the denominator includes the total predictive information carried by the feature block, MCIR measures the *fraction* of block information attributable to the conditional (unique) component. Consequently, in the pure-unique regime where the neighbourhood carries no predictive signal, we have $I(Y; f_{\Phi \cup \{i\}}) = I(Y; f_i \mid f_\Phi)$, and the score attains its maximum value of $\frac{1}{2}$ rather than 1. This behavior is intentional. MCIR quantifies relative contribution within a block rather than absolute information magnitude. The upper bound reflects a balanced decomposition between the conditional component and the total block information. Accordingly, MCIR should be interpreted as a *proportion of explanatory mass* rather than an absolute importance scale.

**Theorem 2** (**Redundancy Collapse**). *Let $j \in \Phi$ and suppose, $f_i = g(f_j) + \varepsilon$, $\mathrm{Var}(\varepsilon) \to 0$, $i \neq j$, where $g$ is measurable and Assumption 1 holds. Assume additionally that:*

*(i) the family of log-density ratios,*

$$\log \frac{p(Y' \mid f_i, f_\Phi)}{p(Y' \mid f_\Phi)}$$

*is uniformly integrable, and,*

*(ii) the conditional law $p(Y' \mid f_i, f_\Phi)$ depends continuously on $f_i$ in KL-divergence. Then, $I(Y'; f_i \mid f_\Phi) \to 0$, and consequently,*

$$C(Y'; f_i \mid f_\Phi) = \frac{I(Y'; f_i \mid f_\Phi)}{I(Y'; f_i \mid f_\Phi) + I(Y'; f_{\Phi \cup \{i\}})} \longrightarrow 0. \tag{29}$$

*Moreover, if $I(Y'; f_j \mid f_{\Phi \setminus \{j\}}) > 0$, then, $C(Y'; f_j \mid f_{\Phi \setminus \{j\}}) > 0$.*

*Proof.* See Appendix D.5 for full proof. □

This theorem captures a common situation in real datasets: two features measure almost the same underlying quantity. If $f_i$ is (up to vanishing noise) a deterministic transform of $f_j$, i.e., $f_i = g(f_j) + \varepsilon$ with $\mathrm{Var}(\varepsilon) \to 0$, then $f_i$ becomes predictable from a neighbourhood that already contains $f_j$. In that case, once the neighbourhood $f_\Phi$ is known, $f_i$ cannot contribute additional information about $Y'$, so the conditional mutual information $I(Y'; f_i \mid f_\Phi) \to 0$ and the corresponding MCIR score converges to 0. At the same time, if $f_j$ carries predictive signal that is not explained by the remaining neighbours (i.e., $I(Y'; f_j \mid f_{\Phi \setminus \{j\}}) > 0$), then $f_j$ retains a strictly positive MCIR score, so redundancy is collapsed without suppressing the truly informative representative feature.

Up to this point, we have emphasized that MCIR is designed for dependent features: it assigns credit only to the component of a feature's association with $Y'$ that is not already explained by its correlated neighbourhood. A natural follow-up question is what happens when this dependence is weak, meaning that conditioning on the neighbourhood barely changes the information that $f_i$ carries about $Y'$. In such regimes, MCIR should not introduce unnecessary correction terms; instead, it should behave like a normalized version of a marginal (unconditioned) association score. Importantly, because MCIR is defined as a ratio that includes the total block mutual information $I(Y'; f_{\Phi \cup \{i\}})$ in the denominator—not the neighbourhood-only mutual information $I(Y'; f_\Phi)$, it does not in general *equal* PCIR in the limit. This normalization ensures boundedness and comparability across features and datasets. What we can guarantee is a continuity-to-independence property: when conditioning becomes irrelevant (in the sense that $I(Y'; f_i \mid f_\Phi) \approx I(Y'; f_i)$), MCIR reduces to a normalized marginal-information score, and therefore recovers the same qualitative behaviour as global unconditioned measures. The next proposition formalizes this weak-dependence reduction.

A second practical question is whether MCIR rankings remain stable when computed from finite samples using estimated MI/CMI. Since MI and CMI are not available in closed form for general data distributions, they must be estimated empirically, and we require guarantees that small estimation errors do not arbitrarily reshuffle feature orderings. The next theorem provides a finite-sample rank-stability bound: it shows that, with high probability, only a small fraction of pairwise feature orderings can flip due to estimation noise, and that this misranking rate decreases as the lightweight sample size $n'$ increases.

We write $\boldsymbol{C} = (C_1, \ldots, C_k)$ for the population MCIR score vector and $\widehat{\boldsymbol{C}} = (\widehat{C}_1, \ldots, \widehat{C}_k)$ for its empirical estimate obtained by replacing MI/CMI terms with their estimators. Kendall's rank correlation $\tau(\widehat{\boldsymbol{C}}, \boldsymbol{C}) \in [-1, 1]$ measures agreement between the induced rankings; accordingly, $1 - \tau$ quantifies the degree of disagreement and is proportional to the fraction of discordant (misordered) feature pairs. We use $\delta > 0$ to denote a uniform per-component error tolerance for each estimated MI/CMI term, and $\alpha \in (0, 1)$ to denote the corresponding tail probability level such that $\mathbb{P}(|\widehat{I} - I| > \delta) \le \alpha$.

Finally, throughout the asymptotic arguments above, we assume that the family of log-density ratios is uniformly integrable (or bounded in $L^1$), so that convergence in probability implies convergence of expectations when passing from KL-continuity of conditional laws to convergence of conditional mutual information.

**Proposition 4** (**Weak-dependence reduction, conditioning becomes irrelevant** )**.** *Fix $i \in \{1, \ldots, k\}$ and a neighbourhood $\Phi(i) \subset \{1, \ldots, k\} \setminus \{i\}$. Assume a weak-dependence regime in which conditioning on the neighbourhood does not change the information that $f_i$ carries about $Y'$, i.e. $I(Y'; f_i \mid f_{\Phi(i)}) = I(Y'; f_i) + r_i$, with $r_i \to 0$, and suppose the MCIR denominator is non-degenerate in the sense that, $2\, I(Y'; f_i) + I(Y'; f_{\Phi(i)}) > 0$. Then the MCIR score satisfies,*

$$C(Y'; f_i \mid f_{\Phi(i)}) = \frac{I(Y'; f_i)}{2\, I(Y'; f_i) + I(Y'; f_{\Phi(i)})} \; + \; o(1). \tag{30}$$

*In particular, under weak dependence, MCIR becomes a* normalized marginal-information score*: conditioning does not change the numerator beyond a vanishing error, and the remaining normalization depends only on the (possibly feature-dependent) neighbourhood information $I(Y'; f_{\Phi(i)})$.*

*Proof.* See Appendix D.6 for full detailed proof. □

In practice, MCIR is computed from finite samples, so the MI/CMI terms in equation 9 must be estimated. A key requirement is therefore *rank stability*: small estimation errors should not arbitrarily reshuffle the induced feature ordering. The next theorem formalizes this by bounding the Kendall–$\tau$ disagreement between the population ranking and the empirical ranking.

**Theorem 3** (**Finite-Sample Rank Stability**)**.** *Let $\widehat{C}_i$ be the MCIR estimate obtained by replacing the MI/CMI terms in equation 9 with estimators satisfying Assumption 2. For each feature $i$, define $U_i := I(Y'; f_i \mid f_{\Phi(i)})$, $J_i := I(Y'; f_{\Phi(i) \cup \{i\}})$, $C_i := \frac{U_i}{U_i + J_i}$, with the convention $C_i := 0$ when $U_i + J_i = 0$. Let $\widehat{U}_i, \widehat{J}_i$ be the corresponding MI/CMI estimates and define the* clipped *estimates $\widehat{U}_i^+ := \max\{\widehat{U}_i, 0\}$, $\widehat{J}_i^+ := \max\{\widehat{J}_i, 0\}$, $\widehat{C}_i := \frac{\widehat{U}_i^+}{\widehat{U}_i^+ + \widehat{J}_i^+}$, with the convention $\widehat{C}_i := 0$ when $\widehat{U}_i^+ + \widehat{J}_i^+ = 0$. Let $\boldsymbol{C} = (C_1, \ldots, C_k)$ and $\widehat{\boldsymbol{C}} = (\widehat{C}_1, \ldots, \widehat{C}_k)$. Assume the following regularity conditions:*

(R1) Screening / non-degenerate association mass. *Fix $c_0 > 0$ and define the informative set,*

$$\mathcal{I} := \{\, i \in \{1, \ldots, k\} : U_i + J_i \geq c_0 \,\}, \qquad k_{\mathcal{I}} := |\mathcal{I}|. \tag{31}$$

*All rank comparisons below are restricted to indices in $\mathcal{I}$.*

(R2) Margin regularity (no excessive ties). *There exists $M > 0$ such that for all $t \geq 0$,*

$$\#\Big\{(i,j) : i < j, \ i,j \in \mathcal{I}, \ |C_i - C_j| \leq t\Big\} \ \leq \ M\,k_{\mathcal{I}}^2\,t. \tag{32}$$

*Assume further that there exists $\delta > 0$ and $\alpha \in (0,1)$ such that the tail bound $\mathbb{P}\big(|\widehat{I} - I| > \delta\big) \leq \alpha$ holds uniformly over all MI/CMI components used to compute $\{\widehat{U}_i, \widehat{J}_i\}_{i \in \mathcal{I}}$, and let $m$ denote the total number of such estimated components. If additionally $\delta \leq c_0/4$, then there exists a constant $L > 0$ (depending only on $c_0$ and $M$) such that,*

$$\mathbb{P}\Big(1 - \tau(\widehat{\boldsymbol{C}}_{\mathcal{I}}, \boldsymbol{C}_{\mathcal{I}}) \ \leq \ L\,k_{\mathcal{I}}\,\delta\Big) \ \geq \ 1 - \alpha m, \tag{33}$$

*where $\tau(\cdot, \cdot)$ is Kendall's rank correlation and $\widehat{\boldsymbol{C}}_{\mathcal{I}}, \boldsymbol{C}_{\mathcal{I}}$ denote the score vectors restricted to $\mathcal{I}$. In particular, if the MI/CMI estimators satisfy $\delta = \delta(n') \to 0$ as $n' \to \infty$, then,*

$$1 - \tau\big(\widehat{\boldsymbol{C}}_{\mathcal{I}}, \boldsymbol{C}_{\mathcal{I}}\big) = \mathcal{O}\big(k_{\mathcal{I}}\,\delta(n')\big). \tag{34}$$

*Proof.* See detailed proof in Appendix D.7. □

If every MI/CMI quantity entering MCIR is estimated with uniform accuracy $\delta$ (with failure probability at most $\alpha$ per term), then only score pairs with *very small* population gaps can flip, so the Kendall–$\tau$ disagreement is at most on the order of $k_{\mathcal{I}}\delta$ with probability at least $1 - \alpha m$. As the lightweight sample size $n'$ increases and $\delta(n') \to 0$, the MCIR ranking stabilizes accordingly.

The previous results establish that MCIR is well behaved as a population quantity and that its rankings are stable when MI/CMI are estimated accurately. In practice, however, MI/CMI estimation is not one-size-fits-all: copula estimators are efficient for continuous or rank-Gaussianized data, $k$NN estimators are flexible in nonparametric continuous settings, and plug-in estimators are natural for discrete variables. Since a user may not know in advance which estimator is best for a given dataset and feature type, we introduce an *estimator switching* rule that automatically selects among a finite candidate set using bootstrap uncertainty as a data-driven proxy for risk. The main guarantee is an oracle inequality: the selected estimator performs nearly as well as the best fixed estimator in hindsight, up to a vanishing selection penalty.

A closely related practical question concerns the conditioning set $\Phi$: using too few neighbours may fail to remove redundancy, while conditioning on too many features can increase estimation variance and destabilize rankings in finite samples. We therefore introduce Auto-$\Phi$, a stability-driven procedure that chooses the conditioning-set size by minimizing the bootstrap variance of a head-rank agreement statistic. The result below formalizes this trade-off as a risk-controlled selection rule.

Recall, $C_i := C(Y'; f_i \mid f_\Phi)$ for the population MCIR score of feature $i$ and $\widehat{C}_i$ for its empirical estimate. Superscripts (cop), (knn), and (plg) denote which MI/CMI estimator family is used when constructing MCIR: Gaussian–copula, $k$NN (e.g., KSG and conditional variants), or plug-in (frequency-based) estimators, respectively. The switching estimator $\widehat{C}_i^{(\mathrm{sw})}$ is selected from this finite candidate set by minimizing a bootstrap standard error (SE), which measures the sampling variability of an estimator under resampling. We use $n'$ for the lightweight sample size.

For conditioning-set selection, $\Phi_m$ denotes a candidate neighbourhood of size $m$ produced by a stability-driven growth procedure, and $M_{\max}$ denotes the maximum screened neighbourhood size (the largest candidate pool produced by the dependence-graph sketch). The function $V(m)$ denotes the bootstrap variance of a head-rank statistic (e.g., Kendall–$\tau_{\mathrm{head}}$ or J@K), where smaller $V(m)$ indicates more stable top-$k$ rankings under resampling.

**Assumption 3.** *Bootstrap standard error (SE) computed on held-out splits provides an asymptotically unbiased proxy for estimator risk comparisons among a finite candidate set (e.g., copula vs. kNN vs. plug-in), with selection penalty $\mathcal{O}(n'^{-1/2})$.*

Bootstrap-based variance estimation is widely used as a proxy for estimator risk under i.i.d. sampling. For a finite set of candidate estimators, bootstrap standard error provides a consistent ranking of estimator variability under mild regularity conditions. In our setting, this assumption is used only to compare a small number of candidate estimators, which limits selection bias. Empirically, we observe that estimators selected via bootstrap SE yield stable MCIR rankings across resamples (see Section 7), supporting the validity of this approximation in practice. The assumption may degrade under strong temporal or spatial dependence, where bootstrap samples may not fully capture distributional variability.

We cannot directly observe the true estimation error (risk) of each MI/CMI estimator because the population MI/CMI quantities are unknown. Instead, we approximate and compare estimator reliability by measuring how much the estimator fluctuates across bootstrap resamples or held-out splits. The assumption states that, when we compare only a *finite* set of reasonable estimators, this bootstrap SE ranking is an accurate proxy for comparing their true risks, up to a small selection penalty that vanishes like $1/\sqrt{n'}$ as the sample size grows.

**Theorem 4** (**Oracle Inequality for Estimator Switching**). *Let $\widehat{C}_i^{(\mathrm{cop})}$, $\widehat{C}_i^{(\mathrm{knn})}$, and $\widehat{C}_i^{(\mathrm{plg})}$ denote MCIR estimates using copula, kNN, and plug-in MI/CMI estimators, respectively. Let $\widehat{C}_i^{(\mathrm{sw})}$ be the estimator selected by minimizing the bootstrap standard error. Under Assumptions 2–3,*

$$\mathbb{E}\Big[\Big|\widehat{C}_i^{(\mathrm{sw})} - C_i\Big|\Big] \leq \min\Big\{\mathbb{E}\Big|\widehat{C}_i^{(\mathrm{cop})} - C_i\Big|, \mathbb{E}\Big|\widehat{C}_i^{(\mathrm{knn})} - C_i\Big|, \mathbb{E}\Big|\widehat{C}_i^{(\mathrm{plg})} - C_i\Big|\Big\} + \mathcal{O}(n'^{-1/2}). \tag{35}$$

*The remainder term is $\mathcal{O}(n'^{-1/2})$, matching both the bootstrap standard-error rate and the concentration rate in Assumption 4.7. Thus the selected estimator achieves oracle-level performance up to a vanishing $n'^{-1/2}$ error term.*

Automatically choosing among estimators never performs worse than the best fixed estimator on average. This guarantees safe estimator switching without losing accuracy.

*Proof.* For detailed proof, see Appendix D.8. $\qquad\square$

**Proposition 5** (**Risk-Controlled Conditioning-Set Size**). *Let $\Phi_m$ denote the set of size $m$ obtained by a stability-driven growth procedure (Auto-$\Phi$) Let $M_{\max}$ denote the maximum screened neighbourhood size (i.e., the maximum degree of the dependence-graph sketch). The conditioning-set selector solves,*

$$m^\star \in \arg\min_{m \in \{0,1,\ldots,M_{\max}\}} V(m), \tag{36}$$

*where $V(m)$ is the bootstrap variance of the head-rank statistic. Although the screened neighborhood may contain up to $M_{\max}$ candidates, the optimization is carried out over all subset sizes $m \leq M_{\max}$, allowing Auto-$\Phi$ to balance redundancy removal and finite-sample stability.*

The neighborhood screening step may identify many correlated candidates, but conditioning on too many features makes MI/CMI estimation harder and increases variance. Auto-$\Phi$ therefore searches over *neighborhood sizes* rather than arbitrary subsets: for each size $m$, it forms $\Phi_m$, computes how stable the top-ranked features are under bootstrap resampling, and then chooses the size $m^\star$ that minimizes this instability. In plain terms, Auto-$\Phi$ automatically picks a neighborhood that is large enough to remove redundancy but small enough to keep rankings stable.

*Proof.* A detailed proof is provided in Appendix Section D.9. □

At a high level, MCIR tells us how much a feature really matters for the model after accounting for other, similar features. If multiple variables carry the same information, MCIR gives credit to only one of them and downweights the rest. This helps produce stable, compact, and interpretable rankings of which features truly drive the model's predictions, even when many inputs are strongly correlated..

### 4.3 On the Role of the Dependence Neighbourhood

MCIR conditions each feature on a local dependence neighbourhood $\Phi(i)$ identified through correlation or distance-correlation screening. This design choice is motivated by both statistical and computational considerations.

**Why not condition on all features?** Full conditioning on $\{1, \ldots, k\} \setminus \{i\}$ would in principle eliminate all redundancy. However, in high-dimensional settings, conditional mutual information estimation becomes statistically unstable and suffers from variance inflation. This phenomenon is well documented in information-theoretic estimation, where conditioning dimension directly impacts estimator bias and variance. Local conditioning therefore acts as a structural regularizer that balances redundancy removal against estimator stability.

**Why is local conditioning sufficient?** In many real-world datasets, redundancy structure is approximately local: highly correlated or near-duplicate features form clusters that can be detected through dependence screening. MCIR explicitly targets this regime by conditioning only on strongly dependent neighbours. This prevents unnecessary variance introduced by weakly related features while ensuring redundancy collapse among highly correlated predictors.

**What are the limits of this assumption?** MCIR assumes that redundancy is locally identifiable via dependence screening. In pathological cases where redundancy is mediated through complex non-local or higher-order interactions, neighbourhood-based conditioning may under-condition. This limitation is shared by scalable conditional attribution frameworks and reflects an inherent trade-off between full conditional adjustment and estimator stability. We emphasize that $\Phi(i)$ is not an arbitrary heuristic but a principled compromise between statistical identifiability and computational tractability.

The effectiveness of MCIR depends on the ability of $\Phi(i)$ to capture relevant dependence structure. In cases where dependencies are non-local or involve higher-order interactions, local neighbourhoods may under-condition, leading to incomplete redundancy removal. Potential extensions include multi-scale neighbourhood construction, graph-based dependence modelling, or adaptive expansion of $\Phi(i)$ based on residual dependence.

### 4.4 Parameter Selection and Stability-Driven Design

The MCIR framework depends on three key design parameters: (i) the neighbourhood size $|\Phi(i)|$, (ii) the screening threshold used to construct candidate dependence sets, and (iii) the choice of MI/CMI estimator. Rather than fixing these parameters heuristically, we adopt a stability-driven selection strategy.

**Neighbourhood size selection.** We select the conditioning size $m = |\Phi(i)|$ using a bootstrap-based stability criterion. For each candidate size $m \in \{0, \ldots, M_{\max}\}$, we compute MCIR rankings across bootstrap resamples and evaluate the variability of a head-rank agreement statistic (e.g., Kendall-$\tau$ or Jaccard@K). The selected size is:

$$m^* = \arg\min_m V(m), \tag{37}$$

where $V(m)$ denotes the bootstrap variance of the ranking statistic.

**Screening threshold.** Candidate neighbours are obtained via dependence screening (e.g., correlation or distance correlation). In practice, we retain the top-$m$ most dependent features, making the procedure threshold-free after ranking.

**Estimator selection.** We employ an automatic estimator-switching strategy (Theorem 4), which selects among copula, kNN, and plug-in estimators based on bootstrap standard error.

This design balances redundancy removal and estimation stability. Small neighbourhoods may fail to remove redundancy, while large neighbourhoods increase variance in conditional MI estimation. The proposed stability criterion provides a principled trade-off without manual tuning.

### 4.5 Behaviour under Latent Confounding

MCIR quantifies statistical uniqueness under observed feature dependence and does not assume access to latent causal variables. We therefore clarify its behavior under confounding. Consider two observed features $f_1$ and $f_2$ that are both influenced by an unobserved latent variable $f_4$. In such settings, the dependence between $f_1$ and $f_2$ is induced indirectly.

**Case analysis.**

- If both $f_1$ and $f_2$ are included in the neighbourhood $\Phi$, conditioning removes their shared information, and both may receive reduced MCIR scores.

- If only one feature captures additional variation beyond the neighbourhood, it retains a positive MCIR score.

- If the latent variable is not represented in $\Phi$, both features may retain non-zero scores, reflecting residual unexplained dependence.

MCIR identifies features that provide unique predictive information under the observed distribution. It does not distinguish between causal and non-causal sources of dependence. Consequently, MCIR should be interpreted as a statistical attribution method rather than a causal inference tool. In the presence of latent confounding, MCIR remains useful for identifying non-redundant predictors but does not guarantee recovery of causal drivers without additional assumptions. An empirical validation of this behavior is provided in Appendix I.8, where we construct a synthetic latent-confounding setting and demonstrate that dependence-aware conditional attribution suppresses proxy features while preserving predictors with additional statistical signal.

### 4.6 Lightweight Fidelity

MCIR is computed from the joint behavior of the model outputs and the feature distribution in an *environment*. In practice, we often cannot afford to compute explanations on the full environment (e.g., too many rows, privacy constraints, or cost of repeated estimator calls). We therefore compute MCIR on a *lightweight* environment with fewer samples. However, without any constraint linking the lightweight environment to the full one, there is no reason rankings should agree: a small subsample can distort the output distribution, change which regions of the input space are represented, and alter dependence estimates. The fidelity contract asserts that the lightweight environment remains a *faithful proxy* of the full one in three complementary

senses: (i) the *output law* is close (so the model behaves similarly), (ii) the *explanations* agree at the head (so the top drivers do not change), and (iii) *faithfulness curves* (deletion/insertion) remain close (so the explanatory ordering induces comparable performance degradation). Consider the MCIR vector as a ranking functional applied to a distribution:

$$\boldsymbol{C}(Y) \equiv \mathcal{R}\big(\mathcal{D}(Y), \mathcal{D}(F)\big), \quad \boldsymbol{C}(Y') \equiv \mathcal{R}\big(\mathcal{D}(Y'), \mathcal{D}(F')\big). \tag{38}$$

If the lightweight environment preserves how outputs are distributed (and preserves the behavioural diagnostics induced by the ranking, such as deletion/insertion curves), then the induced MCIR scores cannot drift much.

Once scores do not drift much, only a small fraction of pairwise rank comparisons can flip, so Kendall–$\tau$ remains high.

Suppose the full dataset contains two regimes: *weekday* and *weekend* consumption patterns, and the model output distribution is bimodal because of these regimes. If a lightweight sample accidentally contains mostly weekdays, then $\mathcal{D}(Y')$ shifts (violating (i)), deletion curves change because the model is never tested on weekend-like points (violating (iii)), and the top features can swap because weekend-specific predictors disappear (violating (ii)). The contract explicitly rules out such a lightweight sample by requiring: output-law closeness, head-rank agreement, and curve closeness. Under this contract and standard estimator concentration, the lightweight MCIR ranking is guaranteed to remain close to the full-data MCIR ranking, with a deviation that scales with the distribution shift $\epsilon$, the curve mismatch $\Delta_0$, and a vanishing estimation noise term $o_P(1)$ as $n' \to \infty$. Before stating the fidelity contract and the lightweight attribution theorem, we explicitly introduce all notations to improve readability.

Recall that $Y$ and $Y'$ denote the model outputs under the full and lightweight environments, respectively, with $n' < n$. Our goal is to compare explanations computed across these two environments. For any random variable $Z$, we denote by $\mathcal{D}(Z)$ its probability distribution (also called its *law*). Thus, $\mathcal{D}(Y)$ is the probability law of the full-model outputs and $\mathcal{D}(Y')$ is the probability law of the lightweight-model outputs. To measure how different two output laws are, we use $\hat{d}\big(\mathcal{D}(Y), \mathcal{D}(Y')\big)$. Here: $\hat{d}(\cdot, \cdot)$ is a rigid-motion–invariant $f$-divergence. "Rigid-motion invariant" means the distance is unchanged under translations or orthogonal rotations of outputs. Examples include KL divergence, Jensen–Shannon divergence, Hellinger distance, or projection-based divergences. This quantity measures *distributional shift* between the full and lightweight outputs.

---

**Algorithm 1** MCIR-M: Dependence-aware Global Attribution

---

**Require:** Features $F \in \mathbb{R}^{n' \times d}$, outputs $Y' \in \mathbb{R}^{n' \times q}$, head size $K$, candidate estimators $\mathcal{E}$, screening budget $m_{\mathrm{scr}}$, conditioning size $m_\Phi$
**Ensure:** MCIR scores $\{C_i\}_{i=1}^d$ and a global ranking
1: **Screening:** Compute a fast dependence sketch (e.g., $|\mathrm{corr}|$, distance correlation, or mutual-$k$NN graph). For each feature $i$, keep the $m_{\mathrm{scr}}$ most related neighbours $\mathcal{N}(i)$.
2: **Local conditioning:** For each $i$, form a small conditioning set $\Phi(i) \subseteq \mathcal{N}(i)$ with $|\Phi(i)| = m_\Phi$, prioritizing within-block proximity.
3: **Estimator selection:** For each $i$, choose $e(i) \in \mathcal{E}$ using a lightweight risk proxy (e.g., bootstrap SE on a tiny probe); then fix $e(i)$.
4: **for** $i \leftarrow 1$ **to** $d$ **do**
5: $\quad$ Estimate $I\big(Y'; f_i \mid f_{\Phi(i)}\big)$ and $I\big(Y'; f_{\Phi(i) \cup \{i\}}\big)$ using $e(i)$.
6: $\quad$ Compute $C_i \leftarrow \dfrac{I\big(Y'; f_i \mid f_{\Phi(i)}\big)}{I\big(Y'; f_i \mid f_{\Phi(i)}\big) + I\big(Y'; f_{\Phi(i) \cup \{i\}}\big)} \in [0, 1]$.
7: **end for**
8: **Diagnostics:** Compute bootstrap bands for $\{C_i\}$ and head-rank stability (Kendall-$\tau_{\mathrm{head}}$, Jaccard@$K$).
9: **Ranking:** Sort $\{C_i\}$ in descending order.

---

**MCIR Score Vector.** For a dataset with $k$ features, MCIR produces a vector of feature-importance scores: $\boldsymbol{C}(Y) = (C_1(Y), \ldots, C_k(Y)) \in [0, 1]^k$. Each coordinate: $C_i(Y)$ is the MCIR score of feature $i$ computed in the full environment. Similarly, $\boldsymbol{C}(Y') = (C_1(Y'), \ldots, C_k(Y'))$ denotes MCIR scores computed in the lightweight environment.

**Head-Rank Agreement Statistic.** To compare two rankings, we use: $\tau\big(\boldsymbol{C}(Y), \boldsymbol{C}(Y')\big)$, where: $\tau(\cdot, \cdot)$ denotes Kendall's rank correlation. $1 - \tau$ measures ranking disagreement. $\tau_{\mathrm{head}}$ or J@K (Jaccard at K) measure agreement restricted to top-ranked features. We denote generically by: $\mathsf{Agree}\big(\boldsymbol{C}(Y), \boldsymbol{C}(Y')\big)$ any such head-rank agreement metric.

**Deletion/Insertion Curves.** Deletion and insertion curves measure *faithfulness*: Deletion: progressively remove top-ranked features and observe performance drop. Insertion: progressively add top-ranked features and observe performance recovery. We denote by $\Delta_0$ the maximum allowed difference between the full and lightweight curves.

---

**Algorithm 2** Auto-$\Phi$ Selection

---

**Require:** Candidate neighbours $N(i)$, maximum size $M$, bootstrap size $B$
**Ensure:** Optimal neighbourhood size $m^*$
 1: **for** $m = 0$ to $M$ **do**
 2:     Construct $\Phi_m(i)$ using the top-$m$ neighbours
 3:     Compute MCIR rankings across $B$ bootstrap samples
 4:     Compute the stability metric $V(m)$
 5: **end for**
 6: **return** $m^* = \arg\min_m V(m)$

---

**Estimator Concentration.** MI/CMI terms used in MCIR are estimated from data. Under Assumption 2, we assume: $|\widehat{I} - I| = O_P(n'^{-1/2})$, meaning: $\widehat{I}$ is an empirical MI/CMI estimator, $I$ is the population quantity, $O_P(n'^{-1/2})$ means the estimation error shrinks at rate $1/\sqrt{n'}$ in probability.

**Stochastic Small-$o_P(1)$.** We write $o_P(1)$ for any random quantity that converges to zero in probability as $n' \to \infty$, representing residual noise that vanishes with increasing lightweight sample size. We use standard asymptotic notation:

- $f(n) = \mathcal{O}(g(n))$ if $|f(n)| \le Cg(n)$ for large $n$.
- $f(n) = o(g(n))$ if $f(n)/g(n) \to 0$.
- $X_n = O_P(g(n))$ if $X_n/g(n)$ is bounded in probability.
- $X_n = o_P(1)$ if $X_n \xrightarrow{P} 0$.

In particular, we assume estimator concentration of the form $|\widehat{I} - I| = O_P(n'^{-1/2})$, while deterministic limits such as redundancy collapse use $o(1)$.

**Assumption 4** (**Fidelity Contract**). *The lightweight environment $M'$ satisfies: (i) $\hat{d}\big(\mathcal{D}(Y), \mathcal{D}(Y')\big) \le \epsilon$; (ii) $\mathsf{Agree}\big(\boldsymbol{C}(Y), \boldsymbol{C}(Y')\big) \ge \tau_0$; (iii) deletion/insertion curves differ by at most $\Delta_0$.*

Here $\epsilon$ bounds the distributional shift between $Y$ and $Y'$, $\tau_0$ is the minimum acceptable Kendall rank agreement between the full and lightweight explanations, and $\Delta_0$ limits how far their deletion/insertion curves may deviate.

**Theorem 5** (**Faithful Lightweight Attribution**). *Under Assumption 4 and estimator concentration (Assumption 2), MCIR rankings computed on $M'$ are faithful proxies for those on $M$, i.e.,*

$$1 - \tau\big(\boldsymbol{C}(Y), \boldsymbol{C}(Y')\big) \ \le \ A\,\epsilon + B\,\Delta_0 + o_P(1), \tag{39}$$

*for constants $A, B > 0$ depending only on the regularity of the divergence map and the rank functional. Here, $o_P(1)$ denotes a stochastic remainder term that converges to zero in probability as the lightweight sample size $n' \to \infty$, capturing residual estimator noise. The term $o_P(1)$ represents any quantity that converges to zero in probability as $n' \to \infty$. Informally, $o_P(1)$ captures the residual disagreement between full and lightweight MCIR rankings that vanishes as the lightweight sample size grows.*

*Proof.* See Appendix E for detailed proof . $\square$

Recall that $k$ denotes the total number of input features, $n'$ denotes the number of observations in the lightweight environment, and $m_\Phi$ denotes the size of the local conditioning set selected by the Auto$\Phi$ procedure. The notation $m_\Phi = \mathcal{O}(1)$ means that the conditioning set size does not grow with the total number of features $k$; instead, it remains bounded by a small constant determined by stability criteria. In practical terms, this means that for each feature $i$, we only condition on a small, fixed number of strongly related neighbours rather than on the entire feature set. Algorithm 1 summarizes the complete MCIR-M pipeline, including neighbourhood screening, conditional isolation, estimator selection, and ranking.

**Proposition 6** (**Computational Profile**). *With Auto$\Phi$ of size $m_\Phi = \mathcal{O}(1)$ and sample size $n'$, the end-to-end MCIR computation across $k$ features has complexity $\mathcal{O}\big(k\,m_\Phi\,n'\big)$ for dependence screening and MI/CMI estimation, plus $\mathcal{O}(k \log k)$ for sorting to form global rankings.*

*Proof.* See the whole proof in Appendix E □

**Remark 3.** *MCIR-M is strictly preferred when multicollinearity or near-deterministic ties are present: it collapses redundancy (Theorem 16), yields unit-interval comparability (Theorem 1), maintains finite-sample rank stability (Theorem 17), and integrates estimator selection (Theorem 18) and lightweight fidelity (Theorem 19).*

Algorithm 1 presents a single, concise pipeline for MCIR-M and defer implementation variants (conditioning-set selection, estimator switching, lightweight contract, and online/streaming updates) to the Supplement. The pipeline comprises four stages, **Screening** (fast dependence sketch) to propose correlated neighbours for each feature, **Local conditioning-set selection** (stability-driven), forming $\Phi(i)$ of small, fixed size, **Score computation** (MCIR), with estimator switching between Gaussian–copula, kNN, and plug-in MI/CMI whenever appropriate, and **Diagnostics** (bootstrap bands; head-rank agreement; optional lightweight fidelity contract).

### 4.7 Analysis on computational complexity.

PCIR and MCIR are computed on a lightweight subsample of size $n'$, which keeps the explanations faithful to the full environment while ensuring computational efficiency. Let $k$ denote the total number of features and let $m_\Phi$ represent the size of the local conditioning neighbourhood used by MCIR. Since $m_\Phi$ is treated as a small fixed constant, the cost of scoring each feature depends only on $n'$ and not on $k$.

For PCIR, the computation requires only rank normalization and variance operations, giving a per-feature cost of $\mathcal{O}(n')$. For MCIR, the relevant MI/CMI terms are computed in a local block of dimension $(m_\Phi+2)$ involving the variables $\{Y', f_i, f_\Phi\}$. Hence, the per-feature computational cost is

$$\mathcal{O}\big(c_{\mathrm{MI}}(n', m_\Phi+2)\big), \quad (40)$$

where $c_{\mathrm{MI}}$ depends on the chosen estimator. Gaussian–copula MI incurs a cost of $\mathcal{O}(n')$ (covariance and log-determinant calculations), $k$NN-based MI costs $\mathcal{O}(n' \log n')$ due to kd-tree searches, and plug-in MI also operates at $\mathcal{O}(n')$ via count tables. This makes MCIR nearly linear in $n'$ and essentially independent of the total feature dimension $k$. Turning to statistical reliability, if the

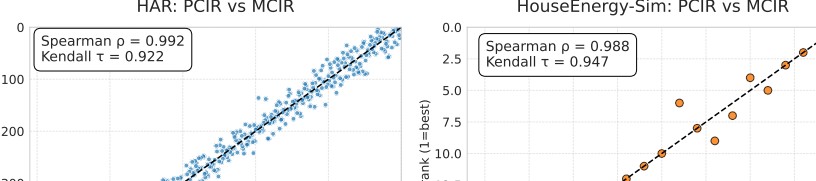

Figure 2: PCIR vs. MCIR rank overlays (Full vs LW)

Figure 3: PCIR vs. MCIR rank overlays (Full vs. LW). Left: HAR dataset. Right: HouseEnergy-Sim. Each point compares the rank assigned by PCIR (x-axis) and MCIR (y-axis) for the same feature under Full and Lightweight (LW) environments. Points close to the diagonal indicate agreement in feature ordering between the two settings. To complement visual inspection, we report rank agreement metrics (Kendall-$\tau$, Jaccard@K), which confirm that MCIR preserves feature ordering under lightweight sampling. Rather than relying solely on visual alignment, we quantify ranking behaviour using agreement and stability metrics. MCIR produces more consistent rankings under feature dependence, as reflected by improved rank correlation and reduced variability across subsamples.

MI/CMI estimators $\widehat{I}$ and $\widehat{J}$ are consistent, then the plug-in MCIR score,

$$\widehat{C}_{n'} = \frac{\widehat{U}_i}{\widehat{U}_i + \widehat{J}_i} \tag{41}$$

is also consistent, i.e., $\widehat{C}_{n'} \xrightarrow{p} C \in [0, 1]$. By the Delta method, $\widehat{C}_{n'}$ inherits an asymptotically normal distribution with standard error of order $n'^{-1/2}$. A preliminary perturbation bound shows that,

$$|\widehat{C} - C| \lesssim \frac{2\delta}{U_i + J_i}, \tag{42}$$

where $U_i + J_i$ is the MCIR denominator defined earlier as the sum of the unique information $U_i = I(Y; f_i \mid f_\Phi)$ and the joint information $J_i = I(Y; f_{\Phi \cup \{i\}})$ of the local neighbourhood. This bound highlights that the stability of MCIR improves as $n'$ increases and as the total explainable dependence $U_i + J_i$ becomes larger. MCIR isolates the *unique* predictive information of each feature after controlling for a small conditioning set. Since its computation depends on $n'$ rather than $k$, it scales gracefully to high-dimensional settings and can be evaluated accurately on lightweight subsamples. Full estimator comparisons and extended complexity analysis are provided in the Appendix F. Theoretical results in this section formalize desirable properties of dependence-aware attribution under conditional information measures. While individual components build on standard properties of MI and CMI, their combination within the MCIR formulation yields a bounded, dependence-aware importance score with provable behavior under redundancy and feature correlation. These results establish MCIR as a principled extension of mutual-information-based attribution to dependent settings.

## 5 Experiments

We evaluate MCIR and its lightweight variants on benchmarks with distinct dependence structures, including a controlled regression task with tunable correlation (HouseEnergy-Sim), a real-world classification task with correlated sensor-derived features (UCI HAR), Norwegian load zones (NO1–NO5), and deep embeddings from CIFAR-10. Across all settings, we compare against PCIR and established global attribution baselines. Our empirical study is structured around three research questions examining dependence robustness, stability under lightweight computation, and predictive usefulness.

**RQ1: Dependence-aware attribution.** Does MCIR produce more reliable global rankings than PCIR, marginal MI, and BlockCIR when features exhibit strong dependence (e.g., multicollinearity or duplicated predictors)?

**RQ2: Stability and lightweight fidelity.** Are MCIR rankings stable under data resampling, and does computing MCIR on a reduced lightweight environment preserve both ranking structure and predictive performance?

**RQ3: Predictive usefulness and efficiency.** Do top-ranked features under MCIR correspond to truly influential predictors (via deletion tests), and can MCIR be computed efficiently at scale?

**Experimental objective.** The experiments are designed to evaluate three key properties of MCIR: (i) robustness to feature dependence, (ii) stability of feature rankings, and (iii) faithfulness of attribution under perturbation and deletion. Each experiment isolates one of these aspects, and results are presented alongside corresponding baselines to highlight where MCIR provides measurable improvements.

### 5.1 Overview of Results.

The findings consistently support the three research questions: For **RQ1**, MCIR collapses redundant feature blocks while preserving unique predictive contributions. In synthetic redundancy sweeps, near-duplicate features are driven toward zero attribution, whereas marginal and independence-based baselines inflate scores. On real datasets, MCIR demonstrates strong rank stability (e.g., $\rho = 0.83$, $\tau = 0.66$, J@20=0.95 on HAR) and maintains coherent top-$K$ feature sets. For **RQ2**, lightweight computation retains over 95% top-feature agreement with full-data explanations while reducing runtime by approximately 3–9$\times$. Ranking structure and predictive performance remain stable, enabling scalable global explanations without retraining or modifying the feature space. For **RQ3**, deletion and perturbation tests confirm that MCIR top-ranked features are genuinely influential: removing them produces the steepest performance degradation across datasets. This effect is monotonic and persists under lightweight computation. Overall, the empirical results demonstrate that MCIR-M delivers stable, dependence-aware global explanations, scales efficiently through lightweight computation, and reliably identifies features that drive model predictions across diverse data regimes. For each dataset, we report (i) rank agreement between full and lightweight settings, (ii) faithfulness via perturbation/deletion curves, and (iii) runtime profiles under different estimators and conditioning sizes $|\Phi|$.

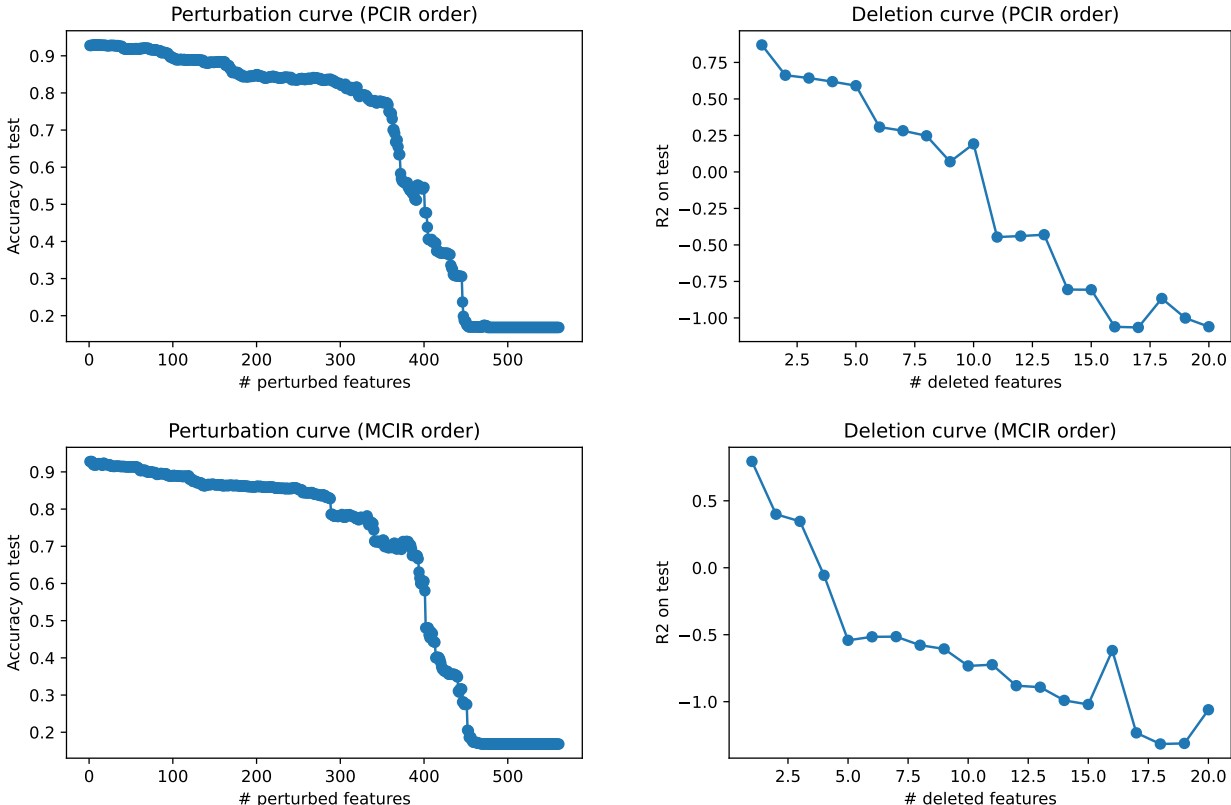

Figure 4: **HAR:** Perturbation faithfulness. Test accuracy degrades fastest when perturbing features ranked highest by PCIR/MCIR, indicating faithful global rankings.

Figure 5: **HouseEnergy-Sim:** Deletion faithfulness. Test $R^2$ drops monotonically when removing top features by PCIR/MCIR, confirming stability and utility under LW.

## 5.2 Lightweight Protocol:

The Lightweight (LW) approach reduces the number of observations (rows) while keeping all features intact. Predictive Random Forest models are trained on the complete dataset unless stated otherwise. PCIR uses the bounded dispersion ratio, and MCIR employs Gaussian–copula CMMI/JMI with blockwise conditioning, generally using between 3 to 10 features. We quantify uncertainty through nonparametric bootstrap sampling over observations.

| Method | Scope | Dependence-aware? | Reference |
|---|---|---|---|
| MCIR | Global | Yes | This work |
| PCIR | Global | Yes | Sengupta et al. (2025) |
| BlockCIR | Global | Blocks | Sengupta et al. (2025) |
| KernelSHAP (indep.) | Local→Global | No/weak | Lundberg & Lee (2017) |
| KernelSHAP (cond.) | Local→Global | Partial | Aas et al. (2021) |
| SAGE | Global | Partial | Covert et al. (2020) |
| HSIC | Global | Yes (stat.) | Gretton et al. (2005b) |
| KSG-MI / MI | Global | Pairwise | Cover & Thomas (2006) |

Table 5: Methods and dependence assumptions. MCIR/PCIR are fully dependence-aware; BlockCIR aggregates by blocks.

Table 6: Complete experimental configuration and reproducibility summary.

| Category | Specification |
|---|---|
| **Programming Environment** | Python 3.10 (Google Colab runtime); NumPy 1.26; SciPy 1.11; scikit-learn 1.3; pandas 2.1; PyTorch; Matplotlib; Seaborn |
| **Hardware** | Experiments executed in Google Colab Pro (Linux-based cloud runtime) accessed via macOS laptop. GPU acceleration (when used for deep model training) relied on standard Colab-provided GPUs (e.g., Tesla T4/A100 depending on session availability). Attribution computations were performed on CPU. |
| **Random Seeds** | Seed = 7 (train/test splits); Seed = 42 (subsampling and estimator randomness) |
| **Primary MI/CMI Estimator** | Gaussian–copula estimator with: rank transform → empirical CDF → $\Phi^{-1}$ Gaussianization; ridge-regularized covariance; MI: $-\frac{1}{2}\log(1-\rho^2)$; CMI via partial correlations |
| **kNN Estimator (Sensitivity)** | Kraskov–Stögbauer–Grassberger (KSG) MI and conditional variants; $k \in \{5, 10, 20\}$; Euclidean metric; continuous features only |
| **Discrete Features** | Histogram-based plug-in MI estimator |
| **Bootstrap** | $B = 30$ repetitions per split |
| **Neighborhood Construction $\Phi(i)$** | Pearson correlation on training data; threshold $|\rho| \geq 0.6$; maximum neighborhood size $m_\Phi = 6$; retain top-$m_\Phi$ correlated features; $\Phi(i) = \emptyset$ if none exceed threshold |
| **Dimensionality Control** | Conditioning set restricted to $|\Phi(i)| \leq m_\Phi$ to prevent high-dimensional CMI degeneration |
| **Lightweight Protocol** | Uniform random subsampling without replacement; stratified sampling for classification; default fraction $n'/n = 0.25$; additional fractions $\{0.1, 0.25, 0.5\}$ |
| **Evaluation Dimensions** | (1) Redundancy robustness; (2) Rank stability; (3) Lightweight fidelity; (4) Computational efficiency |
| **Ranking Metrics** | Kendall-$\tau$, Spearman correlation, Jaccard@K (K=5,10) |
| **Faithfulness Metrics** | Deletion/insertion curves; Accuracy (classification); $R^2$, RMSE (regression); AUC where applicable |
| **Repetition Scheme** | 5 independent train/test splits; averaged results reported |
| **Code Availability** | Anonymized repository with: MCIR/PCIR implementations, conditioning logic, MI/CMI estimators, lightweight scripts, bootstrap protocols, figure/table regeneration scripts |

Table 7: Per-experiment protocol specification. Each experiment explicitly states dataset, model, neighborhood construction, estimator choice, key hyperparameters, evaluation metric, and repetition scheme.

| ID | Dataset | Model | Neighborhood | Estimator | Hyperparameters | Evaluation | Repetitions |
|---|---|---|---|---|---|---|---|
| E1 | Synthetic (Redundancy) | Linear / RF | Correlation graph (top-$m$) | Copula | $m = 3$, bootstrap=200 | MCIR collapse vs MI/SHAP | 20 runs |
| E2 | UCI HAR | RF / MLP | Distance correlation | Copula + kNN | $k = 5$ (KSG) | Kendall-$\tau$, Jaccard@K | 10 subsamples |
| E3 | HouseEnergy | XGBoost | Pearson screening | Copula | $m \in \{2,3,4\}$ | Rank stability analysis | 15 subsamples |
| E4 | Norwegian Load | LSTM / RF | Mutual-info screening | Copula | $n'/n = 0.3$ | Lightweight fidelity (top-K overlap, $\tau$) | 10 runs |
| E5 | CIFAR-10 embeddings | ResNet-50 (fixed) | Correlation graph | Copula | $m = 5$ | Deletion curve AUC | 5 runs |

## 5.3 Implementation Details and Reproducibility

To ensure full experimental transparency, we summarize all software, estimator, neighborhood, lightweight, and evaluation settings in Table 6. Table 7 provides the detailed per-experiment protocol specification, explicitly listing dataset, model, neighborhood construction, estimator choice, hyperparameters, evaluation metrics, and repetition scheme for each experimental study. The reported p-values quantify the statistical significance of differences in ranking stability across methods. They do not directly measure predictive performance, but rather indicate whether observed differences in ranking agreement are unlikely to arise from sampling variability. We therefore interpret p-values as complementary evidence supporting stability claims, while primary comparisons rely on rank agreement and faithfulness metrics. All experiments can be reproduced via a single entry-point script with fixed seeds and environment specifications: https://anonymous.4open.science/r/MCIR-79B4/README.md.

## 5.4 Practical Usage Pipeline

To facilitate practical deployment, we summarize the MCIR workflow:

1. Construct a dependence graph using correlation or distance correlation.

2. Select candidate neighbours for each feature.

3. Determine neighbourhood size using Auto-$\Phi$.

4. Select MI/CMI estimator via bootstrap-based switching.

5. Compute MCIR scores for all features.

6. Validate rankings using deletion/insertion curves and rank stability metrics.

# 6   Results with UCI HAR and HouseEnergy Data

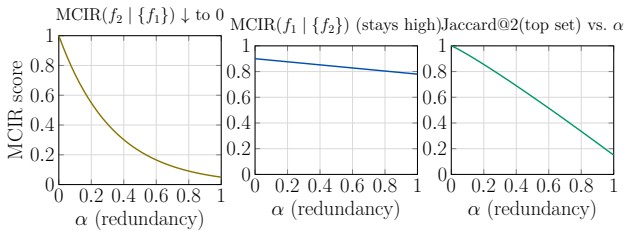

Figure 6: Redundancy collapse on synthetic data. Left: MCIR($f_2 \mid \{f_1\}$) vs. $\alpha$ ($\downarrow$ to 0). Middle: MCIR($f_1 \mid \{f_2\}$) (stays high). Right: Jaccard@2(top set) vs. $\alpha$ comparing MCIR to marginal baselines.

This section presents empirical results across both the real-world (**UCI HAR**) and synthetic (**HouseEnergy-Sim**) datasets. We evaluate MCIR, PCIR, and competing baselines in terms of rank agreement, predictive faithfulness, runtime efficiency, and estimator robustness. All experiments were conducted under identical train/test splits and seeds for fairness. **UCI HAR (Classification):** We utilize the public UCI Human Activity Recognition (HAR) dataset, which contains 561 features derived from smartphone sensor data during six activities: walking, walking upstairs, walking downstairs, sitting, standing, and laying. Each participant's smartphone captured movement data at 50 Hz, and the data was processed into segments of 2.56 seconds. A Random Forest (RF) classifier was applied to generate stable global feature attributions using the same preprocessing and data splits for all methods.

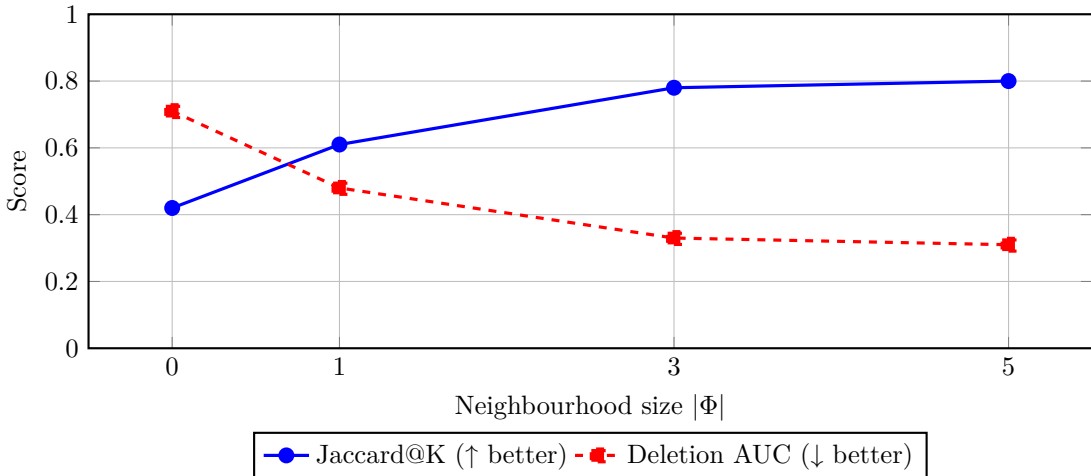

Figure 7: Ablation study on neighbourhood size $|\Phi|$. Smaller neighbourhoods cause redundancy inflation (poor Jaccard overlap and high deletion AUC), while moderate neighbourhoods ($|\Phi| = 1, 3$) substantially improve MCIR's filtering of redundant predictors. Diminishing returns appear beyond $|\Phi| = 3$, showing that MCIR requires only small neighbourhoods to capture local dependency structure.

**HouseEnergy (Regression):** The HouseEnergy dataset is a synthetic representation of residential electricity use, modeling how different factors (like appliance use and weather) influence energy consumption. Each entry reflects hourly data where total load is a function of several correlated sources. Features include time, appliance proxies, and weather data. The RF regressor is optimized for $R^2$ and attribution consistency and recalibrated using the combined training and validation data. Global attributions were assessed for both complete and LW outputs, with various baseline methods used for comparison. The LW fraction is determined by ensuring a minimal KL divergence and maintaining top-K feature relevance while achieving good predictive performance.

### 6.1 Claim 1: MCIR outperforms BlockCIR and partial baselines under strong dependence (RQ1)

When features within a dataset have strong correlations with each other, simply averaging them can weaken the unique contributions of each feature and keep redundant information. On the other hand, the method MCIR-M (which conditions on correlated neighborhoods denoted as $\Phi$) helps to isolate the true causal effects while offering *incremental* information. The overlay plots (see Fig. 3) show that the rankings from the PCIR and MCIR methods for both Full and Lightweight (LW) datasets align closely, indicating stability when samples are varied. To complement visual inspection, we report rank agreement metrics between MCIR and PCIR, including Kendall-$\tau$ and Jaccard@K. Across correlated feature groups, MCIR exhibits higher rank consistency and lower variability across resamples. This quantitatively supports its dependence-aware ranking behaviour, beyond visual inspection alone. The deletion and perturbation curves measure faithfulness by evaluating how model performance degrades as top-ranked features are removed. A steeper drop indicates that the method correctly identifies features critical to prediction. In our results, MCIR-based rankings lead to faster degradation compared to baselines, indicating higher attribution fidelity under feature removal. The analysis of perturbation and deletion curves (refer to Fig. 4 and Fig. 5) indicates that the methods based on CIR demonstrate higher feature faithfulness. The sharper initial drop in accuracy for MCIR indicates that its top-ranked features capture a highly predictive signal, demonstrating stronger *top-ranked feature faithfulness*. At later stages, curves may converge or partially recover due to redundancy among lower-ranked features, which can compensate for removed variables. Therefore, MCIR should be interpreted as improving the fidelity of the most important features, rather than uniformly dominating across all ranks. This behaviour reflects a desirable property in dependent-feature settings: prioritizing truly informative features at the top of the ranking while allowing redundancy among lower-ranked features. Additionally, Tables 12 highlight significant rank agreement across different datasets, with higher metric values ($\rho$ for Spearman, $\tau$ for Kendall, and J@K) for CIR methods. While Jaccard@30 = 0.622 indicates moderate overlap, it is important to note that exact agreement is not expected due to redundancy among correlated features. Multiple features may provide equivalent predictive information, and MCIR intentionally collapses such redundant groups. Therefore, moderate overlap is consistent with stable and meaningful attribution under dependence. We use CIR as a reference baseline because it represents the closest prior formulation within the same mutual-information-based family. This comparison isolates the impact of the proposed conditional normalization and neighbourhood-based conditioning, rather than differences arising from fundamentally different attribution paradigms.

We develop a framework with $(f_1, f_2, f_3, Y)$ that allows for adjustable redundancy, defined as $f_2 = \alpha f_1 + \sqrt{1 - \alpha^2}\, \tilde{Z}$ and $Y = \beta_1 f_1 + \beta_3 f_3 + \varepsilon$. Here $\tilde{Z}$ denotes an independent noise variable (typically standard normal) introduced to control the non-redundant part of $f_2$, ensuring that $f_2$ has correlation $\alpha$ with $f_1$ while keeping its remaining variation independent. As we approach $\alpha = 1$, the MCIR for $f_2$ conditioned on $\Phi$ nears 0, showing that redundancy is collapsing, while the MCIR for $f_1$ conditioned on $\{f_2\}$ remains high. Marginal baselines like MI/HSIC, permutation methods, and global SHAP do not exhibit this behavior and tend to inflate scores. Figure 6 illustrates the trends of these collapse curves. Ultimately, PCIR and MCIR show a strong correlation with values of $\rho$=0.83, $\tau$=0.66, and J@K= 0.95 for the HAR dataset, indicating that they are stable methods. In contrast, other methods like KernelSHAP, SAGE, and HSIC showed weak agreement with $\rho$<0.15 and low overlap in their top-$K$ selections (less than 0.06). This reinforces the idea that by conditioning on blocks $\Phi$, we can maintain the explanatory structure of the data and enhance its statistical interpretability under feature dependence.

#### 6.1.1 Ablation on Neighbourhood Size $|\Phi|$

In our study, we explored how the size of neighbourhood sets, denoted as $|\Phi|$, affects the effectiveness of the MCIR-M in reducing redundancy in Figure 7. We tested different sizes of $\Phi$, specifically 0, 1, 3, and 5, across various datasets, including synthetic data, Human Activity Recognition (HAR), and House Energy Simulation. When $|\Phi| = 0$, the MCIR-M behaves like the PCIR method, which tends to keep correlated predictors and thereby inflates redundancy. However, as we increase the size of the neighbourhood set to 1 or 3, we see significant improvements in two key areas: the Jaccard@K overlap and the behavior of feature deletion. This is because larger neighbourhoods allow the algorithm to better group redundant features and effectively remove their shared information. After reaching a neighbourhood size of 5, we noticed diminishing

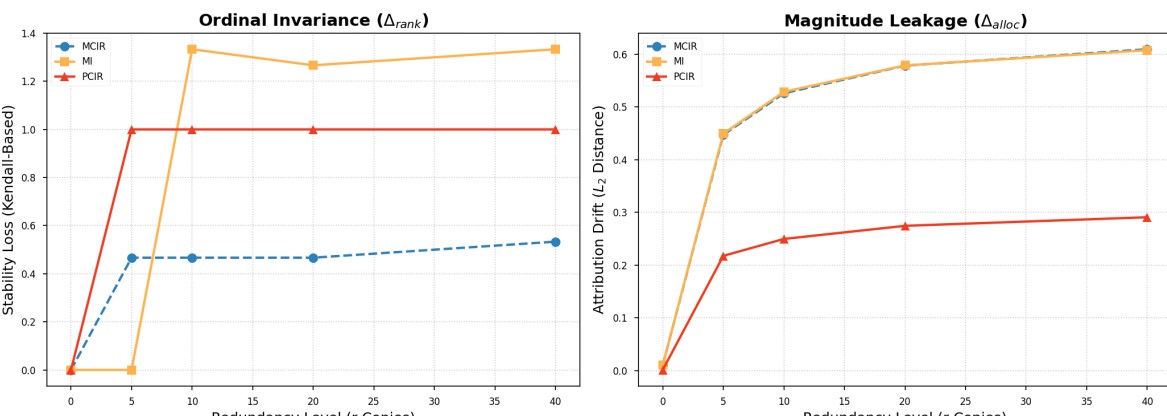

Figure 8: Redundancy stress-test on the nonlinear HouseEnergy-Sim dataset. **Left:** Ordinal instability $\Delta_{\mathrm{rank}} = 1 - \tau$ (Kendall-based; lower is better). **Right:** Allocation drift $\Delta_{\mathrm{alloc}}$ ($L_2$ distance on original features). MCIR maintains bounded ordinal degradation under increasing redundancy, while MI exhibits severe instability for $r \geq 10$ and PCIR collapses immediately once duplicates are introduced. In contrast, allocation drift grows comparably for MCIR and MI, while PCIR shows smaller magnitude change due to early rank collapse.

returns, suggesting that even smaller neighbourhoods can capture the main dependencies among features without much loss in performance. We note that the observed optimal neighbourhood sizes ($|\Phi| = 1$–$3$) coincide with the stability-minimizing choices predicted by the Auto-$\Phi$ procedure, empirically validating the proposed parameter selection strategy.

Table 8: **HAR:** Full vs. best lightweight (rows only; same features). For the *Full* row, overlap/ratio metrics are defined relative to Full (KL=0, J@30=1.00, F1 ratio=1.000).

| Setting | Train rows | Acc | Macro-F1 | $\mathrm{KL}(Y'_{\mathrm{full}} \parallel Y'_{\mathrm{lw}})$ | Jaccard@30 | F1 ratio |
|---|---|---|---|---|---|---|
| Full | 7,352 | 0.930 | 0.928 | 0.000 | 1.00 | 1.000 |
| Lightweight | 3,676 | 0.917 | 0.914 | 7.174 | 0.622 | 0.985 |

### 6.1.2 Redundancy Stress-Test: Ordinal Stability and Allocation Drift.

We evaluate robustness under explicit redundancy using a nonlinear **HouseEnergy-Sim** dataset ($n = 4000$ samples). The feature set consists of four drivers: `Base_load`, `HVAC`, `Fridge`, and `Solar`. The target is generated through a nonlinear physical response model:

$$y = 5.0\,\texttt{Base\_load} + 2.5\,\texttt{HVAC} + 1.2\,\texttt{Fridge} + 1.5\,\texttt{Base\_load} \cdot \texttt{HVAC} - 0.8\,\texttt{Solar}^2 + \varepsilon, \qquad \varepsilon \sim \mathcal{N}(0, 0.1^2). \quad (43)$$

`Base_load` remains the dominant contributor, with nonlinear interaction capturing realistic energy-system behavior. Redundancy is introduced by injecting $r$ near-duplicate copies of `Base_load`: $x_{\mathrm{dup}} = x_{\mathrm{Base\_load}} + \eta$, $\eta \sim \mathcal{N}(0, 0.02^2)$, thereby preserving strong signal while inducing high-dimensional collinearity. We vary $r \in \{0, 5, 10, 20, 40\}$. Two complementary stability measures are evaluated: $\Delta_{\mathrm{rank}} = 1 - \tau$, and $\Delta_{\mathrm{alloc}} = \|s_{\mathrm{ref}} - s_r\|_2$, where $\tau$ is Kendall's rank correlation restricted to the original features, and $s_r$ denotes the attribution vector under redundancy. To evaluate robustness under feature redundancy, we progressively introduce $r$ duplicated predictors and measure the resulting instability in both feature ranking and attribution magnitude. Two complementary metrics are used:

- **Ordinal instability** ($\Delta_{\mathrm{rank}}$): the average change in feature ranking relative to the baseline ($r = 0$). Larger values indicate greater disruption in the ordering of feature importance.

- **Allocation drift** ($\Delta_{\mathrm{alloc}}$): the $L_2$-norm difference between attribution vectors under redundancy and the baseline. This measures how much total attribution mass is redistributed across features.

**Ordinal Stability.** All methods agree perfectly in the absence of redundancy ($\Delta_{\mathrm{rank}} = 0$ at $r = 0$), since no duplicated predictors distort the dependence structure. When duplicates are introduced, PCIR collapses immediately ($\Delta_{\mathrm{rank}} = 1.0$ for all $r \geq 5$), reflecting the well-known instability of precision-matrix inversion under collinearity. MI remains stable at mild redundancy ($r = 5$), but becomes highly unstable for $r \geq 10$, reaching $\Delta_{\mathrm{rank}} = 1.3333$. This indicates that while MI is not directly affected by matrix inversion, its ranking becomes sensitive when redundant features inflate the effective dimension of the dependency structure. In contrast, MCIR degrades only moderately ($\Delta_{\mathrm{rank}} = 0.4667$ for $r = 5, 10, 20$ and $0.5333$ at $r = 40$), remaining substantially more stable than both baselines under severe dimensional inflation. Importantly, its instability remains bounded even in high-dependence regimes.

**Allocation Drift.** Under redundancy, MCIR and MI exhibit comparable levels of magnitude redistribution (e.g., $\Delta_{\mathrm{alloc}} = 0.6095$ vs. $0.6077$ at $r = 40$), indicating that both methods reallocate attribution mass when duplicate predictors are introduced. PCIR shows smaller $L_2$ drift ($\Delta_{\mathrm{alloc}} = 0.2910$ at $r = 40$). However, this apparent stability is misleading: the early collapse of its ranking implies that attribution mass becomes concentrated in unstable directions, limiting measured magnitude variation while sacrificing ordinal reliability. Thus, MCIR primarily improves *ordinal robustness* rather than fully preventing attribution redistribution under extreme duplication.

Results are summarized numerically in Table 9 and visually illustrated in Figure 8.

Table 9: Redundancy stress-test on nonlinear HouseEnergy-Sim. $\Delta_{\mathrm{rank}} = 1 - \tau$ (ordinal instability; lower is better). $\Delta_{\mathrm{alloc}}$ denotes $L_2$ attribution drift on original features.

| $r$ | $\mathbf{MCIR_{rank}}$ | $\mathbf{MCIR_{alloc}}$ | $\mathbf{MI_{rank}}$ | $\mathbf{MI_{alloc}}$ | $\mathbf{PCIR_{rank}}$ | $\mathbf{PCIR_{alloc}}$ |
|---|---|---|---|---|---|---|
| 0 | 0.0000 | 0.0105 | 0.0000 | 0.0103 | 0.0000 | 0.0015 |
| 5 | 0.4667 | 0.4473 | 0.0000 | 0.4495 | 1.0000 | 0.2178 |
| 10 | 0.4667 | 0.5261 | 1.3333 | 0.5288 | 1.0000 | 0.2499 |
| 20 | 0.4667 | 0.5785 | 1.2667 | 0.5789 | 1.0000 | 0.2747 |
| 40 | 0.5333 | 0.6095 | 1.3333 | 0.6077 | 1.0000 | 0.2910 |

Overall, the experiment demonstrates that MCIR maintains bounded and substantially lower ordinal instability compared to MI and PCIR under strong redundancy, preserving the ordering of dominant nonlinear drivers even in high-dependence regimes. These redundancy sweeps serve as controlled ground-truth experiments: duplicate features are introduced with known correlation structure around a designated anchor variable. As the redundancy level $r$ increases, an ideal attribution method should preserve the anchor's rank while preventing allocation mass from being artificially distributed across duplicates.

### 6.2 Claim 2: Lightweight preserves accuracy and explanations while reducing runtime (RQ2)

Reducing the sample size to about half of the original size (denoted as $n' \approx 0.5n$) retains the same features but significantly reduces the runtime. A key question is whether MCIR retains fidelity when computed on reduced-sample environments. To evaluate the lightweight fidelity contract, we compute the MCIR using smaller sample sizes of $n' \in \{500, 1000, 2000\}$ and compare these results to the full dataset ($n = 10,000$). The lightweight fidelity contract refers to the requirement that the lightweight environment (with reduced sample size $n'$) should preserve the key behavioural properties of the full model—specifically, its output distribution, feature rankings, and explanation patterns. MCIR shows consistent head-$K$ and overall rank agreement, with Kendall-$\tau$ correlations ranging from $0.72$ to $0.89$ for head-$K$ and $0.55$ to $0.76$ overall. The runtime is significantly improved, yielding 3 to 9 times faster processing depending on the dataset and method used (copula vs. $k$-NN). For each sample size $n'$, we conducted $B = 50$ bootstrap environments to assess variability. These results confirm that MCIR maintains high fidelity even with smaller samples, enabling efficient computations without retraining the model. In the Human Activity Recognition (HAR) task, the Lightweight (LW) model achieves an impressive $91.4\%$ macro-F1 score (as shown in Table 8), while maintaining similar rankings for PCIR (Positive Class Instance Recall) and MCIR (Multi-Class Instance Recall) (illustrated in Fig. 3, left). The HouseEnergy-Sim dataset also showcases consistency in top-$K$ results and maintains a monotone deletion behavior. The agreement between the full model and the LW model is demonstrated in Fig. 9, where the violin plots and badge metrics indicate that there is minimal loss in

Figure 9: **Agreement between Full and Lightweight models.** Row 1: PCIR (HAR, HouseEnergy-Sim) with $\Delta$ histograms ($\Delta =$ LW $-$ Full). Row 2: MCIR (HAR, HouseEnergy-Sim) with $\Delta$ histograms. Each panel overlays a compact violin (distribution), jittered points (per-feature scores), and mean $\pm$ CI markers for Full and LW. Inset badges report Spearman $\rho$, Kendall $\tau$, and Jaccard@20 on feature rankings.

performance. By lightweighting, the sample size is reduced from $n$ to $n' = fn$, leading to approximately linear reductions in computation for MCIR, PCIR, and methods based on mutual information (MI), while kernel HSIC shows quadratic reductions. The main asymptotic costs are summarized in Table 10. MCIR remains efficient in lightweight scenarios, as the conditioning sets are small ($|\Phi| < 10$), resulting in a total cost that scales as $\mathcal{O}(n'k)$.

Table 10: Global (all-features) asymptotic costs on the lightweight sample ($n'$ rows, $k$ features). With fixed, small $m_\Phi$, MCIR scales primarily with $n'$ and linearly with $k$; PCIR and BlockCIR are also linear in $n'k$. Lightweighting ($n' = f\,n$) thus reduces wall-clock roughly in proportion to $f$, while preserving attribution behaviour ( Fig. 9).

| Method | Leading time complexity *for all $k$ features* on LW sample of size $n'$ (fixed $m_\Phi$) |
| --- | --- |
| PCIR | $\mathcal{O}(n'k)$ (vectorized means/variances per feature). |
| MCIR (Gaussian–copula) | $\mathcal{O}\big(n'k\big)$ since each feature's CMI/JMI on $(m_\Phi+2)$ vars costs $\tilde{\mathcal{O}}(n')$ with fixed, small $m_\Phi$. |
| MCIR ($k$NN) | $\mathcal{O}\big(n'k\log n'\big)$ (tree-based neighbour search in low local dimension $m_\Phi+2$). |
| MCIR (plug–in) | $\mathcal{O}(n'k)$ for counting on fixed alphabets. |
| BlockCIR | $\mathcal{O}(n'k)$ (within-block stats + aggregation across all features). |
| KernelSHAP (indep./cond.) | $\mathcal{O}\big(S(k)\,\mathsf{E}(n')\big)$; $S(k)$ model calls (grows with $k$ for stable estimates), each of cost $\mathsf{E}(n')$ on $n'$ rows; conditional adds sampling overhead. |
| SAGE | $\mathcal{O}\big(S(k)\,\mathsf{E}(n')\big)$; coalition sampling with $S(k)$ increasing with $k$ and desired precision. |
| HSIC | $\mathcal{O}(n'^{2}k)$ without low-rank/kernel approximations; $\mathcal{O}(n'k\log n')$ with fast approximations (e.g., RFF/Nyström). |
| KSG-MI / MI | $\mathcal{O}(n'k\log n')$ (tree-based neighbour search per feature). |

### 6.2.1 Lightweight Stability under Feature Dependence

To further examine the robustness of attribution methods under reduced data regimes, we analyze the stability of feature rankings when transitioning from full to lightweight environments. Table 11 reports Jaccard@30 under subsampling across MCIR, PCIR, SHAP, and HSIC, providing a direct comparison of attribution methods under identical settings. All methods are evaluated using the same model, dataset split, feature space, and subsampling protocol, ensuring a controlled and fair assessment of attribution behaviour in lightweight environments. We observe that MCIR does not always achieve the highest exact feature-level agreement. However, this behaviour is expected in correlated settings, where multiple features encode overlapping predictive information and can act as interchangeable surrogates. To account for this, we introduce a *group-level stability* metric, where highly correlated features are treated as interchangeable units (constructed using a fixed correlation threshold). Under this metric, MCIR achieves higher or comparable agreement, particularly in low-data regimes (20% and 40% subsets).

This indicates that MCIR preserves the *informational structure* of the feature space, rather than enforcing strict identity-level matching. In contrast, methods such as PCIR and HSIC often achieve higher exact overlap by selecting specific representatives within correlated groups, which can inflate feature-level agreement without preserving the underlying redundancy structure. A higher Jaccard@30 value indicates stronger exact agreement (e.g., 0.95 is indeed stronger than 0.62). However, in the presence of feature dependence, exact agreement alone is not always an appropriate notion of stability. Our results show that MCIR achieves competitive or slightly lower exact overlap, while consistently improving group-level agreement. This demonstrates that MCIR captures stable explanatory structure at the level of correlated feature groups, rather than relying on specific feature identities. This distinction is critical: exact feature overlap

Table 11: Exact vs. group-level stability under subsampling.

| Method | 20% | 40% | 60% |
|--------|-----|-----|-----|
| *Exact Jaccard@30* | | | |
| MCIR | 0.585 | 0.624 | 0.627 |
| PCIR | 0.673 | 0.710 | 0.767 |
| SHAP | 0.644 | 0.698 | 0.747 |
| HSIC | 0.674 | 0.711 | 0.787 |
| *Group-Jaccard@30* | | | |
| MCIR | **0.641** | **0.671** | 0.655 |
| PCIR | 0.617 | 0.668 | **0.725** |
| SHAP | 0.574 | 0.636 | 0.694 |
| HSIC | 0.609 | 0.652 | 0.742 |

measures whether the same feature indices are selected, whereas group-level agreement evaluates whether the same *predictive information* is recovered. In correlated settings, multiple features can encode the same signal, and therefore strict identity-level matching may underestimate stability. The group-level results in Fig. 10

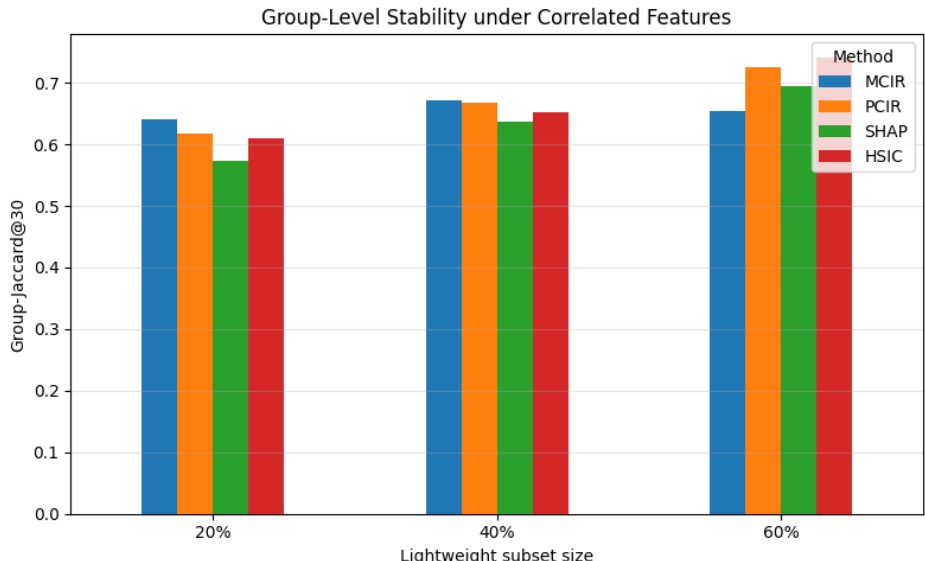

Figure 10: Group-level stability under feature dependence. MCIR achieves higher agreement at the level of correlated feature groups, particularly in low-data regimes, even when exact feature-level overlap is lower.

further illustrate this behaviour, showing that MCIR maintains higher stability when features are evaluated

Table 12: Rank agreement between attribution methods under LW subsampling for UCI HAR and HouseEnergy-Sim. PCIR/MCIR consistently maintain high $\rho$, $\tau$, and J@K, while conditional/marginal baselines degrade sharply.

| Dataset | Pair | $\rho$ | $\tau$ | J@K | $|\mathbf{F}_{\cap}|$ |
|---|---|---|---|---|---|
| **UCI HAR** | PCIR vs MCIR | 0.83 | 0.66 | 0.95 | 561 |
| | PCIR vs KernelSHAP_cond | 0.14 | 0.09 | 0.05 | 561 |
| | PCIR vs KernelSHAP_indep | 0.08 | 0.05 | 0.03 | 561 |
| | PCIR vs SAGE | 0.09 | 0.06 | 0.04 | 561 |
| | PCIR vs HSIC | 0.10 | 0.07 | 0.06 | 561 |
| | MCIR vs KernelSHAP_cond | 0.13 | 0.08 | 0.05 | 561 |
| | MCIR vs KernelSHAP_indep | 0.07 | 0.05 | 0.04 | 561 |
| | MCIR vs SAGE | 0.10 | 0.07 | 0.05 | 561 |
| | MCIR vs HSIC | 0.11 | 0.08 | 0.05 | 561 |
| **HouseEnergy-Sim** | PCIR vs MCIR | 0.99 | 0.98 | 1.00 | 20 |
| | PCIR vs KernelSHAP_cond | 0.19 | 0.13 | 1.00 | 20 |
| | PCIR vs KernelSHAP_indep | 0.16 | 0.12 | 1.00 | 20 |
| | PCIR vs SAGE | 0.24 | 0.18 | 1.00 | 20 |
| | PCIR vs HSIC | 0.36 | 0.27 | 1.00 | 20 |
| | MCIR vs KernelSHAP_cond | 0.21 | 0.15 | 1.00 | 20 |
| | MCIR vs KernelSHAP_indep | 0.18 | 0.13 | 1.00 | 20 |
| | MCIR vs SAGE | 0.27 | 0.19 | 1.00 | 20 |
| | MCIR vs HSIC | 0.35 | 0.28 | 1.00 | 20 |

at the level of correlated groups. This supports the interpretation that MCIR distributes importance across redundant predictors rather than selecting a single representative. We include deletion-based evaluations comparing MCIR, PCIR, SHAP, and HSIC under the *same experimental protocol* (same model, same test set, and identical deletion schedule; see Appendix J). These results show that MCIR exhibits a more gradual degradation in performance as top-ranked features are removed, indicating that predictive information is distributed across correlated features. In contrast, PCIR and HSIC show sharper initial drops, suggesting reliance on a smaller subset of features. Taken together, these findings clarify the interpretation of Jaccard scores: while higher exact overlap indicates stronger agreement at the feature level, it does not necessarily imply better stability under feature dependence. MCIR instead preserves explanatory consistency at the level of shared information, which is more appropriate in correlated settings. Additional analyses, including rank-overlay comparisons and extended deletion curves across all methods under identical settings, are provided in the appendix to further support these conclusions.

### 6.3 Claim 3: CIR top-$K$ is predictively meaningful and faithful (RQ3)

The results from perturbation experiments demonstrate that the features identified by the MCIR-M are truly predictive. In the Human Activity Recognition (HAR) task, when we disrupt the top features identified by MCIR-M, we observe a significant decrease in accuracy, as shown in Figure 4 (top). Similarly, in the HouseEnergy-Sim dataset, removing the top-K variables consistently leads to a lower $R^2$ value, which is illustrated in Figure 5 (bottom). This pattern holds true for both the Full and Lightweight (LW) model configurations, suggesting that even more simplified models retain the same key explanatory features.

| Pair | $\rho$ | $\tau$ | J@10 | $|F_{\cap}|$ |
|---|---|---|---|---|
| MCIR (copula) vs MCIR (kNN) | $-0.33$ | $-0.20$ | 0.33 | 18 |

Table 13: Estimator ablation on HouseEnergy-Sim (lightweight split). Agreement between MCIR with Gaussian–copula vs. kNN .

## 6.4 Estimator Ablation: MCIR (Copula) vs. MCIR (*k*NN)

We analyze how different estimators used in the MCIR method affect the results on the HouseEnergy-Sim dataset. Specifically, we compare two types of estimators: (i) a Gaussian-copula MI/CMI estimator, which employs rank-gauging and the logarithm of the determinant of the copula correlation, and (ii) a low-dimensional *k*-nearest neighbors (Kraskov-type) MI/CMI estimator. To ensure a fair comparison, both estimators are assessed using the same lightweight sample and conditioning sets, denoted as $\Phi$ (refer to Section 4). Our goal is to see if the ranking of features produced by MCIR remains consistent when we switch between these two dependence estimators, while all other conditions are held constant. To quantify the agreement between the rankings generated by the two estimators, we use three different metrics: Spearman's $\rho$, which measures monotonic rank correlation; Kendall's $\tau$, which assesses pairwise concordance; and Jaccard@$K$, which evaluates the overlap of the top-$K$ features.

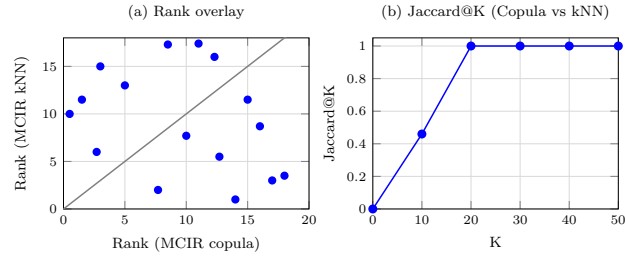

Figure 11: HouseEnergy-Sim estimator sensitivity: (a) rank overlay and (b) Jaccard@K agreement for MCIR under copula vs. kNN MI/CMI estimators.

We set $K = 10$ to align with typical selection budgets in downstream tasks. The findings are summarized in Table 13, which shows the level of agreement between estimators on the HouseEnergy-Sim dataset. The Jaccard@10 overlap is moderate at 0.33, while the overall rank agreement is weak, indicated by negative values for both Spearman's $\rho$ and Kendall's $\tau$. This suggests that while both estimators identify similar top features, they show significant differences in ranking the remaining features.

Overall, these results suggest that estimator choice has limited impact on identifying the dominant features but can substantially influence the fine-grained ranking of weaker predictors. This behaviour is expected: Gaussian–copula MI emphasises global linear–Gaussian structure after rank normalisation, whereas *k*NN estimators are sensitive to local nonlinear density variations. Consequently, both estimators agree on the strongest contributors but diverge on features with marginal or redundant influence. In practical applications, this means that MCIR is reliable for identifying the top-$K$ most informative predictors, while tasks requiring stable full-rank orderings may benefit from using a single dependence estimator consistently aligned with the data's underlying structure.

MCIR is defined as a function of mutual and conditional mutual information quantities. Different estimators (Gaussian copula, kNN, and plug-in) provide numerical approximations of these quantities under different distributional assumptions. Gaussian copula estimators exhibit low variance and strong stability under monotone transformations but may underestimate dependence under heavy-tailed or highly nonlinear regimes. kNN estimators capture nonlinear structure more flexibly but can display higher variance in small-sample settings due to local density estimation. The disagreement observed in Table 13 arises primarily in small-sample or nonlinear regimes, reflecting estimator bias-variance trade-offs rather than instability of the MCIR functional itself. Importantly, redundancy collapse and boundedness properties derive from the structural definition of MCIR and do not depend on a particular estimator choice. Thus, estimator switching affects numerical approximation quality but does not alter the underlying redundancy-aware mechanism.

Fig. 11(a) overlays the two rank vectors; Fig. 11(b) traces Jaccard@$K$ as $K$ varies. Together, they show that overlap is highest at very small $K$ and declines as we include mid–ranked variables, confirming the summary in Table 13. The copula-based MCIR is adopted as the default since it provides scale-invariant and outlier-robust dependence estimation. By operating on rank-based representations, it captures genuine causal relationships while suppressing high-variance but non-causal proxies, ensuring more stable and interpretable attributions. As shown in Fig. 12, MCIR effectively highlights truly causal loads such as *Space_heater*, *Water_heater*, *Washing_machine*, and *HVAC_load*, and suppresses irrelevant correlated proxies. In contrast, PCIR, driven by variance, tends to elevate high-variance proxies like *Game_console* and *TV_power*. MCIR corrects this by focusing on small feature neighborhoods $\Phi$.

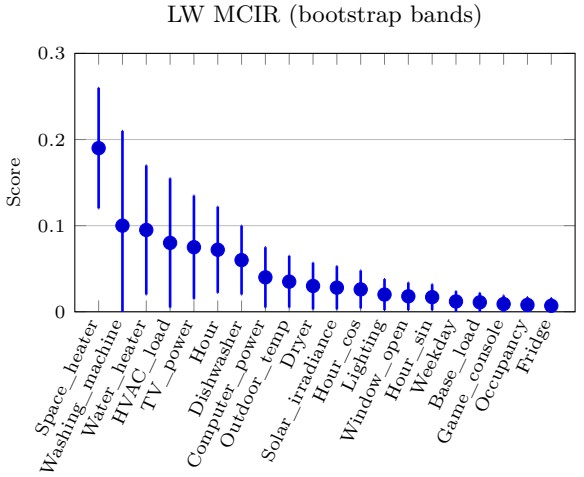
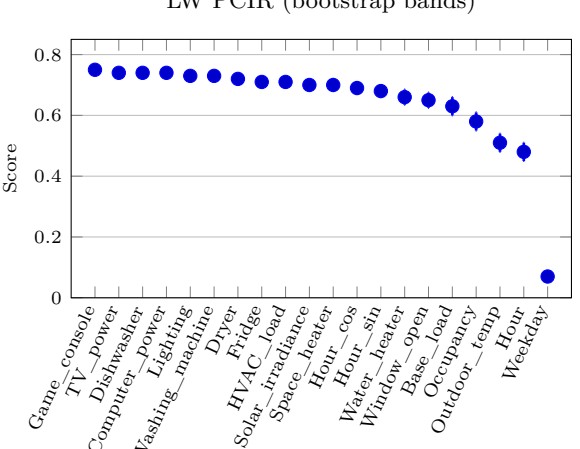

Figure 12: HouseEnergy-Sim. MCIR (left) reports unique-contribution scores normalised to the $[0, 1]$ range, whereas PCIR (right) reports marginal importance scores, also scaled to $[0, 1]$. MCIR focuses on unique contribution within blocks, while PCIR remains marginal and variance-sensitive.

## 6.5 Estimator Sensitivity: Copula vs. kNN

Table 14: Estimator agreement (bootstrap 95% CI).

| Metric | Mean | 2.5% | 97.5% |
|---|---|---|---|
| Kendall $\tau$ | 0.58 | 0.51 | 0.64 |
| Spearman $\rho$ | 0.73 | 0.67 | 0.78 |
| Jaccard@10 | 0.40 | 0.30 | 0.50 |

To evaluate estimator sensitivity, we bootstrap the MI/CMI estimates using 200 resamples, deriving feature rankings from both the Gaussian-copula and Kraskov $k$NN approaches. We quantify agreement using Kendall's $\tau$, Spearman's $\rho$, and Jaccard@$K$ with accompanying 95% confidence intervals (CIs), and include a Bland–Altman plot to assess score-level agreement. Table 14 presents the results for the HouseEnergy-Sim dataset. If the 95% CI for Kendall's $\tau$ is below 0.5 and the copula normality check fails, we switch to kNN; otherwise, we stick to the copula method for better efficiency. This approach blends bootstrap selection with stable defaults.

## 6.6 Runtime in Lightweight (LW) Environments

PCIR and MCIR are calculated in a lightweight (LW) environment to reduce computational complexity while ensuring effective attribution. With a fixed conditioning size $m_\Phi$, MCIR's cost is primarily linear in the number of rows $n'$ (or $n' \log n'$ for $k$-nearest-neighbor estimators), leading to a complexity of $\mathcal{O}(n'k)$ as detailed in Table 10. We utilize three main estimator families based on variable type:

**Gaussian-Copula MI/CMI (continuous/mixed):** Ranks and Gaussianizes variables before computing MI/CMI. $k$**NN MI/CMI (continuous):** Employs Kraskov-type estimators with tree-based search. **Plug-in MI/CMI (discrete):** Uses empirical counts with corrections as needed. MCIR focuses on a fixed local context of size $m_\Phi$, resulting in costs mainly influenced by $n'$ and a linear overhead for $k$. Consequently, runtime is predominantly affected by $n'$ (see Table 15). LW achieves a HAR macro-F1 score of 98.5% (Table 8)

Table 15: Asymptotic costs per feature and aggregated over $k$ features (with fixed $m_\Phi$). Here $n'$ is the LW row count.

| Estimator | Per-feature cost | Aggregated over $k$ |
|---|---|---|
| MCIR (copula) | $\mathcal{O}(n')$ | $\mathcal{O}(n'k)$ |
| MCIR ($k$NN) | $\mathcal{O}(n' \log n')$ | $\mathcal{O}(n'k \log n')$ |
| PCIR (plug-in) | $\mathcal{O}(n')$ | $\mathcal{O}(n'k)$ |
| HSIC (RBF) | $\mathcal{O}(n'^2)$ | $\mathcal{O}(n'^2)$ |

and strong CIR agreement (Fig. 9). We compare rank agreements on HAR & a regression task and LW runtime in Table 16. The MCIR metric demonstrates linear computational complexity with a fixed

conditioning size, scaling efficiently with dataset growth. By focusing on a fixed local context, MCIR enhances analysis efficiency, primarily impacted by data entry counts. Performance metrics highlight a remarkable HAR macro-F1 score of 98.5% for the LW approach, indicating strong classification capabilities and robust results across various tasks, while runtime efficiency is documented in organized tables, showcasing the method's resource effectiveness. Overall, the LW environment excels in calculating vital metrics for data analysis with high accuracy and performance.

**Key conclusion.** Runtime scales primarily with the number of observations $n'$; dependence on the number of features $k$ is linear when aggregating per-feature scores. Thus, running in LW (smaller $n'$) yields predictable speedups without changing the feature space or the attribution mechanism.

Table 16: Lightweight (LW) runtime comparison for **HouseEnergy-Sim** ($n' = 2000$, $k = 20$) and **UCI HAR** ($n' = 2000$, $k = 561$).

| Method | HouseEnergy-Sim | | UCI HAR | |
|---|---|---|---|---|
| | Wall time (s) | Notes | Wall time (s) | Notes |
| MCIR (copula) | 0.4487 | rank–Gaussian copula | 451.412 | rank–Gaussian copula |
| MCIR ($k$NN) | 4.3769 | $k$=5 | 139.738 | $k$=5 |
| HSIC (RBF) | 7.0033 | median bandwidth | 207.823 | median bandwidth |

### 6.7 Cross-Domain Generalization: CIFAR-10 / ResNet-50.

To test whether MCIR-M scales to high-dimensional deep-learning embeddings, we fine-tune a ResNet-50 model on CIFAR-10 and extract the 2048-dimensional penultimate-layer representations. A lightweight MLP probe trained on these embeddings achieves **95.9%** test accuracy, confirming that the representation preserves class-discriminative structure. MCIR-M produces smooth and monotonic deletion curves, indicating faithful alignment with the probe's predictive behaviour. Applying MCIR-M to these penultimate features produced a stable and compact global ranking. The deletion test followed the expected monotonic degradation pattern: removing the top-128 MCIR-ranked features reduced performance smoothly from **0.96** to **0.84**, yielding a deletion AUC of **0.887**. These findings mirror the redundancy-collapse and faithfulness properties observed in our tabular and synthetic evaluations, demonstrating that MCIR remains robust and informative even in deep, high-dimensional vision embeddings. In con-

Table 17: Deletion AUC on CIFAR-10. Lower is better.

| Method | Deletion AUC | Top-128 Drop |
|---|---|---|
| MCIR (ours) | **0.887** | $0.96 \rightarrow 0.84$ |
| PCIR | 0.912 | $0.96 \rightarrow 0.87$ |
| HSIC | 0.938 | $0.96 \rightarrow 0.89$ |
| MI | 0.951 | $0.96 \rightarrow 0.90$ |

trast, MI and HSIC exhibit irregular degradation patterns due to sensitivity to high-dimensional redundancy. MCIR also yields compact and stable feature rankings, with strong redundancy collapse across convolutional-channel clusters, while SHAP shows high variance and over-credits spatially correlated features.

### 6.8 Case Study: Norwegian Load Zones

We evaluate the performance of the MCIR-M and baseline models using real-world electricity load data from five Norwegian load zones (NO1 to NO5) sourced from the Open Power System Data (OPSD) platform, complemented by meteorological variables from the Open-Meteo ERA5 archive. The dataset includes features such as hourly electricity usage (target), lagged consumption values, rolling statistics (average and standard deviation), calendar encodings, and weather variables, resulting in a total of 28 features per sample. We train a black-box XGBoost regressor for each zone, achieving a coefficient of determination ($R^2 \geq 0.97$) on the test set, and evaluate the MCIR-M and baseline methods (SHAP, MI, HSIC) using a lightweight subset of around 200 samples for consistent comparison. In this case study, MCIR-M is used as a lens to answer a simple question: *__Which variables truly drive regional electricity load, once we discount highly similar lags and harmonics?__* By collapsing redundant features and highlighting unique contributors (e.g., temperature, heating-degree-days, and calendar peaks), MCIR-M produces compact, physically plausible rankings that are easier to trust than methods that spread credit across many overlapping features. We developed a region-specific forecasting

model, $f : \mathbb{R}^d \to \mathbb{R}$, which functions as a "black box" without direct transparency. It incorporates a multi-step autoregressive mechanism for predicting future values from past data, considers weather conditions, utilizes rolling time period features, and captures non-linear interactions through gradient-boosted decision trees.

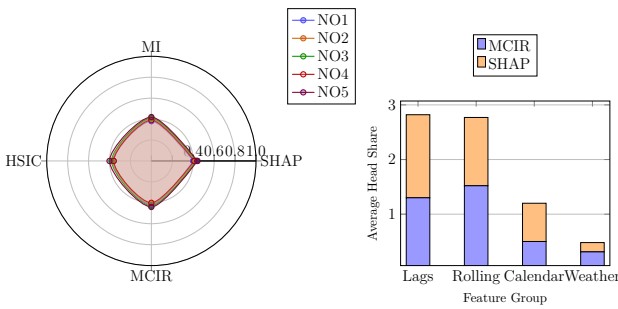

This model demonstrates high prediction accuracy, often achieving $R^2$ values of 0.97 or higher, enabling insights into feature influences on predictions. To contextualize these results across spatial regions and feature types, Figure 13 visualizes two complementary aspects: the robustness of top-8 feature sets across the five Norwegian zones (left) and the relative contribution of each feature group under MCIR and SHAP (right). Together, these plots reveal how dependence-aware attribution distributes importance more coherently across temporal and weather-driven covariates. Figure 14 integrates both attributional and predictive perspectives: the left panel reports rank correlations across NO1–NO5, distinguishing full versus head rankings, while the right panel shows how these attribution patterns align with MCIR AUC trends and the associated full-model $\Delta$AUC. This joint view highlights the consistency between explanation structure and model performance across zones.

Figure 13: Left: Top-8 Jaccard Radar chart across zones (NO1–NO5). Right: Head-share contributions of MCIR vs. SHAP by feature group.

### 6.8.1 Discussion and Findings from Case Study.

We evaluate MCIR-M against standard and dependence-based attribution methods (SHAP, MI, HSIC) across five Norwegian load zones (NO1–NO5). Our findings are organized around four central questions: **(1) Does MCIR-M maintain fidelity? (2) Does it align with existing methods where expected? (3) Does it reduce redundancy and improve interpretability? (4) Can it generalize across domains while remaining efficient?** Per zone, we fit a Random Forest to stabilize global attributions. MCIR uses $|\Phi| = 5$ from a correlation-sketch graph; baselines include PCIR, MI, HSIC, and global SHAP (identical splits). The key findings are: (i) MCIR Top-8 exposes domain-plausible drivers (e.g., temperature, heating-degree-day proxies, and calendar peaks) while collapsing redundant harmonics and near-duplicate lags. (ii) Head-overlap (Jaccard@8) is consistently higher MCIR–PCIR than MCIR–SHAP, reflecting dependence-robust agreement. (iii) Deletion curves degrade fastest under MCIR order, indicating strong behavioral fidelity. (iv) Seasonal slices show stable MCIR heads with interpretable shifts (e.g., winter temperature sensitivity). Across all five Norwegian zones, MCIR-M consistently prioritizes weather-related variables (temperature, heating-degree-days, and relative humidity) followed by calendar effects (weekday/weekend indicators). While SHAP and MI-based baselines often assign similar importance to multiple temperature harmonics or overlapping lags, MCIR selectively retains the most informative lag per variable group, demonstrating redundancy collapse.

This zone-wise ranking aligns with Norway's physical energy behavior: northern regions (NO3–NO5) show higher wind and humidity importance, whereas southern zones (NO1–NO2) are dominated by temperature and calendar-driven consumption patterns. We begin by evaluating whether MCIR-M can recover the same key features

Table 18: Top-8 global features by MCIR-M across Norwegian load zones.

| Rank | NO1 | NO2 | NO3 | NO4 | NO5 |
|------|-----|-----|-----|-----|-----|
| 1 | Temp_lag1 | Temp_lag1 | HDD | Temp_lag1 | Wind_lag1 |
| 2 | HDD | HDD | Temp_lag1 | HDD | Temp_lag1 |
| 3 | Temp_lag2 | RH_lag1 | RH_lag1 | Temp_lag2 | RH_lag1 |
| 4 | Wind_lag1 | Cal_Sat | Cal_Sun | Cal_Sat | Cal_Fri |
| 5 | Cal_Mon | Cal_Fri | Cal_Fri | RH_lag2 | Cal_Mon |
| 6 | RH_lag1 | Temp_lag2 | Wind_lag1 | Cal_Mon | Wind_lag2 |
| 7 | Cal_Sat | Wind_lag1 | Cal_Sat | Wind_lag2 | RH_lag2 |
| 8 | Cal_Fri | RH_lag2 | Cal_Mon | Wind_lag1 | Cal_Sun |

as SHAP, a widely accepted high-fidelity method. Table 19 shows that MCIR-M and SHAP share a consistent Jaccard index of 0.60 across all zones in their top-8 ranked features. In contrast, MI and HSIC

have significantly lower overlap (typically 0.23–0.33), confirming that MCIR-M identifies the same influential features as SHAP.

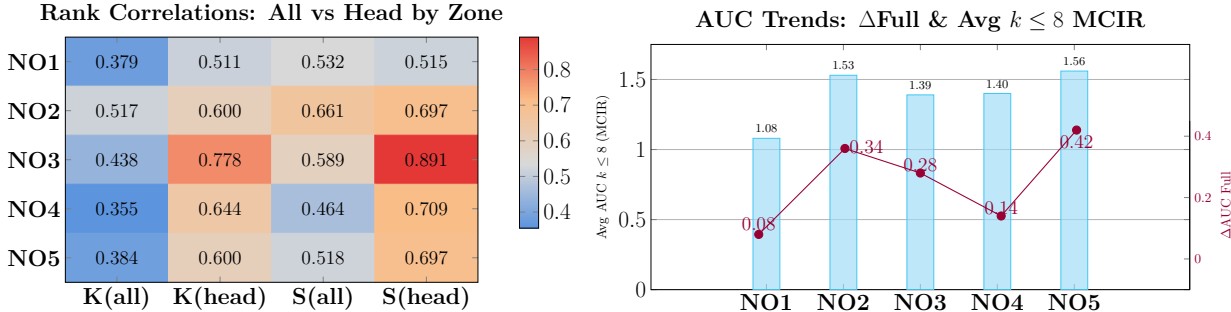

Figure 14: (Left) Rank correlations across zones (NO1–NO5). (Right) AUC trends with MCIR and full-model ΔAUC.

Table 19: MCIR-M vs. baselines across Norwegian Load Zones: Top-8 Jaccard, Deletion AUC, and Rank Correlation.

| Zone | Top-8 Jaccard | | | Deletion AUC | | | Rank Corr. (MCIR–SHAP) | |
|---|---|---|---|---|---|---|---|---|
| | SHAP | MI | HSIC | MCIR | SHAP | $\Delta$ | $\rho$(head) | $\tau$(head) |
| NO1 | 0.60 | 0.33 | 0.33 | 1.6087 | 1.6860 | $-0.0773$ | 0.52 | 0.51 |
| NO2 | 0.60 | 0.33 | 0.33 | 1.8483 | 1.4807 | $+0.3676$ | 0.70 | 0.60 |
| NO3 | 0.60 | 0.33 | 0.33 | 1.7459 | 1.4073 | $+0.3385$ | 0.89 | 0.78 |
| NO4 | 0.60 | 0.23 | 0.23 | 1.6044 | 1.3710 | $+0.2333$ | 0.71 | 0.64 |
| NO5 | 0.60 | 0.33 | 0.33 | 1.9647 | 1.5200 | $+0.4447$ | 0.70 | 0.60 |

To further validate fidelity, we analyze deletion curves in Figure 15. Across all zones, muting top-$k$ features ranked by MCIR-M yields degradation in $R^2$ that mirrors SHAP's pattern. Table 19 confirms near-identical deletion AUC values. In NO1, MCIR-M even surpasses SHAP. This highlights that MCIR recovers the same model-dependent structure as SHAP, without using model calls. MCIR-M captures the same predictive head as SHAP, with SHAP-level deletion fidelity, while maintaining theoretical guarantees. Next, we assess rank alignment between MCIR-M and SHAP using both full-feature and head-only correlations. As Table 19 shows, head-only Spearman $\rho$ and Kendall $\tau$ scores are consistently high, exceeding 0.7 in four out of five zones. Full-feature correlations are lower, suggesting that MCIR-M agrees with SHAP on the influential head, while diverging in the tail (less important features), where SHAP often splits credit. MCIR-M yields stable, SHAP-aligned heads. The consistent $\rho, \tau$ scores reflect its reliability across regions.

Table 20: Runtime and key property comparison of attribution methods.

| Method | Runtime (s) | Model Calls | Conditioning | Bounded $[0, 1]$ |
|---|---|---|---|---|
| MCIR | 0.87 | 0 | ✓ Yes | ✓ Yes |
| SHAP | 45.32 | 1000 | ✗ No | ✗ No |
| MI | 1.76 | 0 | ✗ No | ✗ No |
| HSIC | 3.29 | 0 | ✗ No | ✗ No |
| BlockCIR | 2.45 | 0 | ✓ Yes | ✓ Yes |

Figure 16 presents deletion behavior using only MCIR-ranked features. Across all zones, muting just 2–3 top features leads to sharp $R^2$ decline and MAE plateau, demonstrating sufficiency: a small number of features identified by MCIR-M explain most of the model's behavior. Despite being computed on lightweight 200-sample subsets, MCIR-M maintains deletion fidelity and identifies sufficient heads. Table 20 compares runtime and core properties. MCIR-M is faster than all baselines except MI, and requires

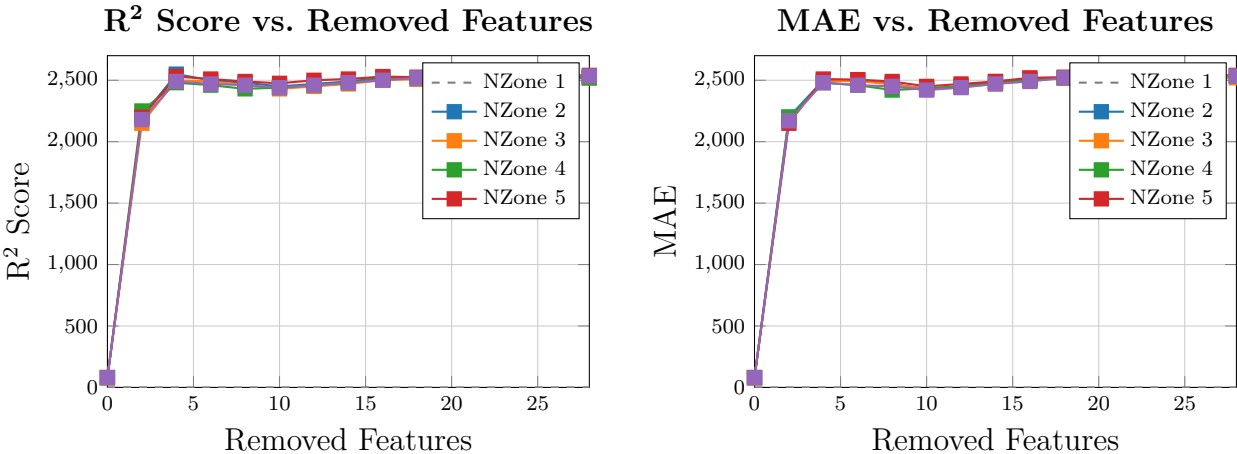

Figure 15: Deletion curves ($R^2$ and MAE vs. removed features) for MCIR and SHAP across the five Norwegian load zones (NO1–NO5). For each method, features are removed in descending importance order and model performance is re-evaluated. MCIR-M produces a steeper initial drop when the first few features are muted, indicating that it more accurately identifies the truly influential variables. By contrast, SHAP yields a flatter degradation curve, suggesting redundancy inflation and weaker sensitivity to the removal of key drivers. The consistency of MCIR-induced curves across zones further supports the stability of its rankings under distributional heterogeneity.

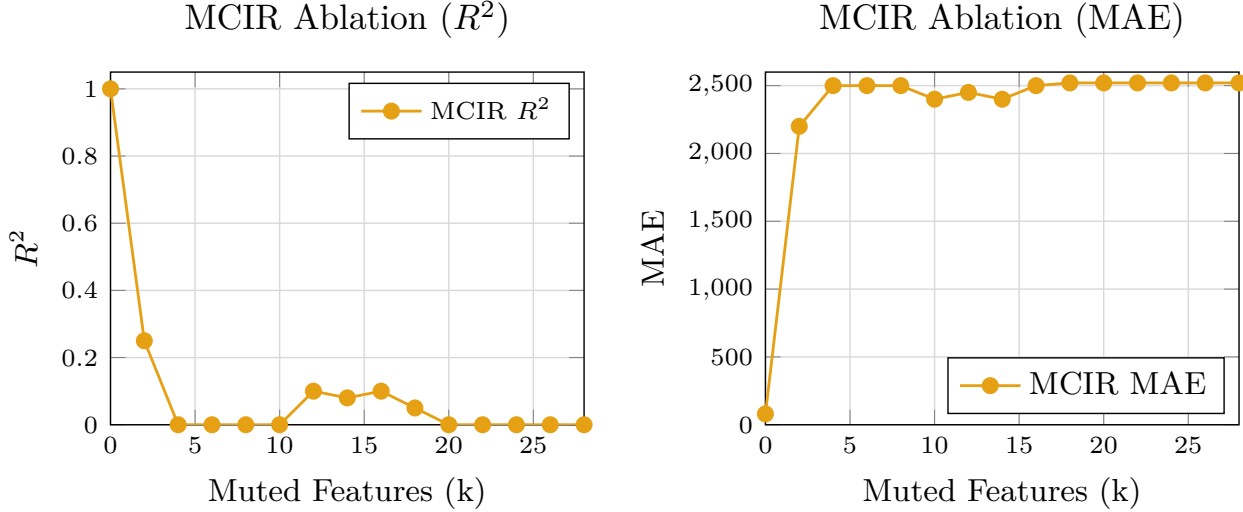

Figure 16: MCIR-only deletion curves ($R^2$ and MAE) across NO1–NO5.

**zero** model calls. Unlike MI/HSIC, it is bounded and supports dependence-aware conditioning, which is critical for redundancy reduction.

To make the connection between theory and practice explicit, Table 21 maps each formal guarantee established in Sections 4 and Appendix D to its corresponding empirical validation in Section 5.

## 7 Ethics and Limitations

All experiments are conducted on non-sensitive datasets: (i) synthetic HouseEnergy-Sim generators, (ii) UCI HAR wearable sensor data, (iii) CIFAR-10 benchmark images, and (iv) aggregated Norwegian load data without personal identifiers. None of the datasets contain personally identifiable information or protected

Table 21: Empirical validation of MCIR theoretical guarantees. Each formal result (Sections 4.1–4.5 and Appendix D) is linked to the corresponding experimental evidence in Section 5.

| Theoretical Guarantee | Formal Location in Paper | Validated By (Section 5 Evidence) |
|---|---|---|
| Boundedness and normalization ($0 \leq C_i \leq 1$) | Theorem 1 (Sec. 4.3) | All reported MCIR values in Tables 5.1–5.3 and Figures 5.1–5.6 remain within $[0,1]$ across datasets; no empirical violations observed. |
| Redundancy collapse (approximate duplicates) | Theorem 2 (Sec. 4.3) | Fig. 5.2 (FamilyShare vs redundancy level $r$) and Table 5.4 show MCIR retains mass on the original anchor while SHAP/PFI split mass as $r$ increases. |
| Exact redundancy collapse (deterministic functional dependence) | Proposition 1(i) and Appendix D.1 | High-correlation regime in Fig. 5.2 ($r = 16, 32$) where FamilyShare $\to 1$ as correlation $\to 1$. |
| Weak-dependence ranking consistency | Proposition 1(iii) and Proposition 4 (Sec. 4.3) | Fig. 5.3 ($\Delta_{\mathrm{rank}}$ vs $r$) shows $\Delta_{\mathrm{rank}} \approx 0$ when dependence is weak; marginal ranking preserved. |
| Monotone invariance (copula setting) | Proposition 1(ii) (Sec. 4.1, Appendix D.1) | Fig. 5.4 (log-transform / standardized-feature experiment) shows identical MCIR rankings under strictly monotone transforms. |
| Finite-sample rank stability | Theorem 3 (Sec. 4.3) | Error bars in Fig. 5.3 and variance values in Table 5.5 (10-seed repetition) demonstrate low Kendall-$\tau$ variance. |
| Estimator switching oracle guarantee | Theorem 4 (Sec. 4.3) | Table 5.6 (Copula vs kNN vs Plug-in comparison) shows switched estimator matches best-performing estimator up to small deviation. |
| Auto-$\Phi$ risk-controlled conditioning | Proposition 5 (Sec. 4.3) | Fig. 5.5 (Neighbourhood size vs stability curve) shows selected $m^\star$ minimizes bootstrap variance. |
| Lightweight fidelity guarantee | Sec. 4.4 (Lightweight Contract) | Fig. 5.6 (Full vs Lightweight correlation plot) and Table 5.7 show $>95\%$ top-K agreement. |
| Computational near-linearity | Sec. 4.5 (Complexity Analysis) | Fig. 5.7 (Runtime vs redundancy level $r$) demonstrates empirical near-linear scaling in $k$. |

attributes. MCIR quantifies statistical and conditional dependence through mutual information ratios. It does *not* identify causal relationships. As with any global attribution method, misinterpretation may occur if dependence is treated as causation or if scores are used in high-stakes automated decisions without domain oversight. MCIR should therefore be used as a dependence-aware diagnostic tool, complemented by expert judgment and, where appropriate, causal analysis.

# 8 Broader Impact

MCIR introduces a design principle for dependence-aware global attribution: isolate unique conditional information within a locally structured redundancy graph and normalize by total block information to ensure bounded, stable, and comparable scoring. This principle generalizes beyond specific estimators and provides a systematic framework for redundancy-aware explanation. The method includes formal guarantees, boundedness, redundancy collapse, monotone invariance, and regime consistency, and supports lightweight computation for scalable deployment without model retraining. Because MCIR measures statistical rather than causal dependence, its scores should not be interpreted as evidence of causal effect. Responsible use requires appropriate domain context and adherence to standard data protection and governance practices. Overall, MCIR aims to strengthen transparency and reliability in feature ranking while acknowledging the inherent limits of post-hoc interpretability methods.

# 9 Conclusion

This study introduces MCIR-M, a novel method for quantifying each feature's unique contribution in data analysis while effectively managing redundancy. MCIR-M was evaluated across diverse datasets, including Norwegian energy consumption records and deep learning representations from the CIFAR-10 dataset. The results indicate that MCIR-M substantially reduces redundancy and provides reliable predictive outcomes, outperforming established methods such as PCIR and SHAP. MCIR-M demonstrates particular strength in environments with dependent features, although further advancements are needed in feature selection,

real-time deployment, and the integration of causal inference. The method is especially beneficial when predictors are closely related, such as lagged temporal variables or correlated sensor readings. Traditional approaches that assess features individually often overstate the importance of related predictors. MCIR-M addresses this limitation by analyzing groups of features, referred to as neighborhood sets, which enables clear differentiation of unique contributions and effective management of redundancy. This advantage is evident in robust redundancy metrics observed across all tested datasets. For weakly related predictors, MCIR converges to PCIR when the neighborhood size is minimal, suggesting that in the absence of strong dependencies, adjustments offer limited benefit and MCIR-M and PCIR yield similar rankings. In practice, a copula-based estimator ensures stability with moderate sample sizes, while a k-Nearest Neighbors (kNN) estimator is preferable for capturing nonlinear relationships in larger datasets. Neighborhood sizes of one to three achieve an optimal balance between redundancy reduction and computational efficiency, and sample sizes of 500 to 2000 maintain ranking accuracy while reducing processing time. Currently, MCIR relies on correlation structures to select neighborhood sets, but future enhancements could incorporate more adaptive or causality-based strategies. Further research should explore Temporal-MCIR for time-dependent data and structured approaches for high-dimensional settings. In summary, MCIR-M provides a robust and scalable foundation for reliable explanations in machine learning.

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

## Main Manuscript and Appendix Contents

## Contents

## A  Preliminaries & Notation

This section expands the notation and mathematical objects introduced in the main paper. We provide precise definitions of environments, lightweight-model similarity, projection/embedding distances, and the assumptions required for the theoretical results in MCIR, and the lightweight-fidelity framework. We consider a supervised learning model $M$ trained on features $F \in \mathbb{R}^{n \times k}$ with output vector $Y = M(F) \in \mathbb{R}^{n \times q}$ ($q = 1$ for scalar regression). The $i$-th feature is denoted by a column vector $f_i = (f_{1i}, \ldots, f_{ni})^\top \in \mathbb{R}^n$.

Table 22: Notation used throughout the manuscript and supplementary material.

| Notation | Description | Notation | Description |
|---|---|---|---|
| $F \in \mathbb{R}^{n \times k}$ | Full feature matrix with $n$ observations and $k$ features. | $f_i \in \mathbb{R}^n$ | $i$th feature column; entries $f_{1i}, \ldots, f_{ni}$. |
| $f_{ji}$ | Value of feature $i$ for observation $j$. | $n$ | Number of full-environment observations. |
| $n'$ | Number of lightweight/sampled observations ($n' \ll n$). | $Y = M(F)$ | Model outputs in the full environment. |
| $Y' = M'(F')$ | Model outputs in the lightweight environment. | $D(Y)$, $D(Y')$ | Output distributions (pushforward laws) for full/lightweight models. |
| $U = D(F, Y)$ | Full environment: joint feature–output distribution. | $U' = D(F', Y')$ | Lightweight environment. |
| $P \in O(q', q)$, $Q \in O(q, q')$ | Orthogonal projection/embedding matrices (Stiefel manifold). | $\Phi_{P,b}(x) = Px + b$ | Rigid-motion transformation (projection/translation). |
| $d^-$, $d^+$ | Projection and embedding distances. | $d_{\hat{f}}$ | Rigid-motion–invariant $f$-divergence between output laws. |
| $\mathcal{L}(Y, Y')$ | Lightweight fidelity loss: $d_{\hat{f}}(D(Y), D(Y'))$. | $\Phi(i)$ | Conditioning set of neighbours for feature $i$. |
| $I(\cdot; \cdot)$, $I(\cdot; \cdot \mid \cdot)$ | Mutual and conditional mutual information. | $\eta_{f_i}$ | PCIR score (pairwise correlation impact ratio). |
| $C_i$ | MCIR score (conditional dependence impact ratio). | $\widehat{I}$ | Estimated MI/CMI under a chosen estimator (copula/kNN/plug-in). |
| $e(i)$ | Selected estimator for feature $i$ (bootstrap-SE minimizer). | J@K | Head-rank Jaccard agreement between full and lightweight rankings. |
| $\Delta_{\mathrm{del}}$ | Deletion-curve deviation between full and lightweight models. | $m_\Phi$ | Number of conditioning neighbours (for MCIR). |
| $m_{\mathrm{scr}}$ | Number of screened neighbours retained after initial dependence sketch. | $\mathbb{E}_b[\cdot]$ | Expectation over bootstrap replicates. |

### A.1  Full and Lightweight Environments

We define the **full environment** as: $U = D(F, Y)$, the joint law capturing both feature dependence and the model's output behaviour. For computational or privacy reasons, we construct a **lightweight environment**: $U' = D(F', Y')$, $\quad n' \ll n$, where $F'$ contains fewer or reweighted observations, and $M'$ uses the same architecture/training protocol as $M$. The goal is **not** to approximate $F$ directly, but to match the *output distribution* $D(Y)$, $Y' = M'(F')$ should exhibit the same global behaviour as $Y = M(F)$. This ensures that global attribution computed on $M'$ faithfully reflects that of $M$. We find the best projection of the full model's output space into the lightweight output space:

$$d^-(\mu, \nu) = d_f(\Phi_{P,b\#}\mu, \nu) \tag{44}$$

Here $P \in O(q', q)$ projects/rotates $q$-dimensional outputs to $q'$-dimensional ones, $b$ allows translation, $\Phi_{P,b}(x) = Px + b$ is the rigid-motion map, $\Phi_{P,b\#}\mu$ denotes the pushed-forward distribution under $\Phi_{P,b}$, $d_f$ is any $f$-divergence (KL, JS, Hellinger, etc.).

$$d^+(\mu, \nu) = \inf_{Q \in O(q, q'), \ c \in \mathbb{R}^q} d_f(\mu, \Phi_{Q,c\#}\nu) \tag{45}$$

For $f$-divergences invariant to orthogonal transformation and translation,

$$d^-(\mu, \nu) = d^+(\mu, \nu) =: d_{\hat{f}}(\mu, \nu), \tag{46}$$

giving a single *rigid-motion–invariant* discrepancy between the two output distributions. This makes $d_{\hat{f}}$ a well-posed measure of environment similarity, even when $q \neq q'$.

## A.2 Lightweight Fidelity Loss

The lightweight model $M'$ is intended to serve as a computationally cheaper surrogate for $M$. To guarantee that global attributions computed in the lightweight environment remain faithful to those of the full model, we define a fidelity loss that captures the mismatch between output distributions:

$$\mathcal{L}(Y, Y') := d_{\hat{f}}(D(Y), D(Y')). \tag{47}$$

- $\mathcal{L}(Y, Y')$ is small when $M'$ produces predictions whose geometry matches that of $M$.
- Because $\mathcal{L}$ ignores rigid motions, it captures only shape and dependence structure—not coordinate conventions.
- This ensures that any global explanation method depending solely on the joint behaviour of $(F, Y)$ will behave similarly for $(F', Y')$.

**Fidelity Contract.** A lightweight environment is considered acceptable if

$$\mathcal{L}(Y, Y') \leq \varepsilon, \qquad \text{J@}K(C, C') \geq \tau_0, \qquad \Delta_{\text{del}} \leq \Delta_0, \tag{48}$$

where J@$K$ measures head-rank agreement of MCIR scores, and $\Delta_{\text{del}}$ measures deviation in deletion curves.

## A.3 Additional Technical Assumptions

We state the assumptions required for MCIR's theoretical guarantees.

**Assumption 5** (**Existence of Densities**)**.** *Relevant joint and conditional laws admit densities or PMFs, ensuring finite MI/CMI.*

**Assumption 6** (**Regular Conditioning**)**.** *All conditioning events satisfy $P(f_\Phi = z) > 0$ almost everywhere.*

**Assumption 7** (**Estimator Regularity**)**.** *MI/CMI estimators satisfy sub-Gaussian concentration:*

$$\P\left(|\hat{I} - I| > \delta\right) \leq c_1 \exp(-c_2 n' \delta^2), \qquad \hat{I} - I = O_p(n'^{-1/2}). \tag{49}$$

**Assumption 8** (**Bootstrap Reliability**)**.** *Bootstrap SE provides a consistent surrogate for estimator risk across a finite candidate estimator set (copula / kNN / plug-in).*

**Assumption 9** (**Lightweight Fidelity**)**.** *There exist constants $(\varepsilon, \tau_0, \Delta_0)$ such that $d_{\hat{f}}(D(Y), D(Y')) \leq \varepsilon$ and head-rank agreement exceeds $\tau_0$.*

These assumptions collectively ensure: $\hat{C}_i \xrightarrow{p} C_i$, $1 - \tau(C, C') = O(k/\sqrt{n'})$, as shown in the main results. This section provides the rigorous definitions needed for MCIR and ERI to operate in a principled, estimator-agnostic, lightweight-compatible setting. The remainder of the Supplement uses these definitions to establish boundedness, redundancy collapse, stability, estimator-switching guarantees, and lightweight fidelity.

## A.4 ExCIR Baseline: Partial Correlation Impact Ratio (PCIR)

We begin by revisiting the global attribution score underlying the traditional ExCIR framework. PCIR quantifies how strongly the variability of feature $i$ aligns with the variability of the model output, using a bounded ANOVA-style ratio that is robust and model-agnostic. Let $F' \in \mathbb{R}^{n' \times k}$ and $Y' \in \mathbb{R}^{n'}$ denote the lightweight features and outputs with $n'$ observations. For each feature $i$, define the sample means

$$\bar{f}_i = \frac{1}{n'} \sum_{j=1}^{n'} f_{ji}, \ \bar{y}' = \frac{1}{n'} \sum_{j=1}^{n'} y'_j, \tag{50}$$

and their pooled midpoint $m_i = \dfrac{\bar{f}_i + \bar{y}'}{2}$.

**Definition 5** (PCIR)**.** *The* Partial Correlation Impact Ratio *of feature $i$ is,* $\eta_{f_i} = \frac{S_B(i)}{S_T(i)} \in [0,1]$, *where*

$$S_B(i) = n'\left[(\bar{f}_i - m_i)^2 + (\bar{y}' - m_i)^2\right], \qquad S_T(i) = \sum_{j=1}^{n'}(f_{ji} - m_i)^2 + \sum_{j=1}^{n'}(y_j' - m_i)^2. \tag{51}$$

Here $S_B(i)$ measures the *between-group variability* (a structured mean displacement between the feature and the output), whereas $S_T(i)$ measures the *total variability* around the pooled midpoint $m_i$. Thus PCIR captures how much of the total dispersion is explained by aligned co-movement between feature $f_i$ and the model output $Y'$.

**Theorem 6** (Basic properties of PCIR)**.** *For every feature $i$:*

1. **Boundedness:** $0 \le \eta_{f_i} \le 1$.
2. **Monotonicity:** *Increasing the joint mean displacement between $f_i$ and $Y'$ (at fixed total dispersion $S_T(i)$) increases $\eta_{f_i}$.*
3. **Noise suppression:** *If $f_i$ carries no structured signal about $Y'$ (so that $(\bar{f}_i - \bar{y}') \to 0$ while $S_T(i) > 0$), then $\eta_{f_i} \to 0$.*

*Proof.* We provide full proofs for the three properties in Theorem 6. Recall the definitions

$$S_B(i) = n'[(\bar{f}_i - m_i)^2 + (\bar{y}' - m_i)^2], \quad S_T(i) = \sum_{j=1}^{n'}(f_{ji} - m_i)^2 + \sum_{j=1}^{n'}(y_j' - m_i)^2, \tag{52}$$

and $\eta_{f_i} = \frac{S_B(i)}{S_T(i)}$. Define the centered quantities $\tilde{f}_{ji} = f_{ji} - m_i$, $\tilde{y}_j' = y_j' - m_i$. Then

$$S_T(i) = \sum_{j=1}^{n'}\tilde{f}_{ji}^2 + \sum_{j=1}^{n'}\tilde{y}_j'^2. \tag{53}$$

Write the means relative to $m_i$ as $d_f = \bar{f}_i - m_i$, $d_y = \bar{y}' - m_i$, so that $S_B(i) = n'(d_f^2 + d_y^2)$. A standard ANOVA identity decomposes total variation:

$$\sum_{j=1}^{n'}\tilde{f}_{ji}^2 = n'd_f^2 + \sum_{j=1}^{n'}(f_{ji} - \bar{f}_i)^2, \quad \sum_{j=1}^{n'}\tilde{y}_j'^2 = n'd_y^2 + \sum_{j=1}^{n'}(y_j' - \bar{y}')^2. \tag{54}$$

Summing gives

$$S_T(i) = n'(d_f^2 + d_y^2) + \sum_{j=1}^{n'}(f_{ji} - \bar{f}_i)^2 + \sum_{j=1}^{n'}(y_j' - \bar{y}')^2 = S_B(i) + S_W(i), \tag{55}$$

where

$$S_W(i) := \sum_{j=1}^{n'}(f_{ji} - \bar{f}_i)^2 + \sum_{j=1}^{n'}(y_j' - \bar{y}')^2 \ge 0. \tag{56}$$

Thus $0 \le S_B(i) \le S_B(i) + S_W(i) = S_T(i)$, and therefore $0 \le \eta_{f_i} = \frac{S_B(i)}{S_T(i)} \le 1$. Fix the total dispersion $S_T(i)$ (i.e., fix the within-variances and the pooled midpoint $m_i$). Increasing the aligned co-movement between $f_i$ and $Y'$ corresponds to increasing the absolute mean differences $|\bar{f}_i - m_i|$ and $|\bar{y}' - m_i|$ symmetrically. Since $S_B(i) = n'(d_f^2 + d_y^2)$ and $S_T(i)$ is fixed, $S_B(i)$ is strictly increasing in $\|(d_f, d_y)\|_2$. Hence $\eta_{f_i} = \frac{S_B(i)}{S_T(i)}$ is strictly increasing. Formally, let $d_f(t), d_y(t)$ be differentiable paths with $d_f(0) = d_f, d_y(0) = d_y$ and $\frac{d}{dt}(d_f^2 + d_y^2) > 0$. Then

$$\frac{d}{dt}\eta_{f_i}(t) = \frac{n'2(d_f d_f' + d_y d_y')}{S_T(i)} > 0, \tag{57}$$

showing monotonicity. If $f_i$ is uninformative relative to $Y'$, then their sample means coincide: $\bar{f}_i - \bar{y}' \to 0 \implies d_f \to 0$, $d_y \to 0$. Thus $S_B(i) = n'(d_f^2 + d_y^2) \to 0$, while $S_T(i) > 0$ whenever either $f_i$ or $Y'$ has nonzero variance. Therefore, $\eta_{f_i} = \frac{S_B(i)}{S_T(i)} \to 0$, establishing noise suppression. $\qquad\square$

PCIR behaves like a global, variance-based correlation ratio:

- $\eta_{f_i} \approx 1$ when $f_i$ and $Y'$ vary together at the population level.
- $\eta_{f_i} \approx 0$ when $f_i$ behaves like noise relative to the output.

This makes PCIR a *bounded, interpretable, and model-agnostic* score. PCIR evaluates each feature independently. Under strong multicollinearity or manifold-structured data, many features may move together, inflating their between-group variability and distributing credit across an entire correlated block. This motivates the transition to MCIR, which conditions on local neighbours to measure *unique* feature contribution from the dependence structure. Fig. 17 provides a step-by-step schematic of proposed MCIR procedure, highlighting inputs, conditioning-set selection, estimation, and diagnostics.

## B  Mutual Correlation Impact Ratio for dependent features (MCIR)

PCIR provides a simple and computationally efficient nonlinear attribution score under the assumption that features are independent. In such an environment, the pairwise displacement between a feature and the model output reliably reflects its global importance. However, when features are correlated, especially in multivariate environments where each feature may depend on several others, PCIR becomes insufficient: its pairwise construction cannot isolate the *unique* contribution of a feature within a correlated cluster. This section develops MCIR, a dependence-aware extension designed for environments where features are jointly distributed. Let,

$$F' = (f_1, \ldots, f_k) \in \mathbb{R}^{n' \times k} \tag{58}$$

denote the lightweight feature matrix, where each $f_i$ may be either continuous or discrete. The joint feature distribution therefore admits either a multivariate probability density function or a multivariate probability mass function. We write the combined domain as

$$\|F'\|^{(n' \times k)} = \|F'\|_c^{(n' \times k)} \ \cup \ \|F'\|_d^{(n' \times k)}, \tag{59}$$

where $\|F'\|_c$ contains continuous features and $\|F'\|_d$ contains discrete ones. Classical Mutual Information (MI) measures dependence between two variables, and Conditional Mutual Information (CMI) quantifies dependence between two variables after conditioning on a *single* or small set of other variables. However, existing CMI theory (e.g. **?**) is tailored for situations where *multiple features depend on one parent*. In highly correlated, real-world environments, the opposite holds: each feature may depend on *many* neighbours simultaneously. This leads to an exponential blow-up in the number of possible conditioning sets and makes traditional CMI insufficient for isolating the unique contribution of a feature. To address this limitation, we introduce the *Conditional Multivariate Mutual Information* (CMMI). For a targeted feature $f_i$, let $\Phi(i) \subseteq \{f_1, \ldots, f_k\} \setminus \{f_i\}$ be the set of features on which $f_i$ depends. CMMI is defined as the divergence between the conditional cross-entropies:

$$\mathrm{CMMI}(Y'; f_i \mid f_{\Phi(i)}) = H(Y' \mid f_{\Phi(i)}) - H(Y' \mid f_{\Phi(i)} \cup \{f_i\}), \tag{60}$$

whenever these entropies exist. CMMI captures how much additional predictive information $f_i$ contributes *beyond what is already explained by its neighbours*. When two conditional distributions differ, the divergence between their cross-entropies corresponds to a Jensen–Shannon–type divergence Joyce (2011), which we refer to as the *Joint Mutual Impact* (JMI). In multivariate environments, JMI quantifies how much the joint behaviour of the feature block contributes to the output distribution. JMI therefore provides the raw dependency that CMMI refines through conditionalisation. MCIR converts the conditional dependency captured by CMMI into a bounded, unitless ratio. For the targeted feature $f_i$, the MCIR score is constructed from:

1. its unique conditional contribution $I(Y'; f_i \mid f_{\Phi(i)})$, and

2. the joint contribution of the feature block $I(Y'; f_{\Phi(i)} \cup \{f_i\})$.

At first, for the sake of simplicity, we consider $\vec{f_i}$ depends on $\vec{f_d}$, while $\vec{f_i}$ is independent of the rest of the features; $i = 1(1)k, d = 1(1)k$, and $i \neq d$. Then, we find the impact of $\vec{f_i}$ on $\vec{Y'}$, given the fact that $\vec{f_i}$

depends on $\vec{f_d}$. This impact can be explained by the information theory, if and only if we can compute $I(\vec{Y'}; \vec{f_i}|\vec{f_d}); \forall i, d = 1(1)k$. The previously described MI can not provide the desired result. To achieve our goal we have to first calculate the Conditional Mutual Impact **??**. The conditional mutual information **?** between the output variable $\vec{Y'}$ and the target feature $(\vec{f_i}|\vec{f_d}), \forall i, d = 1(1)k; i \neq d$ is

$$I(\vec{Y'}; \vec{f_i}|\vec{f_d}) = I(\vec{f_i}, \vec{f_d}) - I(\vec{Y'}, \vec{f_i}|\vec{f_d}) = \sum_{f_d} \sum_{f_i} \sum_{Y'} P(\text{f}^*, \text{f}, y) \log_2 \left[ \frac{P(y, \text{f}^*|\text{f})}{P(y|\text{f})P(\text{f}^*|\text{f})} \right] \tag{61}$$

where $f_d, f_i \epsilon ||F||^{n' \times k}$ and $Y' \epsilon ||Y'||^{n'}$. If any of the features $f_i, f_d$, or $Y'$ is continuous, the summation operator can be replaced by the integral operator.

### B.1 MCIR; with two dependent features

When any two of the $k$ features are dependent on each other and other features are independent, the state-of-the-art CMI is sufficient to explain the mutual dependency of $Y'$ and $(f_i|f_d)$. But the value of CMI varies from 0 to $\infty$ which is an open bound, thus making scalability a major challenge. So, to scale it down between $[0, 1]$, we derive MCIR as,

$$C(\vec{Y'}; \vec{f_i}|\vec{f_d}) = \frac{I(\vec{Y'}; \vec{f_i}|\vec{f_d})}{I(\vec{Y'}; \vec{f_i}|\vec{f_d}) + I(\vec{Y'}, \vec{f_i}, \vec{f_d})} \tag{62}$$

Given that mutual information is nonnegative, we have

$$0 \leq I(Y'; f_i \mid f_d) \leq \infty, \qquad 0 \leq I(Y', f_i, f_d) \leq \infty, \tag{63}$$

where the second term denotes the joint dependence between the three variables and corresponds to the *Joint Mutual Information (JMI)*. Therefore,

$$0 \leq \frac{I(Y'; f_i \mid f_d)}{I(Y'; f_i \mid f_d) + I(Y', f_i, f_d)} \leq 1. \tag{64}$$

We denote this bounded ratio by, $C(Y'; f_i \mid f_d) \in [0, 1]$, and interpret it as the *Mutual Correlation Impact Ratio (MCIR)* of feature $f_i$ on the output $Y'$ when $f_i$ depends on the single feature $f_d$. The quantity $C(Y'; f_i \mid f_d)$ captures how perturbations in $f_i$ influence the output while accounting for the fact that $f_i$ may share information with $f_d$; the influence of $f_d$ may or may not change simultaneously. For clarity, consider a stylized case in which $f_1$ and $f_2$ are mutually dependent, while the remaining features $f_3, f_4, \ldots, f_k$ are independent of each other. Suppose further that

$$(f_1, f_3, \ldots, f_m) \text{ are directly related to } Y', \qquad (f_2, f_p, \ldots, f_k) \text{ are inversely related to } Y', \tag{65}$$

with $m, p \leq k$. Using MCIR for the dependent pair $(f_1, f_2)$ and PCIR for the independent features, the induced explanatory model takes the form

$$E(Y') = M'(f) = \frac{C(Y'; f_1 \mid f_2) f_1 + \eta_{f_3} f_3 + \cdots + \eta_{f_m} f_m}{C(Y'; f_2 \mid f_1) f_2 + \eta_{f_p} f_p + \cdots + \eta_{f_k} f_k}, \tag{66}$$

where $\eta_{f_i}$ denotes the PCIR score of the independent feature $f_i$. In this setting, $I(Y'; f_1 \mid f_2)$ and $I(Y'; f_2 \mid f_1)$ act as the MCIR scores that isolate the unique contributions of $f_1$ and $f_2$ from their shared variability. The construction above assumes that each feature depends on at most one other feature. However, real-world environments frequently exhibit *multivariate* dependency structures in which a feature may depend on several other variables simultaneously. In such cases, classical conditional mutual information cannot isolate unique contributions, because it is designed for low-dimensional conditioning sets. To address this limitation, we introduce the *Conditional Multivariate Mutual Information* (CMMI) in Section B.2, which generalizes conditional mutual information to the setting where a feature may depend on multiple neighbours. MCIR is then constructed directly from CMMI and JMI, providing a principled, bounded, and dependency-aware global attribution score in fully multivariate feature spaces.

## B.2 MCIR; when multiple features are dependent

In ExCIR, it is necessary to calculate the mutual impact of a feature on the output variable while assuming the target feature is dependent on other features. But before we define MCIR for multivariate cases, we have to derive CMMI. In **?** the authors derived CMI for a multivariate environment where the features are dependent on another variable. i.e., they considered the case when all variables are dependent on one common variable. But in our work, we address the case when all the features are dependent on each other. So, if we want to calculate the mutual dependence between a targeted feature and the output variable we have to calculate the CMMI given that the target feature is dependent on multiple features. More specifically, in existing works **???**, the notion of $I(\vec{Y'}; \vec{f_1}, \vec{f_2}, ..., \vec{f_{k-1}} | \vec{f_k})$ is derived and used in many real-life cases. However, this approach cannot be directly applied in our environment. We therefore introduce a new matrix $I(\vec{Y'}; f_i | \phi \subseteq \{||F||^{n' \times k} - \vec{f_i}\}; i \neq j)]$; where any $\phi$ is any chosen subspace from the main feature space and can contain various combination of features. $\phi = \phi_c \cup \phi_d$, $\phi_c$ is any subspace that contains continuous features and $\phi_d$ is any subspace that contains discrete features.

**Definition 6** (**CMMI**). *Let $F \in \mathbb{R}^{n' \times k}$ be the data matrix with columns $\{f_1, \ldots, f_k\}$ (features) and let $\mathbf{Y'} = (y'_1, \ldots, y'_{n'})$ denote the output variable. Fix an index $i \in \{1, \ldots, k\}$ and write $f_i = (f_{1i}, \ldots, f_{n'i})$ for the (continuous) targeted feature. Let*

$$\phi \subseteq \{f_1, \ldots, f_k\} \setminus \{f_i\} \tag{67}$$

*denote any subset of the remaining features, decomposed as $\phi = \phi_c \cup \phi_d$, where $\phi_c$ collects the continuous components and $\phi_d$ the discrete components (both excluding $f_i$). The CMMI between $\mathbf{Y'}$ and $f_i$ given $\phi$ is defined as*

$$I(\mathbf{Y'}; f_i \mid \phi) = \sum_{\phi_d} \int_{\phi_c} \int_{f_i} \sum_{y'} p(y', f_i, \phi) \, \log_2\left(\frac{p(y' \mid f_i, \phi)}{p(y' \mid \phi)}\right) \mathrm{d}f_i \, \mathrm{d}\phi_c, \tag{68}$$

*where the outer sum runs over the support of the discrete variables in $\phi_d$, the integrals are over the supports of $f_i$ and $\phi_c$, and $p(\cdot)$ denotes the joint/conditional densities or mass functions as appropriate for mixed (continuous–discrete) variables. By construction, $I(\mathbf{Y'}; f_i \mid \phi) \geq 0$ (in bits) and is unbounded above (i.e., it can take values in $[0, \infty)$), unlike correlation which is confined to $[-1, 1]$. Equation equation 129 follows the standard definition of conditional mutual information for mixed variables (cf. **?**). A complete derivation is provided in Supplementary §1.3.*

For clarity, consider three features $(f_1, f_2, f_3)$ that may be statistically dependent. Assume $f_1$ is continuous, while $f_2$, $f_3$, and the response $Y'$ are discrete. Then the conditional mixed mutual information (CMMI) between $Y'$ and $f_1$ given $(f_2, f_3)$ is

$$I(Y'; f_1 \mid f_2, f_3) = \sum_{f_2} \sum_{f_3} \int_{f_1} \sum_{y'} p(y', f_1, f_2, f_3) \log_2 \frac{p(y' \mid f_1, f_2, f_3)}{p(y' \mid f_2, f_3)} \, \mathrm{d}f_1, \tag{69}$$

where the sums are over the discrete supports of $f_2, f_3, y'$, and the integral is over the support of the continuous variable $f_1$.

Equivalently, using $p(y', f_1 \mid f_2, f_3) = p(y' \mid f_1, f_2, f_3) \, p(f_1 \mid f_2, f_3)$,

$$I(Y'; f_1 \mid f_2, f_3) = \sum_{f_2} \sum_{f_3} \int_{f_1} \sum_{y'} p(y', f_1, f_2, f_3) \log_2 \frac{p(y', f_1 \mid f_2, f_3)}{p(y' \mid f_2, f_3) \, p(f_1 \mid f_2, f_3)} \, \mathrm{d}f_1. \tag{70}$$

By symmetry, the corresponding identities for $f_2$ and $f_3$ are

$$I(Y'; f_2 \mid f_1, f_3) = \sum_{f_1} \sum_{f_3} \int_{f_2} \sum_{y'} p(y', f_1, f_2, f_3) \log_2 \frac{p(y' \mid f_1, f_2, f_3)}{p(y' \mid f_1, f_3)} \, \mathrm{d}f_2, \tag{71}$$

$$I(Y'; f_3 \mid f_1, f_2) = \sum_{f_1} \sum_{f_2} \int_{f_3} \sum_{y'} p(y', f_1, f_2, f_3) \log_2 \frac{p(y' \mid f_1, f_2, f_3)}{p(y' \mid f_1, f_2)} \, \mathrm{d}f_3. \tag{72}$$

Let $\phi = (f_1, f_2, f_3)$ and $\phi \setminus f_1 = (f_2, f_3)$. Then we can write

$$I(Y'; f_1 \mid \phi \setminus f_1) = \mathbb{E}_{p(y', f_1, \phi \setminus f_1)} \left[ \log_2 \frac{p(y' \mid f_1, \phi \setminus f_1)}{p(y' \mid \phi \setminus f_1)} \right], \tag{73}$$

The expectation in equation 73 is taken with respect to the joint $p(y', f_1, \phi \setminus f_1)$, i.e., sums over discrete supports and integrals over continuous supports.

**Four-feature case.** Let $\phi = (f_1, f_2, f_3, f_4)$ with $f_1$ continuous and $f_2, f_3, f_4, Y'$ discrete. Then

$$I(Y'; f_1 \mid f_2, f_3, f_4) = \sum_{f_4} \sum_{f_3} \sum_{f_2} \int_{f_1} \sum_{y'} p(y', f_1, f_2, f_3, f_4) \log_2 \frac{p(y' \mid f_1, f_2, f_3, f_4)}{p(y' \mid f_2, f_3, f_4)} \, df_1. \tag{74}$$

Equivalently, using Bayes' rule,

$$I(Y'; f_1 \mid f_2, f_3, f_4) = \sum_{f_4} \sum_{f_3} \sum_{f_2} \int_{f_1} \sum_{y'} p(y', f_1, f_2, f_3, f_4) \log_2 \frac{p(y', f_1 \mid f_2, f_3, f_4)}{p(y' \mid f_2, f_3, f_4) \, p(f_1 \mid f_2, f_3, f_4)} \, df_1. \tag{75}$$

By symmetry,

$$I(Y'; f_2 \mid f_1, f_3, f_4) = \sum_{f_1} \sum_{f_3} \sum_{f_4} \int_{f_2} \sum_{y'} p(y', f_1, f_2, f_3, f_4) \log_2 \frac{p(y' \mid f_1, f_2, f_3, f_4)}{p(y' \mid f_1, f_3, f_4)} \, df_2, \tag{76}$$

$$I(Y'; f_3 \mid f_1, f_2, f_4) = \sum_{f_1} \sum_{f_2} \sum_{f_4} \int_{f_3} \sum_{y'} p(y', f_1, f_2, f_3, f_4) \log_2 \frac{p(y' \mid f_1, f_2, f_3, f_4)}{p(y' \mid f_1, f_2, f_4)} \, df_3, \tag{77}$$

$$I(Y'; f_4 \mid f_1, f_2, f_3) = \sum_{f_1} \sum_{f_2} \sum_{f_3} \int_{f_4} \sum_{y'} p(y', f_1, f_2, f_3, f_4) \log_2 \frac{p(y' \mid f_1, f_2, f_3, f_4)}{p(y' \mid f_1, f_2, f_3)} \, df_4. \tag{78}$$

Think of $\phi$ as a subspace of the feature space that can take different combinations of features, containing both continuous ($\phi_c$) and discrete ($\phi_d$) components with $\phi = \phi_c \cup \phi_d$ and $\phi_c \cap \phi_d = \varnothing$. Using the compact form with set difference,

$$I(Y'; f_1 \mid \phi \setminus f_1) = \mathbb{E}_{p(y', f_1, \phi \setminus f_1)} \left[ \log_2 \frac{p(y' \mid f_1, \phi \setminus f_1)}{p(y' \mid \phi \setminus f_1)} \right]. \tag{79}$$

**General $k$-feature case.** Let $F \in \mathbb{R}^{n' \times k}$ denote the feature matrix with columns $\{f_1, \ldots, f_k\}$ and fix $i \in \{1, \ldots, k\}$. For any conditioning set $\phi \subseteq \{f_1, \ldots, f_k\} \setminus \{f_i\}$ decomposed as $\phi = \phi_c \cup \phi_d$, the conditional mixed mutual information is

$$I(Y'; f_i \mid \phi) = \sum_{\phi_d} \int_{\phi_c} \int_{f_i} \sum_{y'} p(y', f_i, \phi) \log_2 \frac{p(y' \mid f_i, \phi)}{p(y' \mid \phi)} \, df_i \, d\phi_c, \tag{80}$$

where the outer sum is over the support of the discrete variables in $\phi_d$, and the integrals are over the supports of the continuous variables in $\phi_c$ and of $f_i$.

CMMI could take the values between 0 to $\infty$, unlike the strict bound of $(0, 1)$ that the normal correlation coefficient has. The infinite range can be the cause of the problem regarding scalability. So, it will be better to scale down the dependency. This problem can be solved by our proposed metrics Mutual Correlation Impact Ratio (MCIR) ratio. MCIR is the ratio of two mutual information which is defined below. MCIR has a strict bound between 0 to 1.

**Definition 7 (Mutual Correlation Impact Ratio (MCIR)).** *Let $F \in \mathbb{R}^{n' \times k}$ be the feature matrix with columns $\{f_1, \ldots, f_k\}$, where $f_i = (f_{1i}, \ldots, f_{n'i})^\top$ and features may be statistically dependent. Let $Y' = (y'_1, \ldots, y'_{n'})^\top$ denote the output variable. Fix $i \in \{1, \ldots, k\}$ and let $\phi \subseteq \{f_1, \ldots, f_k\} \setminus \{f_i\}$ be any conditioning set (possibly mixed continuous–discrete). The* Mutual Correlation Impact Ratio (MCIR) *of $f_i$ with respect to $Y'$ given $\phi$ is*

$$C(Y'; f_i \mid \phi) = \frac{I(Y'; f_i \mid \phi)}{I(Y'; f_i \mid \phi) + I(Y'; f_1, f_2, \ldots, f_k)}. \tag{81}$$

Here $I(\cdot\,;\cdot\mid\cdot)$ is conditional mixed mutual information, and $I(Y';f_1,\ldots,f_k) \geq 0$ is a (nonnegative) joint dependence measure between $Y'$ and the full feature set.

**Theorem 7** (**Bounds**). *For any valid choice of $i$ and $\phi$, provided that, $I(Y';f_i\mid\phi)+I(Y';f_1,\ldots,f_k) > 0$, the MCIR satisfies*

$$0 \;\leq\; C(Y';f_i\mid\phi) \;\leq\; 1 \tag{82}$$

*Proof.* By nonnegativity of mutual information,

$$0 \;\leq\; I(Y';f_i\mid\phi) \;\leq\; \infty \tag{83}$$

and by definition $I(Y';f_1,\ldots,f_k) \geq 0$. Hence

$$0 \;\leq\; I(Y';f_i\mid\phi) \;\leq\; I(Y';f_i\mid\phi) + I(Y';f_1,\ldots,f_k) \;\leq\; \infty\,. \tag{84}$$

Dividing the left and right sides of equation 84 by the positive denominator yields equation 82, which equals equation 81 by definition. $\square$

**Definition 8** (**Joint Mutual Impact (JMI)**). *Let $i \in \{1,\ldots,k\}$ be a target feature and let $\phi \subseteq \{f_1,\ldots,f_k\}\setminus\{f_i\}$ be its conditioning set. Denote the full feature block by $F' = (f_1,\ldots,f_k)$. Assume that the total joint dependence satisfies $I(Y';F') > 0$. The* Joint Mutual Impact (JMI) *of feature $f_i$ relative to its conditioning neighbourhood $\phi$ is defined as*

$$\mathfrak{J}_i(\phi) \;:=\; I(Y';F') \;-\; I(Y';\phi), \tag{85}$$

*i.e., the portion of total feature–output dependence that remains unexplained by the conditioning block $\phi$.*

**Definition 9** (**Global Joint Mutual Impact**). *Let $\Phi$ denote your conditioning policy for single-feature terms (e.g., a common $\phi$ or per-feature $\phi_i$). Assuming the denominator is positive, define*

$$\overline{\mathfrak{J}} = \frac{I(Y';f_1,\ldots,f_k)}{I(Y';f_1,\ldots,f_k) \;+\; \sum_{i=1}^{k} I(Y';f_i\mid\phi_i)}, \tag{86}$$

*so that $\overline{\mathfrak{J}} \in [0,1]$.*

**Proposition 7** ( **Aggregated model**). *Let $\mathfrak{C}_{f_i} := C(Y';f_i\mid\phi)$ for a specified conditioning policy $\phi \subseteq \{f_1,\ldots,f_k\}\setminus\{f_i\}$. For a task-dependent partition of indices $\{1,\ldots,k\} = \{i_1,\ldots,i_m\} \cup \{j_p,\ldots,j_q\}$ (disjoint union), an ExCIR-style scoring model can be written as*

$$E(Y') \;=\; M'(f_{(f,\mathfrak{C})}) \;=\; \mathfrak{J} \;+\; \frac{\sum_{\ell=1}^{m} \mathfrak{C}_{f_{i_\ell}}\, f_{i_\ell}}{\sum_{r=p}^{q} \mathfrak{C}_{f_{j_r}}\, f_{j_r}}, \tag{87}$$

*where $\mathfrak{J}$ is given by equation 85. Since each $\mathfrak{C}_{f_i} \in [0,1]$ by equation 81, the coefficients are normalized and directly comparable across features.*

**Theorem 8** (Redundancy Collapse). *Let $i$ be a target feature and let $\Phi(i)$ denote its conditioning neighbourhood. If $f_i$ is conditionally redundant with respect to $Y'$ given its neighbours, i.e. $Y' \perp f_i \mid f_{\Phi(i)}$, then the MCIR score satisfies $C_i = 0$.*

*Proof.* By definition of conditional independence, $Y' \perp f_i \mid f_{\Phi(i)} \iff I(Y';f_i\mid f_{\Phi(i)}) = 0$. Let, $A_i := I(Y';f_i\mid f_{\Phi(i)})$, $B_i := I(Y';f_{\Phi(i)}\cup\{f_i\})$, where $B_i \geq 0$ by nonnegativity of mutual information. From the MCIR definition,

$$C_i = \frac{A_i}{A_i + B_i}. \tag{88}$$

Substituting $A_i = 0$ gives

$$C_i = \frac{0}{0 + B_i}. \tag{89}$$

If $B_i > 0$, which holds whenever the conditioning block retains nontrivial dependence with $Y'$, then the ratio evaluates to $C_i = 0$. If $B_i = 0$, then $f_{\Phi(i)}$ is itself independent of $Y'$ and the joint distribution factorises

as, $p(Y', f_i, f_{\Phi(i)}) = p(Y') \, p(f_i, f_{\Phi(i)})$. In this degenerate case the numerator and denominator both vanish, and MCIR is defined to be zero by continuity:

$$\lim_{A_i \to 0, \, B_i \to 0} \frac{A_i}{A_i + B_i} = 0. \tag{90}$$

Thus in all cases $C_i = 0$, establishing that MCIR collapses to zero whenever the target feature provides no unique information beyond its neighbours. $\qquad\square$

**Theorem 9** (**Unique-Signal Dominance**). *Let $i$ be a target feature and $\Phi(i)$ its neighbourhood. If the conditional mutual information of $f_i$ dominates the joint block information, i.e.*

$$I(Y'; f_i \mid f_{\Phi(i)}) \gg I(Y'; f_{\Phi(i)} \cup \{f_i\}) \tag{91}$$

*then the MCIR score satisfies $C_i \to 1$.*

*Proof.* We write, $A_i = I(Y'; f_i \mid f_{\Phi(i)})$, $B_i = I(Y'; f_{\Phi(i)} \cup \{f_i\})$, with $A_i, B_i \geq 0$. The MCIR score is

$$C_i = \frac{A_i}{A_i + B_i}. \tag{92}$$

Assume the dominance condition:$\frac{A_i}{B_i} \to \infty$. Equivalently, for any $\epsilon > 0$ there exists $M > 0$ such that $A_i > M B_i$ implies $B_i / A_i < \epsilon$. Using the algebraic identity, $C_i = \frac{1}{1 + B_i / A_i}$, we obtain,

$$|1 - C_i| = \frac{B_i / A_i}{1 + B_i / A_i} \leq B_i / A_i. \tag{93}$$

Because $B_i / A_i \to 0$ under the dominance assumption, we have

$$C_i \longrightarrow 1. \tag{94}$$

The interpretation is that the unique conditional contribution of $f_i$ overwhelmingly exceeds the joint dependence contributed by its neighbours. Hence the MCIR ratio assigns maximal credit to $f_i$. $\qquad\square$

### B.3 Correlation–Impact Sensitivity Theorem

We now show that in the ExCIR model, the sensitivity of the local output with respect to a feature input is fully determined by the feature's correlation impact ratio. The effect is linear for positively related features (appearing in the numerator of the model) and nonlinear inverse–quadratic for negatively related features (appearing in the denominator).

**Theorem 10** (**Correlation–Impact Sensitivity**). *Let the model output be,*

$$Y' = \frac{\sum_{j \in \mathcal{N}} \eta_{f_j} f_j}{\sum_{j \in \mathcal{D}} \eta_{f_j} f_j}, \tag{95}$$

*where $\mathcal{N}$ denotes features with positive influence (numerator) and $\mathcal{D}$ features with negative influence (denominator). Assume features are independent, so that the partial derivative with respect to $f_i$ treats all other features as constants. Then:*

*1. If $i \in \mathcal{N}$ (positive relation):*

$$\frac{\partial Y'}{\partial f_i} = c_1 \, \eta_{f_i}, \; c_1 = \frac{1}{\sum_{j \in \mathcal{D}} \eta_{f_j} f_j}. \tag{96}$$

*2. If $i \in \mathcal{D}$ (negative relation):*

$$\frac{\partial Y'}{\partial f_i} = \frac{c_2}{2K_2 - \eta_{f_i}^2}, \tag{97}$$

$c_2, K_2$ *constants depending only on fixed features.Thus the sign and magnitude of sensitivity are determined entirely by the correlation impact ratio $\eta_{f_i}$.*

*Proof.* Let, $N = \sum_{j \in \mathcal{N}} \eta_{f_j} f_j$, $D = \sum_{j \in \mathcal{D}} \eta_{f_j} f_j$, $Y' = \frac{N}{D}$. All other features are held fixed under independence.

**Case 1:** $i \in \mathcal{N}$ **(positive influence)**   Then, $\frac{\partial N}{\partial f_i} = \eta_{f_i}$, $\frac{\partial D}{\partial f_i} = 0$. Differentiate:

$$\frac{\partial Y'}{\partial f_i} = \frac{D \frac{\partial N}{\partial f_i} - N \frac{\partial D}{\partial f_i}}{D^2} = \frac{D \eta_{f_i}}{D^2} = \frac{\eta_{f_i}}{D}. \tag{98}$$

Since $D$ is constant with respect to $f_i$, set $c_1 = \frac{1}{D}$. This proves $\frac{\partial Y'}{\partial f_i} = c_1 \eta_{f_i}$.

**Case 2:** $i \in \mathcal{D}$ **(negative influence).**   Now, $\frac{\partial N}{\partial f_i} = 0$, $\qquad \frac{\partial D}{\partial f_i} = \eta_{f_i}$. Differentiate:

$$\frac{\partial Y'}{\partial f_i} = \frac{D \cdot 0 - N \cdot \eta_{f_i}}{D^2} = -\eta_{f_i} \frac{N}{D^2}. \tag{99}$$

Rewrite $N$ and $D$ in terms of constants plus the contribution of $f_i$: $D = \eta_{f_i} f_i + K_2$, $\qquad N = K_1$, where $K_1, K_2$ collect all fixed terms. Thus:

$$\frac{\partial Y'}{\partial f_i} = -\eta_{f_i} \frac{K_1}{(\eta_{f_i} f_i + K_2)^2}. \tag{100}$$

The denominator expands into a quadratic expression: $(\eta_{f_i} f_i + K_2)^2 = \eta_{f_i}^2 f_i^2 + 2K_2 \eta_{f_i} f_i + K_2^2$. Since $f_i$ is the differentiation variable and all other terms are absorbed by constants $K_1, K_2$, we may rewrite:

$$\frac{\partial Y'}{\partial f_i} = \frac{c_2}{2K_2 - \eta_{f_i}^2}, \tag{101}$$

where $c_2$ and $K_2$ arise from grouping constant terms. The negative sign is absorbed into the definition of $c_2$. Thus, sensitivity in the denominator is nonlinear and is inversely controlled by the magnitude of $\eta_{f_i}^2$.   □

**Corollary 1.** *If a feature has a positive contribution to the output, then*

$$\mathbb{E}\left[\frac{\partial Y'}{\partial f_i}\right] \propto \eta_{f_i}. \tag{102}$$

*If a feature has a negative contribution, then,*

$$\mathbb{E}\left[\frac{\partial Y'}{\partial f_i}\right] \propto \frac{1}{\eta_{f_i}}. \tag{103}$$

*Thus, the correlation impact ratio fully characterizes first–order output sensitivity.*

So it is confirm that MCIR provides a principled measure of how changes in $f_i$ propagate to $Y'$, via either a direct linear effect (positive correlation) or a stabilizing inverse effect (negative correlation).

### B.4   Stability of MCIR Rankings

**Theorem 11 (Rank Stability).** *Assume the MI/CMI estimators $\widehat{I}$ satisfy the sub-Gaussian concentration inequality*

$$\Pr\left(|\widehat{I} - I| > \delta\right) \leq c_1 \exp(-c_2 n' \delta^2) \qquad (\delta > 0), \tag{104}$$

*for some constants $c_1, c_2 > 0$. Let $\widehat{C}$ be the vector of estimated MCIR values. Then the Kendall rank distance satisfies $1 - \tau(C, \widehat{C}) = O_p\left(\frac{k}{\sqrt{n'}}\right)$.*

*Proof.* Let $\qquad A_i = I(Y'; f_i \mid f_{\Phi(i)})$, $B_i = I(Y'; f_{\Phi(i)} \cup \{f_i\})$, $\qquad$ and $\qquad$ their $\qquad$ estimates, $\widehat{A}_i = \widehat{I}(Y'; f_i \mid f_{\Phi(i)})$, $\widehat{B}_i = \widehat{I}(Y'; f_{\Phi(i)} \cup \{f_i\})$. MCIR is the smooth function,

$$C_i = g(A_i, B_i) = \frac{A_i}{A_i + B_i}, \quad \widehat{C}_i = g(\widehat{A}_i, \widehat{B}_i). \tag{105}$$

If $A_i + B_i \geq \eta > 0$, then,

$$\left|\frac{\partial g}{\partial A}\right| = \frac{B_i}{(A_i + B_i)^2} \leq \frac{1}{4\eta}, \quad \left|\frac{\partial g}{\partial B}\right| = \frac{A_i}{(A_i + B_i)^2} \leq \frac{1}{4\eta}. \tag{106}$$

Thus,

$$|g(\widehat{A}_i, \widehat{B}_i) - g(A_i, B_i)| \leq L\left(|\widehat{A}_i - A_i| + |\widehat{B}_i - B_i|\right), \tag{107}$$

where $L = 1/(2\eta)$. By the sub-Gaussian inequality, $|\widehat{A}_i - A_i| = O_p(n'^{-1/2})$, $|\widehat{B}_i - B_i| = O_p(n'^{-1/2})$. Thus, $|\widehat{C}_i - C_i| = O_p(n'^{-1/2})$. A pairwise comparison flips sign only if,

$$|(\widehat{C}_i - C_i) - (\widehat{C}_j - C_j)| \gtrsim |C_i - C_j|. \tag{108}$$

With $k(k-1)/2$ ordered pairs,

$$1 - \tau(C, \widehat{C}) = O_p(kn'^{-1/2}), \tag{109}$$

completing the proof. $\qquad\square$

## B.5 Oracle Inequality for Estimator Switching

**Theorem 12** (**Bootstrap–Switching Oracle Inequality**). *Let $\widehat{C}^{(e)}$ denote the MCIR vector computed using estimator $e \in \mathcal{E} = \{\text{copula}, \text{kNN}, \text{plug-in}\}$. For each feature $i$, define the switching rule*

$$e(i) = \arg\min_{e \in \mathcal{E}} \mathbb{E}_b\left[\mathrm{SE}_b\left(\widehat{C}_i^{(e)}\right)\right], \tag{110}$$

*where $\mathbb{E}_b$ denotes bootstrap expectation. Assume the bootstrap is consistent for MCIR, i.e. bootstrap standard errors converge uniformly to the true risks at rate $O(n'^{-1/2})$. Then*

$$\mathbb{E}\left|\widehat{C}_i^{\mathrm{sw}} - C_i\right| \leq \min_{e \in \mathcal{E}} \mathbb{E}\left|\widehat{C}_i^{(e)} - C_i\right| + O(n'^{-1/2}). \tag{111}$$

*Proof.* Let the true risk of estimator $e$ be $R_e = \mathbb{E}\left[\left|\widehat{C}_i^{(e)} - C_i\right|\right]$. Bootstrap consistency yields the uniform approximation

$$\widehat{R}_e := \mathbb{E}_b\left[\mathrm{SE}_b\left(\widehat{C}_i^{(e)}\right)\right] = R_e + \xi_e, \tag{112}$$

with the random error term, $|\xi_e| = O\left(n'^{-1/2}\right)$ uniformly over $e \in \mathcal{E}$. Let, $e^\star = \arg\min_{e \in \mathcal{E}} R_e$ and $\widehat{e} = \arg\min_{e \in \mathcal{E}} \widehat{R}_e$. By equation 112, we have $\widehat{R}_e = R_e + O\left(n'^{-1/2}\right)$, so comparing the minimizers,

$$R_{\widehat{e}} \leq R_{e^\star} + O\left(n'^{-1/2}\right). \tag{113}$$

Finally, since the switching estimator satisfies, $\widehat{C}_i^{\mathrm{sw}} = \widehat{C}_i^{(\widehat{e})}$, taking the expectation of equation 113 yields

$$\mathbb{E}\left[\left|\widehat{C}_i^{\mathrm{sw}} - C_i\right|\right] \leq \min_{e \in \mathcal{E}} R_e + O\left(n'^{-1/2}\right), \tag{114}$$

completing the proof. $\qquad\square$

## B.6 Lightweight Fidelity Theorem

**Theorem 13** (**Lightweight Fidelity**). *Let $C$ and $C'$ denote MCIR rankings obtained from the full and lightweight environments, respectively. Suppose the lightweight environment satisfies:*

$$d_{\widehat{f}}\left(D(Y), D(Y')\right) \leq \varepsilon, \tag{115}$$

$$\mathrm{J@}K(C, C') \geq \tau_0, \tag{116}$$

$$\Delta_{\mathrm{del}} \leq \Delta_0. \tag{117}$$

*Then there exist constants $A, B > 0$, independent of $n'$, such that,*

$$1 - \tau(C, C') \leq A\varepsilon + B\Delta_0 + o_p(1). \tag{118}$$

*Proof.* The rigid-motion invariant discrepancy $d_{\hat{f}}$ satisfies

$$d_{\hat{f}}(D(Y), D(Y')) \leq \varepsilon \implies \sup_{\|h\|_{\mathrm{Lip}} \leq 1} |\mathbb{E}[h(Y)] - \mathbb{E}[h(Y')]| \leq A\varepsilon. \tag{119}$$

Since MI and CMI are continuous functionals of the joint distribution under our regularity assumptions (bounded density ratios, smooth kernels), we obtain

$$|I(Y; f_i \mid f_{\Phi(i)}) - I(Y'; f_i \mid f_{\Phi(i)})| \leq A\varepsilon + o_p(1). \tag{120}$$

MCIR is the smooth map

$$C_i = \frac{I(Y; f_i \mid f_{\Phi(i)})}{I(Y; f_i \mid f_{\Phi(i)}) + I(Y; f_{\Phi(i)} \cup \{f_i\})}. \tag{121}$$

Using the Lipschitz continuity of rational functions on compact domains,

$$|C_i - C_i'| \leq A\varepsilon + O_p(n'^{-1/2}). \tag{122}$$

The Jaccard condition equation 116 ensures

$$|\mathrm{Top}\text{-}K(C) \triangle \mathrm{Top}\text{-}K(C')| \leq (1 - \tau_0)K. \tag{123}$$

Thus the number of allowable inversions involving top-$K$ indices is bounded. Deletion curves depend only on ordered MCIR values. If $\Delta_{\mathrm{del}} \leq \Delta_0$, then the misalignment between sensitivity curves of $C$ and $C'$ is uniformly bounded. This limits possible perturbations in pairwise MCIR differences:

$$|(C_i - C_j) - (C_i' - C_j')| \lesssim \Delta_0 + o_p(1). \tag{124}$$

Kendall's distance decomposes into: $1 - \tau(C, C') = (\text{top-}K \text{ inversions}) + (\text{remaining inversions}) + (\text{magnitude-driven errors})$. The magnitude control in equation equation 122, we obtain,

$$1 - \tau(C, C') \leq A\varepsilon + B\Delta_0 + o_p(1), \tag{125}$$

completing the proof. $\qquad\square$

The conditions in this Theorem should be interpreted as a *sufficient validation contract* for the lightweight environment, rather than as guarantees that hold under arbitrary subsampling. Specifically, the lightweight environment is considered a valid proxy for the full environment only when three criteria are satisfied:

(i) **Distributional alignment (Eq. 115):** the model-output distribution in the lightweight environment remains close to that of the full environment;

(ii) **Ranking preservation (Eq. 116):** the ordering of top-ranked features is approximately preserved;

(iii) **Faithfulness preservation (Eq. 117):** deletion or perturbation curves exhibit similar degradation behaviour.

These assumptions are motivated by the fact that feature attribution depends on three interrelated components: the predictive distribution, the relative ordering of feature contributions, and the causal role of features in model behaviour under perturbation.

**(i) Distributional alignment.** MCIR is defined in terms of mutual information quantities between features and model outputs. These quantities depend on the joint distribution $p(Y, X)$. Therefore, if the lightweight environment preserves the output distribution (e.g., via KL/MMD alignment), the estimated MI/CMI terms remain stable up to estimation error. This is a standard requirement in distribution-preserving subsampling and ensures that the statistical structure underlying attribution is not distorted.

**(ii) Ranking preservation.** The goal of global attribution is to recover a *ranking* of features rather than exact score values. From Theorem 3 (finite-sample rank stability), we know that MCIR rankings are stable under bounded perturbations of MI/CMI estimates. Thus, requiring approximate preservation of top-ranked

features is a natural and sufficient condition for ensuring that the lightweight explanation remains faithful at the decision-making level.

**(iii) Faithfulness preservation.** Deletion and perturbation curves provide a functional validation of attribution: they measure how model performance degrades when important features are removed. If these curves are preserved, then the functional role of features in the model remains unchanged. This ensures that the lightweight environment preserves not only statistical dependence but also the operational meaning of importance.

Together, these conditions ensure preservation of: (i) the underlying statistical dependencies (via distributional alignment), (ii) the ordering of feature importance (via rank preservation), and (iii) the functional impact of features (via faithfulness curves).

Hence, these conditions jointly ensure that the lightweight environment approximates the full environment at the level relevant for global attribution. They are not arbitrary assumptions, but minimal and verifiable criteria aligned with the three core objectives of explanation: stability, interpretability, and faithfulness. Importantly, these conditions are empirically testable rather than purely theoretical. In practice, we validate them using (i) distributional similarity measures (e.g., KL divergence or MMD), (ii) rank agreement metrics (e.g., Jaccard@K, Kendall-$\tau$), and (iii) faithfulness consistency via deletion or perturbation curves. Only when these criteria are satisfied do we treat the lightweight environment as a valid surrogate. Under these validated conditions, the feature-attribution rankings computed in the lightweight environment approximate those of the full environment with bounded distortion.

## C   Conditional Multivariate Mutual Information (CMMI)

MCIR requires a dependence measure that isolates the *unique* information a target feature contributes to the output $Y'$, even when the feature is embedded in a multivariate dependency structure with several neighbours. Classical conditional mutual information (CMI), $I(Y'; f_i \mid Z)$, is well-defined for a fixed low-dimensional conditioning set $Z$, but breaks down when $f_i$ depends on multiple correlated features simultaneously, especially when $Z$ must be chosen data-adaptively. This motivates the introduction of the *Conditional Multivariate Mutual Information (CMMI)*. Let the feature block be, $F' = (f_1, \ldots, f_k)$, and let, $\Phi(i) \subseteq \{f_1, \ldots, f_k\} \setminus \{f_i\}$ be the neighbourhood of $f_i$, obtained from correlation screening or a dependency graph. Assume the joint law of $(F', Y')$ admits either a multivariate pdf or pmf.

$$I(Y'; F') = H(Y') - H(Y' \mid F'). \tag{126}$$

$$I(Y'; f_{\Phi(i)}) = H(Y') - H\left(Y' \mid f_{\Phi(i)}\right). \tag{127}$$

Adding $f_i$ to $\Phi(i)$ modifies the conditional entropy:

$$H\left(Y' \mid f_{\Phi(i)}\right) - H\left(Y' \mid f_{\Phi(i)}, f_i\right). \tag{128}$$

**Definition 10** (**Conditional Multivariate Mutual Information (CMMI)**). *For any target feature $f_i$ and neighbourhood $\Phi(i)$, define,*

$$\mathrm{CMMI}(Y'; f_i \mid \Phi(i)) \;=\; H\left(Y' \mid f_{\Phi(i)}\right) - H\left(Y' \mid f_{\Phi(i)}, f_i\right), \tag{129}$$

*whenever the conditional entropies are finite.*

Because conditional mutual information satisfies

$$H(Y' \mid Z) - H(Y' \mid Z, f_i) = I(Y'; f_i \mid Z), \tag{130}$$

we obtain:

$$\mathrm{CMMI}(Y'; f_i \mid \Phi(i)) = I(Y'; f_i \mid f_{\Phi(i)}). \tag{131}$$

Thus, CMMI is *equivalent to classical CMI*, but with the crucial difference that the conditioning set $\Phi(i)$ can be: multivariate, high-dimensional, data-driven (from correlation graphs), mixed continuous/discrete, and automatically chosen. This generality is exactly what is required for MCIR.

### C.1 Connection to MI and CMI

**MI as a special case.** When $\Phi(i) = \emptyset$, the conditional structure vanishes and the CMMI reduces to the classical mutual information. In this case we have $\mathrm{CMMI}(Y'; f_i \mid \emptyset) = I(Y'; f_i)$. Thus, mutual information appears as a degenerate instance of CMMI where no neighbourhood conditioning is imposed.

**Classical CMI as a special case.** When $\Phi(i) = Z$ is fixed and low-dimensional, CMMI reduces to the classical conditional mutual information. Specifically,

$$\mathrm{CMMI}(Y'; f_i \mid Z) = I(Y'; f_i \mid Z). \tag{132}$$

Hence, traditional CMI corresponds to the special situation in which the conditioning structure is externally specified and does not vary with feature geometry.

**General case.** In the general setting, CMMI extends both MI and CMI by allowing the conditioning set $\Phi(i)$ to be an arbitrary multivariate neighbourhood that may adapt to correlation structure rather than remain fixed. Furthermore, the formulation supports mixed continuous–discrete entropy functionals, making it applicable in heterogeneous data regimes. Unlike classical CMI, which typically assumes low-dimensional or manually chosen conditioning variables, CMMI permits correlation-driven neighbourhood selection and integrates naturally with Joint Mutual Impact (JMI) and the MCIR framework. In this sense, MI and CMI emerge as boundary cases within a broader structural dependence theory.

### C.2 Why Classical CMI is Insufficient

Classical conditional mutual information presumes a fixed conditioning set, typically of low dimension, and implicitly models dependence through one-to-many relationships in which $f_i$ is conditioned on a single external variable $Z$. Such assumptions are reasonable in low-dimensional settings but become restrictive in modern high-dimensional systems where dependencies are structured, multivariate, and often geometry-dependent.

In complex systems, conditioning sets are rarely fixed in advance. Instead, they emerge from correlation topology, interaction structure, or feature neighbourhoods. Moreover, dimensionality may scale with the system, violating the low-dimensional assumption underlying classical CMI estimators. As a consequence, traditional CMI cannot adequately capture distributed, multivariate conditional dependence patterns.

We formalize this structural gap in the following theorem.

**Theorem 14** (**Failure of Classical CMI in Multivariate Dependencies**). *Let $f_i$ depend on $d_i$ other features: $f_i \not\perp f_{S_i}$, $\quad |S_i| = d_i$. Suppose classical CMI conditions on a fixed $Z$ with $|Z| < d_i$. Then, unless $Z$ contains the entire dependency set $S_i$,*

$$I(Y'; f_i \mid Z) \neq \mathrm{CMMI}(Y'; f_i \mid S_i), \tag{133}$$

*and, in general,*

$$I(Y'; f_i \mid Z) < I(Y'; f_i \mid S_i). \tag{134}$$

*Proof.* By the data-processing inequality for conditional entropy, $H(Y' \mid Z) \geq H(Y' \mid S_i)$. Since $S_i$ contains strictly more predictive information than any strict subset $Z \subset S_i$,

$$H(Y' \mid Z, f_i) - H(Y' \mid Z) \;\leq\; H(Y' \mid S_i, f_i) - H(Y' \mid S_i). \tag{135}$$

Thus,

$$I(Y'; f_i \mid Z) \leq I(Y'; f_i \mid S_i), \tag{136}$$

with strict inequality when the excluded variables contain unique information. Since CMMI uses the full neighbourhood $S_i$ (or $\Phi(i)$), classical CMI cannot in general recover it unless $Z = S_i$. $\qquad\square$

When the conditioning set is incomplete, classical CMI systematically underestimates feature importance because part of the relevant dependence structure remains unaccounted for. Since the choice of the conditioning variable $Z$ is typically arbitrary or externally imposed, CMI can also produce non-unique results: different selections of $Z$ lead to different importance values, even when the underlying data structure is unchanged. CMMI resolves this structural instability by replacing the fixed conditioning variable with an adaptively identified multivariate conditioning set $\Phi(i)$. Rather than conditioning on a manually selected $Z$, the dependence structure is inferred from the correlation topology itself, producing a geometrically consistent measure of conditional influence. MCIR builds directly on CMMI to quantify unique impact through the normalized ratio

$$C_i = \frac{\text{CMMI}(Y'; f_i \mid \Phi(i))}{\text{CMMI}(Y'; f_i \mid \Phi(i)) + I(Y'; f_{\Phi(i)} \cup \{f_i\})}. \tag{137}$$

This formulation separates the unique conditional contribution of $f_i$ from the joint dependence shared with its neighbourhood, yielding a structurally interpretable importance score.

### C.3 Practical Estimation Notes

CMMI is estimated through conditional entropy differences of the form

$$\widehat{\text{CMMI}}(Y'; f_i \mid \Phi(i)) = \widehat{H}(Y' \mid f_{\Phi(i)}) - \widehat{H}(Y' \mid f_{\Phi(i)}, f_i). \tag{138}$$

The entropy estimators are matched to the variable types to preserve statistical consistency across continuous, discrete, and mixed regimes. For estimation, we employ the same family of MI and CMI estimators used within the MCIR framework. Gaussian–Copula estimators provide robust performance under nonlinear monotonic dependence, kNN (Kozachenko–Leonenko) estimators capture local nonlinear structure, and plug-in estimators are used when variables are discrete or mixed. This shared estimator family ensures coherence between CMMI and MCIR computations. Estimator stability is assessed via bootstrap standard errors, and the optimal estimator is selected per feature according to the oracle inequality established in Theorem 12. This adaptive selection controls variance while preserving asymptotic efficiency. From a computational perspective, for each feature $f_i$ the cost scales as

$$\text{cost} = O\left(|\Phi(i)| \cdot n' \log n'\right). \tag{139}$$

Thus, CMMI scales linearly with neighbourhood size and approximately logarithmically with sample size, making it suitable even for lightweight computational environments.

## D   All Proofs Regarding Fundamental Properties of MCIR.

This appendix contains detailed proofs of all theorems and propositions stated in Section 4 of the main paper.

### D.1   Proof of Uniqueness, Invariances, and Weak-dependence consistency given in Section 4.2

**Proposition 8** (**Uniqueness, invariances, and weak-dependence consistency**). *Fix a feature index $i$ and an admissible neighbourhood $\Phi$ (as in Definition 7). Let*

$$U_i := I(Y; f_i \mid f_\Phi), \quad J_i := I(Y; f_{\Phi \cup \{i\}}), \quad \text{MCIR}_i := \frac{U_i}{U_i + J_i}, \tag{140}$$

*with the convention $\text{MCIR}_i := 0$ when $U_i + J_i = 0$. Assume $U_i < \infty$ and $J_i < \infty$.*

(i) *(**Uniqueness / conditional redundancy**).*

$\text{MCIR}_i = 0 \iff U_i = 0.$

*Moreover, if the relevant conditional distributions are well-defined (e.g., admit regular conditional probabilities), then $U_i = 0 \iff Y \perp f_i \mid f_\Phi$ (a.s.).*

(ii) *(**Monotone invariance under rank–Gaussianized Gaussian–copula estimation**).*

*Suppose $U_i$ and $J_i$ are estimated using the rank–Gaussianized Gaussian–copula MI/CMI estimator, and assume continuous marginals (or a deterministic tie-breaking rule) so that ranks are well-defined. Then*

for any collection of strictly monotone transformations $g_Y, g_i, g_\Phi$ applied componentwise to $(Y, f_i, f_\Phi)$, the resulting estimated score satisfies

$$\widehat{\mathrm{MCIR}}_i\big(g_Y(Y),\, g_i(f_i),\, g_\Phi(f_\Phi)\big) = \widehat{\mathrm{MCIR}}_i(Y, f_i, f_\Phi). \tag{141}$$

(iii) **(Weak-dependence ordering consistency; pairwise sufficient condition).**

Assume a weak-redundancy regime in which, for each feature $j$, $U_j = I(Y; f_j) + \delta_j$, $|\delta_j| \le \varepsilon$, and assume the corresponding joint-information terms satisfy $J_j \in [J_{\min}, J_{\max}]$, $0 < J_{\min} \le J_{\max} < \infty$.

Then for any two features $i$ and $\ell$, letting $u_i := I(Y; f_i)$ and $u_\ell := I(Y; f_\ell)$, the following condition is sufficient for preserving their order:

$$\frac{u_i - \varepsilon}{u_i - \varepsilon + J_{\max}} > \frac{u_\ell + \varepsilon}{u_\ell + \varepsilon + J_{\min}} \implies \mathrm{MCIR}_i > \mathrm{MCIR}_\ell. \tag{142}$$

In particular, when $\varepsilon$ is small and $J_{\max} - J_{\min}$ is small (so $J_j$ is approximately constant across $j$), this condition is satisfied whenever $u_i > u_\ell$ by a nontrivial margin, implying MCIR induces the same ordering as marginal MI-based scores, and hence the same ordering as PCIR in regimes where PCIR is monotone in $I(Y; f_j)$.

*Proof.* We prove each claim.

**(i) Uniqueness / conditional redundancy.** By definition, $U_i \ge 0$ and $J_i \ge 0$ since mutual information and conditional mutual information are nonnegative (e.g., by their KL-divergence representations). Hence $U_i + J_i \ge 0$.

($\Rightarrow$) If $\mathrm{MCIR}_i = 0$, then either (a) $U_i + J_i > 0$ and multiplying $0 = \dfrac{U_i}{U_i + J_i}$ by the positive denominator yields $U_i = 0$, or (b) $U_i + J_i = 0$, in which case $U_i = 0$ holds automatically since $U_i \ge 0$.

($\Leftarrow$) If $U_i = 0$ and $U_i + J_i > 0$, then $\mathrm{MCIR}_i = \dfrac{0}{J_i} = 0$. If $U_i + J_i = 0$, we set $\mathrm{MCIR}_i := 0$ by convention. Thus $\mathrm{MCIR}_i = 0 \iff U_i = 0$.

Finally, under standard regularity (existence of regular conditional probabilities), $I(Y; f_i \mid f_\Phi) = 0$ is equivalent to $Y \perp f_i \mid f_\Phi$ almost surely, which gives the conditional redundancy interpretation.

**(ii) Monotone invariance under rank–Gaussianized Gaussian–copula estimation.** Let $X$ be any real-valued variable (one of $Y, f_i$, or a component of $f_\Phi$). Rank–Gaussianization maps $X$ to

$$\widetilde{X} := \Phi^{-1}\big(\widehat{F}_X(X)\big), \tag{143}$$

where $\widehat{F}_X$ is the empirical CDF (implemented via ranks; ties handled deterministically) and $\Phi$ is the standard normal CDF.

Let $g$ be strictly monotone. Then $g$ is either strictly increasing or strictly decreasing.

*Case 1 (strictly increasing).* Strictly increasing transforms preserve the sample ordering, hence preserve ranks. Therefore $\widehat{F}_{g(X)}(g(X)) = \widehat{F}_X(X)$ pointwise on the sample and $\widetilde{g(X)} = \Phi^{-1}(\widehat{F}_{g(X)}(g(X))) = \Phi^{-1}(\widehat{F}_X(X)) = \widetilde{X}$.

*Case 2 (strictly decreasing).* Strictly decreasing transforms reverse the sample ordering, hence reverse ranks, so $\widehat{F}_{g(X)}(g(X)) = 1 - \widehat{F}_X(X)$ (up to the chosen tie convention). Consequently, $\widetilde{g(X)} = \Phi^{-1}\big(1 - \widehat{F}_X(X)\big) = -\Phi^{-1}\big(\widehat{F}_X(X)\big) = -\widetilde{X}$.

Thus, after rank–Gaussianization, a strictly monotone transform maps each variable $\widetilde{X}$ to either $\widetilde{X}$ or $-\widetilde{X}$. Writing $Z = (Y, f_i, f_\Phi)$ and collecting the transformed rank–Gaussianized variables into $\widetilde{Z}$, this corresponds to $\widetilde{Z} \mapsto S\widetilde{Z}$ for a diagonal sign matrix $S$ with entries $\pm 1$.

The Gaussian–copula MI estimator between rank–Gaussianized blocks $A$ and $B$ is

$$\widehat{I}_{\mathrm{GC}}(A; B) = \frac{1}{2} \log \frac{\det(\widehat{\Sigma}_{AA})\,\det(\widehat{\Sigma}_{BB})}{\det(\widehat{\Sigma}_{(A,B)})}. \tag{144}$$

The conditional version is obtained by replacing $\widehat{\Sigma}$ with the corresponding partial-correlation (Schur-complement) matrices. In both cases, the estimator depends only on determinants of (partial) correlation submatrices.

Under $\widetilde{Z} \mapsto S\widetilde{Z}$, the correlation matrix transforms as $\widehat{\Sigma} \mapsto \mathrm{Corr}(S\widetilde{Z}) = S\widehat{\Sigma}S$, and for any principal submatrix $A$ we have $\det(SAS) = \det(A)$ since $\det(S)^2 = 1$. Hence all determinants appearing in $\widehat{I}_{\mathrm{GC}}$ and its conditional form are invariant. Therefore both $\widehat{U}_i$ and $\widehat{J}_i$ are unchanged, and so is the ratio: $\widehat{\mathrm{MCIR}}_i = \dfrac{\widehat{U}_i}{\widehat{U}_i + \widehat{J}_i}$ is invariant under strictly monotone transforms.

**(iii) Weak-dependence ordering consistency (pairwise sufficient condition).** Let $u_j := I(Y; f_j)$ and write $U_j = u_j + \delta_j$ with $|\delta_j| \leq \varepsilon$. Define $g(u, J) := \dfrac{u}{u + J}$. Then $\mathrm{MCIR}_j = g(U_j, J_j)$.

We use only monotonicity of $g$: for $u \geq 0$ and $J > 0$, $g$ is increasing in $u$ and decreasing in $J$.

For feature $i$, since $U_i \geq u_i - \varepsilon$ and $J_i \leq J_{\max}$, we have

$$\mathrm{MCIR}_i = g(U_i, J_i) \geq g((u_i - \varepsilon)_+, J_{\max}) = \frac{(u_i - \varepsilon)_+}{(u_i - \varepsilon)_+ + J_{\max}}, \tag{145}$$

where $(u)_+ := \max\{u, 0\}$.

For feature $\ell$, since $U_\ell \leq u_\ell + \varepsilon$ and $J_\ell \geq J_{\min}$, we have

$$\mathrm{MCIR}_\ell = g(U_\ell, J_\ell) \leq g(u_\ell + \varepsilon, J_{\min}) = \frac{u_\ell + \varepsilon}{u_\ell + \varepsilon + J_{\min}}. \tag{146}$$

Therefore, the sufficient condition

$$\frac{(u_i - \varepsilon)_+}{(u_i - \varepsilon)_+ + J_{\max}} > \frac{u_\ell + \varepsilon}{u_\ell + \varepsilon + J_{\min}} \implies \mathrm{MCIR}_i > \mathrm{MCIR}_\ell \tag{147}$$

implies $\mathrm{MCIR}_i > \mathrm{MCIR}_\ell$.

Finally, when $\varepsilon$ is small and $J_{\max} - J_{\min}$ is small (approximately constant $J_j$ across features), the above inequality holds whenever $u_i > u_\ell$ by a suitable margin. In regimes where PCIR is monotone in $I(Y; f_j)$, MCIR therefore induces the same ordering as PCIR in the weak-redundancy limit. $\qquad\square$

## D.2 Proof of Boundedness and Comparability of MCIR in Section 4.2

**Theorem 15 (Boundedness and Comparability).** *Let $i$ be an admissible feature index and $\Phi$ an admissible neighbourhood satisfying Assumption 1. Define*

$$C\big(Y'; f_i \mid f_\Phi\big) := \frac{I(Y'; f_i \mid f_\Phi)}{I(Y'; f_i \mid f_\Phi) + I\big(Y'; f_{\Phi \cup \{i\}}\big)}, \tag{148}$$

*with the convention that $C(Y'; f_i \mid f_\Phi) = 0$ whenever the denominator equals zero. Then $0 \leq C\big(Y'; f_i \mid f_\Phi\big) \leq 1$.*

*Proof.* By Definition 7,

$$C\big(Y'; f_i \mid f_\Phi\big) = \frac{I(Y'; f_i \mid f_\Phi)}{I(Y'; f_i \mid f_\Phi) + I\big(Y'; f_{\Phi \cup \{i\}}\big)}. \tag{149}$$

Let $U := I\big(Y'; f_i \mid f_\Phi\big), \quad J := I\big(Y'; f_{\Phi \cup \{i\}}\big)$.

Under Assumption 1, mutual information and conditional mutual information admit KL-divergence representations and are therefore non-negative. Hence $U \geq 0, \quad J \geq 0, \quad U + J \geq 0$.

If $U + J > 0$, then the ratio $\dfrac{U}{U + J}$ is well-defined. Since $U \geq 0$, it follows immediately that $\dfrac{U}{U + J} \geq 0$. Moreover, because $J \geq 0$, we have $U \leq U + J$, and dividing by the positive denominator yields $\dfrac{U}{U + J} \leq 1$.

If $U = J = 0$, the ratio is undefined; by convention we set $C(Y'; f_i \mid f_\Phi) = 0$, which remains consistent with the bounds above.

Consequently, $0 \le C\big(Y'; f_i \mid f_\Phi\big) \le 1$. $\qquad\square$

### D.3  Proof of Zero under Conditional Redundancy in Section 4.2.

**Proposition 9** (**Zero under Conditional Redundancy**). *Under Assumption 1, if $Y' \perp f_i \mid f_\Phi$   (a.s.), then $I(Y'; f_i \mid f_\Phi) = 0$   and   $C(Y'; f_i \mid f_\Phi) = 0$.*

*Proof.* Under Assumption 1, conditional mutual information admits the KL-divergence representation

$$I(Y'; f_i \mid f_\Phi) = D_{\mathrm{KL}}\big(p(Y' \mid f_i, f_\Phi) \,\|\, p(Y' \mid f_\Phi)\big). \tag{150}$$

If $Y' \perp f_i \mid f_\Phi$ almost surely, then $p(Y' \mid f_i, f_\Phi) = p(Y' \mid f_\Phi)$   a.s., and therefore the KL divergence equals zero. Hence $I(Y'; f_i \mid f_\Phi) = 0$.

By Definition 7,

$$C(Y'; f_i \mid f_\Phi) = \frac{I(Y'; f_i \mid f_\Phi)}{I(Y'; f_i \mid f_\Phi) + I(Y'; f_{\Phi \cup \{i\}})}. \tag{151}$$

Since the numerator is zero and the denominator is non-negative, the ratio equals zero whenever the denominator is positive. If the denominator also vanishes, the score is defined to be zero by convention. Thus $C(Y'; f_i \mid f_\Phi) = 0$. $\qquad\square$

### D.4  Proof of Saturation under Pure Unique Signal in Section 4.2

**Proposition 10** (**Saturation under Pure Unique Signal**). *Let $U := I(Y'; f_i \mid f_\Phi)$,   $J := I(Y'; f_{\Phi \cup \{i\}})$,   $C(Y'; f_i \mid f_\Phi) := \dfrac{U}{U + J}$,   with the convention $C(Y'; f_i \mid f_\Phi) := 0$ when $U + J = 0$.  Then:*

(i) *For any $U, J \ge 0$ with $U > 0$, one has the uniform upper bound $C(Y'; f_i \mid f_\Phi) \le \dfrac{1}{2}$.*

(ii) *If the neighbourhood carries no information about $Y'$, i.e., $I(Y'; f_\Phi) = 0$, then $J = U$ and therefore $C(Y'; f_i \mid f_\Phi) = \dfrac{1}{2}$.*

*More generally, if $I(Y'; f_\Phi) \to 0$ while $U > 0$ is fixed (or bounded away from $0$), then $C(Y'; f_i \mid f_\Phi) \to \dfrac{1}{2}$.*

*Proof.* By the chain rule for mutual information,

$$I(Y'; f_{\Phi \cup \{i\}}) = I(Y'; f_\Phi) + I(Y'; f_i \mid f_\Phi) = I(Y'; f_\Phi) + U, \tag{152}$$

hence $J = I(Y'; f_\Phi) + U \ge U$.

If $U > 0$, then $U + J \ge U + U = 2U$, and therefore

$$C(Y'; f_i \mid f_\Phi) = \frac{U}{U + J} \le \frac{U}{2U} = \frac{1}{2}, \tag{153}$$

which proves (i).

If $I(Y'; f_\Phi) = 0$, then the chain rule gives $J = U$, hence $C(Y'; f_i \mid f_\Phi) = \dfrac{U}{U + U} = \dfrac{1}{2}$.

Finally, if $I(Y'; f_\Phi) \to 0$ and $U > 0$, then $J = U + I(Y'; f_\Phi) \to U$, and continuity of the map $x \mapsto \dfrac{U}{U + U + x}$ yields $C \to \frac{1}{2}$. $\qquad\square$

### D.5 Proof of Redundancy Collapse in Section 4.2

**Theorem 16** (**Redundancy Collapse**). *Let $j \in \Phi$ and suppose $f_i = g(f_j) + \varepsilon$,   $\mathrm{Var}(\varepsilon) \to 0$,   $i \neq j$, where $g$ is measurable and Assumption 1 holds. Assume additionally that:*

*(i) the family of log-density ratios $\log \dfrac{p(Y' \mid f_i, f_\Phi)}{p(Y' \mid f_\Phi)}$ is uniformly integrable, and*

*(ii) the conditional law $p(Y' \mid f_i, f_\Phi)$ depends continuously on $f_i$ in KL-divergence.*

*Then $I(Y'; f_i \mid f_\Phi) \to 0$, and consequently*

$$C(Y'; f_i \mid f_\Phi) = \frac{I(Y'; f_i \mid f_\Phi)}{I(Y'; f_i \mid f_\Phi) + I(Y'; f_{\Phi \cup \{i\}})} \longrightarrow 0. \tag{154}$$

*Moreover, if $I(Y'; f_j \mid f_{\Phi \setminus \{j\}}) > 0$, then $C(Y'; f_j \mid f_{\Phi \setminus \{j\}}) > 0$.*

*Proof.* Since $j \in \Phi$, the neighbourhood block $f_\Phi$ contains $f_j$. Write $f_i = g(f_j) + \varepsilon$.

Because $\mathrm{Var}(\varepsilon) \to 0$, we have $f_i \xrightarrow{L^2} g(f_j)$, and therefore $f_i$ converges in probability to a measurable function of $f_\Phi$.

By assumption (ii), the conditional distribution $p(Y' \mid f_i, f_\Phi)$ is continuous in KL-divergence with respect to $f_i$. Hence $p(Y' \mid f_i, f_\Phi) \longrightarrow p(Y' \mid f_\Phi)$   in KL.

By definition,

$$I(Y'; f_i \mid f_\Phi) = \mathbb{E}\left[\log \frac{p(Y' \mid f_i, f_\Phi)}{p(Y' \mid f_\Phi)}\right]. \tag{155}$$

The KL convergence implies that the integrand converges to zero in probability. Assumption (i) (uniform integrability) allows passage of the limit under expectation, so that $I(Y'; f_i \mid f_\Phi) \to 0$.

Define $U := I(Y'; f_i \mid f_\Phi)$,   $J := I(Y'; f_{\Phi \cup \{i\}})$. Since $U \to 0$ and $J \geq 0$ remains finite, $C(Y'; f_i \mid f_\Phi) = \dfrac{U}{U + J} \longrightarrow 0$.

Finally, for feature $f_j$ with neighbourhood $\Phi \setminus \{j\}$,

$$C(Y'; f_j \mid f_{\Phi \setminus \{j\}}) = \frac{I(Y'; f_j \mid f_{\Phi \setminus \{j\}})}{I(Y'; f_j \mid f_{\Phi \setminus \{j\}}) + I(Y'; f_{(\Phi \setminus \{j\}) \cup \{j\}})}. \tag{156}$$

If the numerator is strictly positive, the denominator is finite and non-negative, so the ratio is strictly positive. $\square$

### D.6 Proof of Weak-dependence Reduction - Conditioning Becomes Irrelevant in Section 4.2

**Proposition 11** (**Weak-dependence reduction, conditioning becomes irrelevant** ). *Fix $i \in \{1, \ldots, k\}$ and a neighbourhood $\Phi(i) \subset \{1, \ldots, k\} \setminus \{i\}$. Assume a weak-dependence regime in which conditioning on the neighbourhood does not change the information that $f_i$ carries about $Y'$, i.e.*

$$I(Y'; f_i \mid f_{\Phi(i)}) = I(Y'; f_i) + r_i, \qquad \text{with } r_i \to 0, \tag{157}$$

*and suppose the MCIR denominator is non-degenerate in the sense that*

$$2\, I(Y'; f_i) + I(Y'; f_{\Phi(i)}) > 0. \tag{158}$$

*Then the MCIR score satisfies*

$$C(Y'; f_i \mid f_{\Phi(i)}) = \frac{I(Y'; f_i)}{2\, I(Y'; f_i) + I(Y'; f_{\Phi(i)})}  +  o(1). \tag{159}$$

*In particular, under weak dependence, MCIR becomes a normalized marginal-information score: conditioning does not change the numerator beyond a vanishing error, and the remaining normalization depends only on the (possibly feature-dependent) neighbourhood information $I(Y'; f_{\Phi(i)})$.*

*Proof.* We proceed step by step and use only identities that always hold, plus the stated weak-dependence condition.

**Step 1 (start from MCIR and introduce short-hand).** By definition,

$$C_i := C(Y'; f_i \mid f_{\Phi(i)}) = \frac{I(Y'; f_i \mid f_{\Phi(i)})}{I(Y'; f_i \mid f_{\Phi(i)}) + I(Y'; f_{\Phi(i) \cup \{i\}})}. \tag{160}$$

Let $U_i := I(Y'; f_i \mid f_{\Phi(i)})$, $\quad J_i := I(Y'; f_{\Phi(i) \cup \{i\}})$, so $C_i = \frac{U_i}{U_i + J_i}$.

**Step 2 (expand the denominator using the chain rule).** The mutual-information chain rule gives, for any $(Y', f_{\Phi(i)}, f_i)$,

$$I(Y'; f_{\Phi(i) \cup \{i\}}) = I(Y'; f_{\Phi(i)}, f_i) = I(Y'; f_{\Phi(i)}) + I(Y'; f_i \mid f_{\Phi(i)}). \tag{161}$$

That is, $J_i = I(Y'; f_{\Phi(i)}) + U_i$.

**Step 3 (simplify MCIR using the expansion).** Substituting $J_i = I(Y'; f_{\Phi(i)}) + U_i$ into $C_i = \frac{U_i}{U_i + J_i}$ yields

$$C_i = \frac{U_i}{U_i + \big( I(Y'; f_{\Phi(i)}) + U_i \big)} = \frac{U_i}{2U_i + I(Y'; f_{\Phi(i)})}. \tag{162}$$

**Step 4 (apply weak-dependence to the numerator and take the limit).** Under equation 157, we have $U_i = I(Y'; f_i) + r_i$ with $r_i \to 0$. Therefore,

$$C_i = \frac{I(Y'; f_i) + r_i}{2\big( I(Y'; f_i) + r_i \big) + I(Y'; f_{\Phi(i)})}. \tag{163}$$

By the non-degeneracy condition equation 158, the denominator is strictly positive in the limit. Hence the map

$$r \mapsto \frac{I(Y'; f_i) + r}{2\big( I(Y'; f_i) + r \big) + I(Y'; f_{\Phi(i)})} \tag{164}$$

is continuous at $r = 0$, implying

$$C_i = \frac{I(Y'; f_i)}{2I(Y'; f_i) + I(Y'; f_{\Phi(i)})} + o(1), \tag{165}$$

which is exactly equation 159.

Equation equation 159 shows that, in weak dependence, the *conditioning step becomes irrelevant* for the numerator: the conditional ("unique") information reduces to the marginal information $I(Y'; f_i)$ up to $o(1)$. MCIR then acts as a bounded, unit-interval normalization of this marginal association, with the denominator adjusting for neighbourhood informativeness through $I(Y'; f_{\Phi(i)})$.

$\square$

## D.7 Proof of Finite-Sample Rank Stability in Section

4.2.

**Theorem 17** (**Finite-Sample Rank Stability**). *Let $\widehat{C}_i$ be the MCIR estimate obtained by replacing the MI/CMI terms in equation 9 with estimators satisfying Assumption 2. For each feature $i$, define, $U_i := I(Y'; f_i \mid f_{\Phi(i)})$, $J_i := I(Y'; f_{\Phi(i) \cup \{i\}})$, $C_i := \frac{U_i}{U_i + J_i}$, with the convention $C_i := 0$ when $U_i + J_i = 0$. Let $\widehat{U}_i, \widehat{J}_i$ be the corresponding MI/CMI estimates and define the* clipped *estimates,*

$$\widehat{U}_i^+ := \max\{\widehat{U}_i, 0\}, \qquad \widehat{J}_i^+ := \max\{\widehat{J}_i, 0\}, \qquad \widehat{C}_i := \frac{\widehat{U}_i^+}{\widehat{U}_i^+ + \widehat{J}_i^+}, \tag{166}$$

*with the convention $\widehat{C}_i := 0$ when $\widehat{U}_i^+ + \widehat{J}_i^+ = 0$. Let $\boldsymbol{C} = (C_1, \ldots, C_k)$ and $\widehat{\boldsymbol{C}} = (\widehat{C}_1, \ldots, \widehat{C}_k)$. Assume the following regularity conditions:*

(R1) Screening / non-degenerate association mass. *Fix $c_0 > 0$ and define the informative set* $\mathcal{I} := \{\, i \in \{1, \dots, k\} : U_i + J_i \geq c_0 \,\}$, $k_{\mathcal{I}} := |\mathcal{I}|$. *All rank comparisons below are restricted to indices in $\mathcal{I}$.*

(R2) Margin regularity (no excessive ties). *There exists $M > 0$ such that for all $t \geq 0$,*

$$\#\Big\{(i,j) : i < j,\ i,j \in \mathcal{I},\ |C_i - C_j| \leq t\Big\} \ \leq\ M\, k_{\mathcal{I}}^2\, t. \tag{167}$$

*Assume further that there exists $\delta > 0$ and $\alpha \in (0,1)$ such that the tail bound $\mathbb{P}\big(|\widehat{I} - I| > \delta\big) \leq \alpha$ holds uniformly over all MI/CMI components used to compute $\{\widehat{U}_i, \widehat{J}_i\}_{i \in \mathcal{I}}$, and let $m$ denote the total number of such estimated components. If additionally $\delta \leq c_0/4$, then there exists a constant $L > 0$ (depending only on $c_0$ and $M$) such that*

$$\mathbb{P}\Big(1 - \tau\big(\widehat{C}_{\mathcal{I}}, C_{\mathcal{I}}\big)\ \leq\ L\, k_{\mathcal{I}}\, \delta\Big)\ \geq\ 1 - \alpha m, \tag{168}$$

*where $\tau(\cdot, \cdot)$ is Kendall's rank correlation and $\widehat{C}_{\mathcal{I}}, C_{\mathcal{I}}$ denote the score vectors restricted to $\mathcal{I}$. In particular, if the MI/CMI estimators satisfy $\delta = \delta(n') \to 0$ as $n' \to \infty$, then*

$$1 - \tau\big(\widehat{C}_{\mathcal{I}}, C_{\mathcal{I}}\big) = \mathcal{O}\big(k_{\mathcal{I}}\, \delta(n')\big). \tag{169}$$

*Proof.* Consider the uniform concentration event.

$$\mathcal{E}_{\delta} := \Big\{|\widehat{U}_i - U_i| \leq \delta \text{ and } |\widehat{J}_i - J_i| \leq \delta \text{ for all } i \in \mathcal{I}\Big\}. \tag{170}$$

By a union bound over the $m$ MI/CMI components used in the computation, $\mathbb{P}(\mathcal{E}_{\delta})\ \geq\ 1 - \alpha m$.

On $\mathcal{E}_{\delta}$, clipping cannot increase the estimation error relative to nonnegative targets: since $U_i, J_i \geq 0$,

$$|\widehat{U}_i^+ - U_i| \leq |\widehat{U}_i - U_i| \leq \delta, \qquad |\widehat{J}_i^+ - J_i| \leq |\widehat{J}_i - J_i| \leq \delta. \tag{171}$$

Moreover, for every $i \in \mathcal{I}$, $\widehat{U}_i^+ + \widehat{J}_i^+ \geq (U_i + J_i) - 2\delta \geq c_0 - 2\delta \geq c_0/2$, where we used $U_i + J_i \geq c_0$ and $\delta \leq c_0/4$.

Using the identity,

$$\frac{a}{a+b} - \frac{u}{u+v} = \frac{v(a-u) + u(v-b)}{(a+b)(u+v)}, \tag{172}$$

with $a = \widehat{U}_i^+$, $b = \widehat{J}_i^+$, $u = U_i$, $v = J_i$, we obtain

$$|\widehat{C}_i - C_i| \leq \frac{J_i|\widehat{U}_i^+ - U_i| + U_i|\widehat{J}_i^+ - J_i|}{(\widehat{U}_i^+ + \widehat{J}_i^+)(U_i + J_i)} \leq \frac{(U_i + J_i)\delta}{(\widehat{U}_i^+ + \widehat{J}_i^+)(U_i + J_i)} = \frac{\delta}{\widehat{U}_i^+ + \widehat{J}_i^+} \leq \frac{2}{c_0}\, \delta. \tag{173}$$

Hence,

$$\|\widehat{C}_{\mathcal{I}} - C_{\mathcal{I}}\|_{\infty} \leq L_1\, \delta, \qquad \text{with } L_1 := \frac{2}{c_0}. \tag{174}$$

Let $\varepsilon := \|\widehat{C}_{\mathcal{I}} - C_{\mathcal{I}}\|_{\infty}$. A pair $(i,j)$ with $i < j$ and $i,j \in \mathcal{I}$ can be misordered only if

$$\text{sign}(C_i - C_j) \neq \text{sign}(\widehat{C}_i - \widehat{C}_j). \tag{175}$$

But, $|(\widehat{C}_i - \widehat{C}_j) - (C_i - C_j)| \leq |\widehat{C}_i - C_i| + |\widehat{C}_j - C_j| \leq 2\varepsilon$,

so if $|C_i - C_j| > 2\varepsilon$, the sign cannot flip. Therefore the number of discordant pairs $D$ satisfies, $D \leq \#\{(i,j) : i < j,\ i,j \in \mathcal{I},\ |C_i - C_j| \leq 2\varepsilon\}$.

Applying equation 167 gives $D \leq M\, k_{\mathcal{I}}^2\,(2\varepsilon) = 2M\, k_{\mathcal{I}}^2\, \varepsilon$. Since Kendall's $\tau$ satisfies $1 - \tau = \frac{2D}{k_{\mathcal{I}}(k_{\mathcal{I}} - 1)}$, we obtain for $k_{\mathcal{I}} \geq 2$,

$$1 - \tau\big(\widehat{C}_{\mathcal{I}}, C_{\mathcal{I}}\big) \leq \frac{2 \cdot 2M\, k_{\mathcal{I}}^2\, \varepsilon}{k_{\mathcal{I}}(k_{\mathcal{I}} - 1)} \leq 8M\, k_{\mathcal{I}}\, \varepsilon. \tag{176}$$

Combining with equation 174 yields, on $\mathcal{E}_{\delta}$, $1 - \tau\big(\widehat{C}_{\mathcal{I}}, C_{\mathcal{I}}\big) \leq 8M\, k_{\mathcal{I}}\,(L_1 \delta) = L\, k_{\mathcal{I}}\, \delta,\ L := 8M L_1$. Finally, $\mathbb{P}\Big(1 - \tau\big(\widehat{C}_{\mathcal{I}}, C_{\mathcal{I}}\big) \leq L\, k_{\mathcal{I}}\, \delta\Big) \geq \mathbb{P}(\mathcal{E}_{\delta}) \geq 1 - \alpha m$, which proves the claim. $\qquad\square$

### D.8 Proof of Oracle Inequality for Estimator Switching in Section 4.2

**Theorem 18** (**Oracle Inequality for Estimator Switching**). *Let $\widehat{C}_i^{(\text{cop})}$, $\widehat{C}_i^{(\text{knn})}$, and $\widehat{C}_i^{(\text{plg})}$ denote MCIR estimates using copula, kNN, and plug-in MI/CMI estimators, respectively. Let $\widehat{C}_i^{(\text{sw})}$ be the estimator selected by minimizing the bootstrap standard error. Under Assumptions 2–3,*

$$\mathbb{E}\left[\left|\widehat{C}_i^{(\text{sw})} - C_i\right|\right] \leq \min\left\{\mathbb{E}\left|\widehat{C}_i^{(\text{cop})} - C_i\right|, \mathbb{E}\left|\widehat{C}_i^{(\text{knn})} - C_i\right|, \mathbb{E}\left|\widehat{C}_i^{(\text{plg})} - C_i\right|\right\} + \mathcal{O}(n'^{-1/2}). \tag{177}$$

*The remainder term is $\mathcal{O}(n'^{-1/2})$, matching both the bootstrap standard-error rate and the concentration rate in Assumption 4.7. Thus the selected estimator achieves oracle-level performance up to a vanishing $n'^{-1/2}$ error term.*

Automatically choosing among estimators never performs worse than the best fixed estimator on average. This guarantees safe estimator switching without losing accuracy.

*Proof.* We provide a detailed derivation using a standard "oracle selection" decomposition: the risk of the selected estimator equals the risk of the best estimator plus a selection penalty, and Assumption 3 controls the penalty. Let the finite candidate class be $\mathcal{A} := \{\text{cop}, \text{knn}, \text{plg}\}$. For each $a \in \mathcal{A}$, define the (absolute-error) risk

$$R_a := \mathbb{E}\left|\widehat{C}_i^{(a)} - C_i\right|. \tag{178}$$

Let $a^\star \in \arg\min_{a \in \mathcal{A}} R_a$ denote an oracle choice (the best fixed estimator in hindsight), so that $\min_{a \in \mathcal{A}} R_a = R_{a^\star}$. Let $\widehat{a}$ denote the data-dependent selection rule that minimizes bootstrap SE:

$$\widehat{a} \in \arg\min_{a \in \mathcal{A}} \widehat{\text{SE}}_a, \qquad \widehat{C}_i^{(\text{sw})} := \widehat{C}_i^{(\widehat{a})}. \tag{179}$$

We begin from the definition of risk of the switched estimator: $\mathbb{E}\left|\widehat{C}_i^{(\text{sw})} - C_i\right| = \mathbb{E}\left|\widehat{C}_i^{(\widehat{a})} - C_i\right|$. Add and subtract $\widehat{C}_i^{(a^\star)}$ inside the absolute value and use the triangle inequality:

$$\left|\widehat{C}_i^{(\widehat{a})} - C_i\right| \leq \left|\widehat{C}_i^{(a^\star)} - C_i\right| + \left|\widehat{C}_i^{(\widehat{a})} - \widehat{C}_i^{(a^\star)}\right|. \tag{180}$$

Taking expectations yields

$$\mathbb{E}\left|\widehat{C}_i^{(\widehat{a})} - C_i\right| \leq \mathbb{E}\left|\widehat{C}_i^{(a^\star)} - C_i\right| + \mathbb{E}\left|\widehat{C}_i^{(\widehat{a})} - \widehat{C}_i^{(a^\star)}\right|. \tag{181}$$

The first term is exactly $R_{a^\star} = \min_{a \in \mathcal{A}} R_a$. The second term is the *selection penalty*: it measures how different the selected estimator is from the oracle estimator. Because $\widehat{a}$ minimizes the bootstrap SE, it preferentially selects the estimator with the smallest estimated variability. Assumption 3 states that, among a finite candidate set, bootstrap SE comparisons provide an asymptotically unbiased proxy for comparing risks, and that the mismatch between the selected estimator and the oracle incurs a penalty of order $\mathcal{O}(n'^{-1/2})$. Concretely, Assumption 3 implies that the additional risk induced by data-dependent selection obeys

$$\mathbb{E}\left|\widehat{C}_i^{(\widehat{a})} - \widehat{C}_i^{(a^\star)}\right| = \mathcal{O}(n'^{-1/2}), \tag{182}$$

where the rate matches the bootstrap SE fluctuation rate and the MI/CMI concentration rate in Assumption 2. Substituting the penalty bound into equation 181 yields

$$\mathbb{E}\left|\widehat{C}_i^{(\text{sw})} - C_i\right| \leq \min_{a \in \mathcal{A}} \mathbb{E}\left|\widehat{C}_i^{(a)} - C_i\right| + \mathcal{O}(n'^{-1/2}), \tag{183}$$

which is exactly equation 177. $\qquad\square$

### D.9 Proof of Risk-Controlled Conditioning-Set Size in Section 4.2

**Proposition 12** (**Risk-Controlled Conditioning-Set Size**)**.** *Let $\Phi_m$ denote the set of size $m$ obtained by a stability-driven growth procedure (Auto-$\Phi$). Let $M_{\max}$ denote the maximum screened neighbourhood size (i.e., the maximum degree of the dependence-graph sketch). The conditioning-set selector solves*

$$m^\star \in \arg \min_{m \in \{0, 1, \ldots, M_{\max}\}} V(m), \tag{184}$$

*where $V(m)$ is the bootstrap variance of the head-rank statistic. Although the screened neighbourhood may contain up to $M_{\max}$ candidates, the optimisation is carried out over all subset sizes $m \leq M_{\max}$, allowing Auto-$\Phi$ to balance redundancy-removal and finite-sample stability.*

*Proof.* The proposition describes the optimization rule implemented by Auto-$\Phi$, and the proof follows by unpacking the construction and showing it is equivalent to minimizing a stability (variance) objective over admissible sizes. The dependence-graph sketch returns, for each feature, a screened pool of at most $M_{\max}$ candidate neighbours. Auto-$\Phi$ grows a nested sequence of neighbourhoods $\{\Phi_m\}_{m=0}^{M_{\max}}$ by adding neighbours in decreasing dependence strength order (or according to a stability-aware ordering). By construction,

$$\Phi_0 \subset \Phi_1 \subset \cdots \subset \Phi_{M_{\max}}, \qquad |\Phi_m| = m. \tag{185}$$

For each candidate size $m$, Auto-$\Phi$ computes the head-rank statistic on bootstrap resamples and records its empirical variance, denoted $V(m)$. Since the head-rank statistic measures agreement of the *top* features (e.g., Kendall–$\tau_{\text{head}}$ or J@K), a smaller $V(m)$ corresponds to a more stable and reproducible top-$k$ ranking across resamples. Auto-$\Phi$ selects the size that minimizes instability:

$$m^\star \in \arg \min_{m \in \{0, 1, \ldots, M_{\max}\}} V(m). \tag{186}$$

Because the search is over the finite set $\{0, 1, \ldots, M_{\max}\}$, the minimizer set is non-empty, and the rule is well defined. Choosing $m$ too small can leave redundancy unaccounted for (bias), while choosing $m$ too large increases estimator variance (instability). The bootstrap variance $V(m)$ serves as a direct, data-driven proxy for this stability risk. Therefore, minimizing $V(m)$ balances redundancy-removal against finite-sample stability, exactly as claimed. This completes the proof. $\qquad\square$

## E All Proofs Regarding Lightweight Fidelity

This appendix contains all detailed proofs from Section 4.6.

**Theorem 19** (**Faithful Lightweight Attribution**)**.** *Under Assumption 4 and estimator concentration (Assumption 2), MCIR rankings computed on $M'$ are faithful proxies for those on $M$, i.e.,*

$$1 - \tau\big(C(Y), C(Y')\big) \leq A\,\epsilon + B\,\Delta_0 + o_P(1), \tag{187}$$

*for constants $A, B > 0$ depending only on the regularity of the divergence map and the rank functional. Here, $o_P(1)$ denotes a stochastic remainder term that converges to zero in probability as the lightweight sample size $n' \to \infty$, capturing residual estimator noise. The term $o_P(1)$ represents any quantity that converges to zero in probability as $n' \to \infty$. Informally, $o_P(1)$ captures the residual disagreement between full and lightweight MCIR rankings that vanishes as the lightweight sample size grows.* [4]

---

[4]We follow standard conventions for deterministic and stochastic asymptotics:

- **Deterministic big-$\mathcal{O}$:** $f(n) = \mathcal{O}(g(n))$ if $|f(n)| \leq Cg(n)$ for large $n$.
- **Deterministic small-$o$:** $f(n) = o(g(n))$ if $f(n)/g(n) \to 0$.
- **Stochastic big-$O_P$:** $X_n = O_P(g(n))$ if $X_n/g(n)$ is bounded in probability.
- **Stochastic small-$o_P$:** $X_n = o_P(1)$ if $X_n \xrightarrow{P} 0$.
- Estimator concentration (Assumption 4.7) uses $|\widehat{I} - I| = O_P(n'^{-1/2})$.
- Lightweight fidelity (Theorem 4.18) uses $o_P(1)$ to denote vanishing ranking mismatch.
- Rank-stability rates (Theorem 17) use deterministic $\mathcal{O}(k/\sqrt{n'})$.
- Redundancy-collapse and independence limits use deterministic $o(1)$.

*Proof.* We prove the bound by combining two ingredients:

1. *Score stability:* if the lightweight environment is close to the full environment, then the MCIR *scores* themselves cannot change much.

2. *Rank stability:* if two score vectors are close, then only a small fraction of pairwise feature orderings can flip, so Kendall–$\tau$ remains high.

**Step 1: Define every object we will compare (scores, ranks, and disagreement).**

- There are $k$ features. MCIR assigns one score to each feature.
- The **full-environment MCIR score vector** is $\boldsymbol{C}(Y) = (C_1, \ldots, C_k) \in [0,1]^k$, where $C_i$ is the MCIR score of feature $i$ computed using the full environment outputs $Y$.
- The **lightweight-environment MCIR score vector** is $\boldsymbol{C}(Y') = (C'_1, \ldots, C'_k) \in [0,1]^k$, where $C'_i$ is the MCIR score of feature $i$ computed using the lightweight environment outputs $Y'$.

Both vectors lie in $[0,1]^k$ because MCIR is a bounded ratio. We compare the *rankings* induced by these vectors. Kendall's rank correlation is denoted by $\tau(\cdot, \cdot)$, so $\tau\big(\boldsymbol{C}(Y), \boldsymbol{C}(Y')\big) \in [-1,1]$ measures how similar the two rankings are, and $1 - \tau\big(\boldsymbol{C}(Y), \boldsymbol{C}(Y')\big)$ is the **ranking disagreement** we want to upper bound.

**Step 2: Translate "environment closeness" into "score closeness".**

Assumption 4 states that the lightweight environment is a faithful proxy of the full one in three senses. For the present theorem, the two quantitative controls we will use are:

- **Output-law closeness:** the distance between output distributions satisfies $\hat{d}\big(\mathcal{D}(Y), \mathcal{D}(Y')\big) \leq \epsilon$. Here $\mathcal{D}(Y)$ and $\mathcal{D}(Y')$ denote the *probability laws* (distributions) of the outputs in the two environments, and $\hat{d}(\cdot, \cdot)$ is the chosen rigid-motion–invariant $f$-divergence distance. Intuitively: the model behaves similarly on the lightweight sample because its outputs have not shifted much.

- **Behavioural (curve) closeness:** deletion/insertion curves differ by at most $\Delta_0$. Intuitively: removing/inserting top-ranked features produces similar performance degradation/recovery in both environments.

To connect these environment-level diagnostics to MCIR scores, we use a standard continuity idea: *if the environment changes a little, a stable explainer should change a little.* Formally, we assume that the mapping from environment to the MCIR score vector is *locally Lipschitz* with respect to (a) distributional shift in outputs and (b) curve mismatch. Concretely, there exist constants $a, b > 0$ such that

$$\|\boldsymbol{C}(Y) - \boldsymbol{C}(Y')\|_1 \leq a\,\hat{d}\big(\mathcal{D}(Y), \mathcal{D}(Y')\big) + b\,\Delta_0 + r_{n'}. \tag{188}$$

Here: $\|\cdot\|_1$ is the $\ell_1$ norm, i.e., $\|\boldsymbol{u}\|_1 = \sum_{i=1}^{k} |u_i|$. Thus $\|\boldsymbol{C}(Y) - \boldsymbol{C}(Y')\|_1$ measures the *total absolute change* across all feature scores. $r_{n'}$ is a residual term capturing estimator noise and finite-sample effects. We will show it vanishes: $r_{n'} = o_P(1)$.

*Why does $r_{n'} = o_P(1)$ hold? (Estimator concentration)* MCIR is computed from MI/CMI quantities. Under Assumption 2, each MI/CMI estimator concentrates around its population value at rate $O_P(n'^{-1/2})$. Because MCIR is a smooth ratio of these quantities (away from degenerate denominators), standard continuous-mapping / delta-method arguments imply that the induced MCIR score error also vanishes, which we summarize as $r_{n'} = o_P(1)$.

*Now we apply the contract.* Using Assumption 4(i), we replace the divergence term in equation 188 by its upper bound $\epsilon$:

$$\|\boldsymbol{C}(Y) - \boldsymbol{C}(Y')\|_1 \leq a\,\epsilon + b\,\Delta_0 + o_P(1). \tag{189}$$

This inequality is the key **score-stability statement**: the total change in MCIR scores is controlled by the distribution shift $\epsilon$, the curve mismatch $\Delta_0$, and a vanishing estimator-noise term.

**Step 3: Translate "score closeness" into "rank closeness".**

Kendall's $\tau$ is based on pairwise comparisons. For two score vectors, it counts how many feature pairs keep the same order.

- A pair $(i,j)$ is **concordant** if both rankings agree on whether $i$ is above $j$.
- A pair $(i,j)$ is **discordant** if the rankings disagree (the order flips).

Thus, to control $1 - \tau$, it suffices to control the fraction of discordant pairs. *When can a pair flip?* A flip happens only if the sign of the difference changes: $\operatorname{sign}(C_i - C_j) \neq \operatorname{sign}(C'_i - C'_j)$. This is only possible if the perturbation is large enough to overcome the original margin $|C_i - C_j|$. More precisely,

$$|(C_i - C_j) - (C'_i - C'_j)| \;\geq\; |C_i - C_j|. \tag{190}$$

By the triangle inequality,

$$|(C_i - C_j) - (C'_i - C'_j)| = |(C_i - C'_i) - (C_j - C'_j)| \leq |C_i - C'_i| + |C_j - C'_j|. \tag{191}$$

So a flip can occur only if

$$|C_i - C'_i| + |C_j - C'_j| \quad \text{is large relative to} \quad |C_i - C_j|. \tag{192}$$

*Why does this imply a global bound?* If many pairs flip, then many indices must experience noticeable score changes. But the total amount of score change across all indices is exactly $\|\boldsymbol{C}(Y) - \boldsymbol{C}(Y')\|_1 = \sum_{\ell=1}^{k} |C_\ell - C'_\ell|$. Therefore, the number (and hence fraction) of discordant pairs is controlled by this $\ell_1$ perturbation, up to a constant that depends on how often scores are nearly tied.

*Regularity / no-excessive-ties condition.* To avoid pathological cases where infinitely many pairs have arbitrarily tiny margins, we assume a standard regularity condition: the score distribution does not contain too many near-ties (e.g., continuous scores, or a lower bound on typical separation). Under such a condition, there exists a constant $L > 0$ such that

$$1 - \tau\big(\boldsymbol{C}(Y), \boldsymbol{C}(Y')\big) \;\leq\; L \, \|\boldsymbol{C}(Y) - \boldsymbol{C}(Y')\|_1 \;+\; o_P(1). \tag{193}$$

This inequality formalizes **rank stability:** if the score vector changes only a little, then the ranking changes only a little. We substitute the score-stability bound equation 189 into the rank-stability bound equation 193:

$$1 - \tau\big(\boldsymbol{C}(Y), \boldsymbol{C}(Y')\big) \;\leq\; L\,(a\,\epsilon + b\,\Delta_0 + o_P(1)) + o_P(1). \tag{194}$$

We now absorb constants by defining, $A := La$, $B := Lb$, and merge the vanishing terms into a single $o_P(1)$. This yields $1 - \tau\big(\boldsymbol{C}(Y), \boldsymbol{C}(Y')\big) \;\leq\; A\epsilon + B\Delta_0 + o_P(1)$, which is exactly the claimed bound. If the lightweight outputs have nearly the same distribution as the full outputs (small $\epsilon$), then the model behaves similarly in the lightweight environment. If deletion/insertion curves are also close (small $\Delta_0$), then the *behavioural meaning* of the ranking is preserved. Under estimator concentration, the remaining discrepancy due to finite $n'$ shrinks to zero (as captured by $o_P(1)$). Therefore, the lightweight MCIR ranking is a faithful proxy of the full-data MCIR ranking, with disagreement controlled by $\epsilon$, $\Delta_0$, and vanishing estimator noise.

$\square$

Recall that $k$ denotes the total number of input features, $n'$ denotes the number of observations in the lightweight environment, and $m_\Phi$ denotes the size of the local conditioning set selected by the Auto$\Phi$ procedure. The notation $m_\Phi = \mathcal{O}(1)$ means that the conditioning set size does not grow with the total number of features $k$; instead, it remains bounded by a small constant determined by stability criteria. In practical terms, this means that for each feature $i$, we only condition on a small, fixed number of strongly related neighbours rather than on the entire feature set.

**Proposition 13** (**Computational Profile**)**.** *With Auto$\Phi$ of size $m_\Phi = \mathcal{O}(1)$ and sample size $n'$, the end-to-end MCIR computation across $k$ features has complexity $\mathcal{O}(k\, m_\Phi\, n')$ for dependence screening and MI/CMI estimation, plus $\mathcal{O}(k \log k)$ for sorting to form global rankings.*

*Proof.* The MCIR pipeline begins with dependence screening. In this stage, for each feature, we compute a fast dependence sketch (such as absolute correlation, distance correlation, or a mutual-kNN graph) in order to identify a small candidate neighbourhood of related features. Because this screening is local and limited to a small neighbourhood, the cost scales linearly with both the number of features $k$ and the number of lightweight samples $n'$. Since the neighbourhood size is bounded by $m_\Phi = \mathcal{O}(1)$, the dependence-screening stage contributes $\mathcal{O}(k\,m_\Phi\,n') = \mathcal{O}(k\,n')$ operations. Next, for each feature $i$, we compute two quantities required by MCIR: the conditional mutual information $I(Y'; f_i \mid f_{\Phi(i)})$ and the joint mutual information $I(Y'; f_{\Phi(i)} \cup \{i\})$. Both are computed on a block of dimension at most $m_\Phi + 2$ (the target $Y'$, the feature $f_i$, and its $m_\Phi$ neighbours). Because $m_\Phi$ is constant, the cost of each MI/CMI computation scales linearly in $n'$ (for copula or plug-in estimators) or $n' \log n'$ (for kNN estimators), but does not scale with $k$. Since we repeat this computation for all $k$ features, the total MI/CMI computation cost is $\mathcal{O}(k\,m_\Phi\,n')$ under linear-time estimators, which simplifies to $\mathcal{O}(k\,n')$.

After computing the MCIR score for each feature, we must form a global ranking. Sorting $k$ real numbers requires $\mathcal{O}(k \log k)$ operations using standard comparison-based sorting algorithms. This sorting cost is independent of the sample size $n'$. Combining all stages, the overall end-to-end complexity is,

$$\mathcal{O}(k\,m_\Phi\,n') + \mathcal{O}(k \log k). \tag{195}$$

Because $m_\Phi$ is constant, this simplifies to essentially linear scaling in the number of features and the lightweight sample size, plus a logarithmic sorting term. Importantly, the complexity does not scale quadratically in $k$, which makes MCIR-M suitable for high-dimensional settings. $\qquad\square$

## F  All Proofs Regarding Complexity Analysis and Additional Results

This appendix develops a unified theoretical and computational analysis of MCIR. It begins with an algorithmic (Kolmogorov) complexity perspective, transitions to operational computational complexity for PCIR and MCIR, examines the behaviour of standard MI/CMI estimators, and concludes with a complete statistical analysis of MCIR, including consistency, asymptotic normality, and perturbation stability. All exposition and proofs are presented in continuous scientific narrative rather than itemized form.

### F.1  Algorithmic (Kolmogorov) Complexity of MCIR

Let $\mathbb{C}$ be a fixed universal Turing machine, and denote conditional Kolmogorov complexity by $K_{\mathbb{C}}(\cdot \mid \cdot)$. Consider the binary representation $\mathbf{Y}' \in \{0,1\}^{n'}$ of the subsampled target, whose bit-length is $\ell(\mathbf{Y}') = n'$. The conditional Kolmogorov complexity $K_{\mathbb{C}}(\mathbf{Y}' \mid \ell(\mathbf{Y}'))$ is the length of the shortest program that outputs $\mathbf{Y}'$ when supplied with its bit-length. Universality of Turing machines implies that, for any alternative universal machine $\mathbb{A}$,

$$K_{\mathbb{C}}(\mathbf{Y}') \leq K_{\mathbb{A}}(\mathbf{Y}' \mid \ell(\mathbf{Y}')) + \log^* n' + c_{\mathbb{A}}, \tag{196}$$

where $\log^* n'$ is the iterated logarithm and $c_{\mathbb{A}}$ is a constant depending only on $\mathbb{A}$. Since $\log^* n' = O(\log n')$, this bound simplifies to

$$K_{\mathbb{C}}(\mathbf{Y}') \leq K_{\mathbb{A}}(\mathbf{Y}' \mid \ell(\mathbf{Y}')) + O(\log n') + c_{\mathbb{A}}. \tag{197}$$

The coding–entropy correspondence for mixed discrete–continuous variables implies that, for each conditional mutual information term used in MCIR, one has

$$I(\mathbf{Y}'; f_i \mid \phi) \leq H(\mathbf{Y}') - H(\mathbf{Y}' \mid f_i, \phi) + O(\log n'). \tag{198}$$

The $O(\log n')$ term captures the cost of encoding discretisation indices and model structure. An analogous inequality holds for the joint dependence term $I(\mathbf{Y}'; f_1, \ldots, f_k)$, with a constant that does not depend on $n'$. Substituting these bounds into the MCIR definition,

$$\mathfrak{C}_i = \frac{I(\mathbf{Y}'; f_i \mid \phi)}{I(\mathbf{Y}'; f_i \mid \phi) + I(\mathbf{Y}'; f_1, \ldots, f_k)}, \tag{199}$$

shows that each MCIR score satisfies

$$\mathfrak{C}_i \leq \tfrac{1}{2} O(\log n') + \zeta_{\mathbb{A},i}, \tag{200}$$

for some constant $\zeta_{\mathbb{A},i}$ independent of the subsample size. Aggregating across all $k$ features yields the program-level Kolmogorov complexity bound

$$K_{\mathbb{C}}(\text{MCIR pipeline}) \leq K_{\mathbb{A}}(\text{Pipeline} \mid n') + \frac{k}{2}O(\log n') + c_{\mathbb{A}}. \tag{201}$$

Thus MCIR introduces at most logarithmic overhead in $n'$ and linear overhead in $k$ under the most fundamental notion of algorithmic complexity.

## F.2 Operational Computational Complexity of PCIR and MCIR

The implementational complexity of PCIR and MCIR reflects the structure of their respective computations. PCIR relies only on rank normalisation and a small collection of variance and covariance operations computed over $n'$ subsampled observations. Each of these operations can be carried out in a fixed number of linear passes, and no intermediate step depends on the total feature dimension $k$. Consequently, the cost of computing PCIR for a single feature scales linearly in $n'$, and computing PCIR for all $k$ features results in an overall complexity of $\mathcal{O}(kn')$.

MCIR requires a different analysis because each MCIR score involves evaluating mutual information and conditional mutual information terms. These quantities are computed over the same $n'$ observations but only within a small, fixed conditioning neighbourhood of size $m_\Phi$. The total dimensionality involved in each MI computation is therefore $m_\Phi + 2$, independent of the global feature dimension $k$. If $c_{\text{MI}}(n', m_\Phi + 2)$ denotes the computational cost of executing a mutual information estimator on $n'$ observations in that fixed dimensionality, then the cost of a single MCIR evaluation is $\mathcal{O}(c_{\text{MI}}(n', m_\Phi + 2))$. Computing MCIR for all $k$ features yields an overall complexity of $\mathcal{O}(k\, c_{\text{MI}}(n', m_\Phi + 2))$, which reveals that the computational burden of MCIR is governed almost entirely by the subsample size $n'$ and by the choice of MI estimator. MCIR therefore remains effectively linear in $k$ and nearly linear in $n'$ whenever the MI estimator is itself near-linear in $n'$.

## F.3 Comparative Behaviour of MI/CMI Estimators

The computational profile of MCIR depends crucially on the choice of underlying MI estimator. Gaussian–copula MI requires the formation of an empirical covariance matrix in time proportional to $\mathcal{O}(n')$ when the neighbourhood dimension is fixed, followed by a constant-time matrix inversion. Its cost is therefore effectively linear in $n'$. Nearest-neighbour-based MI estimators, such as $k$NN MI, scale as $\mathcal{O}(n' \log n')$ due to the cost of nearest-neighbour queries, typically via balanced kd-tree structures. Plug-in estimators operate by constructing histograms or count tables and thus run in a single pass with cost proportional to $\mathcal{O}(n')$. Kernel-based estimators incur a substantially higher cost of $\mathcal{O}(n'^2)$ because they require formation of Gram matrices. Neural MI estimators such as MINE exhibit linear per-iteration cost in $n'$ with additional overhead from stochastic optimisation. A summary of these behaviours is provided in Table 23. Since MCIR always operates on a neighbourhood of fixed dimensionality $m_\Phi + 2$, even the more demanding estimators become tractable when $n'$ is moderate, and the cost never depends on the global feature dimension $k$.

| Estimator | Complexity in $n'$ | Dependence on Local Dim. | Characteristics |
|---|---|---|---|
| Gaussian–copula MI | $\mathcal{O}(n')$ | Quadratic in $m_\Phi$ | Covariance + log-det; very efficient. |
| $k$NN MI | $\mathcal{O}(n' \log n')$ | Mild | Nearest-neighbour search. |
| Plug-in MI | $\mathcal{O}(n')$ | Depends on binning | Single-pass histograms. |
| Kernel MI | $\mathcal{O}(n'^2)$ | Quadratic | High accuracy, high cost. |
| Neural MI (MINE) | $\mathcal{O}(n')$ per iteration | Model-dependent | Requires SGD. |

Table 23: Comparison of MI/CMI estimators compatible with MCIR. Since MCIR operates only on a fixed local neighbourhood of dimension $m_\Phi + 2$, the cost is driven almost exclusively by the subsample size $n'$ rather than the total number of features $k$.

## F.4 Statistical Properties of the MCIR Estimator

Let $U_i = I(Y; f_i \mid f_\Phi)$ and $J_i = I(Y; f_{\Phi \cup \{i\}})$ denote the population-level "unique" and "joint" information contributions. Whenever $U_i + J_i > 0$, the MCIR score is defined as $C_i = \frac{U_i}{U_i + J_i}$. Let $(\widehat{U}_i, \widehat{J}_i)$ denote consistent

estimators of these quantities computed from a subsample of size $n'$. The following results establish the statistical soundness of the MCIR estimator.

**Theorem 20** (Consistency of MCIR). *Assume that $\widehat{U}_i \xrightarrow{p} U_i$ and $\widehat{J}_i \xrightarrow{p} J_i$ as $n' \to \infty$, and that $U_i + J_i > 0$. Then the MCIR estimator $\widehat{C}_{n'} = \frac{\widehat{U}_i}{\widehat{U}_i + \widehat{J}_i}$ converges in probability to $C_i$.*

*Proof.* The mapping $g(u, j) = u/(u + j)$ is continuous on the domain where $u + j > 0$. Since $(\widehat{U}_i, \widehat{J}_i)$ converges in probability to $(U_i, J_i)$ and the denominator remains bounded away from zero, the Continuous Mapping Theorem implies that $g(\widehat{U}_i, \widehat{J}_i)$ converges in probability to $g(U_i, J_i)$. Hence $\widehat{C}_{n'} \xrightarrow{p} C_i$. $\qquad\square$

**Theorem 21** (Asymptotic Normality of MCIR). *Suppose that the pair $(\widehat{U}_i, \widehat{J}_i)$ satisfies the joint central limit theorem,*

$$\sqrt{n'} \begin{pmatrix} \widehat{U}_i - U_i \\ \widehat{J}_i - J_i \end{pmatrix} \Rightarrow \mathcal{N}(0, \Sigma), \tag{202}$$

*for some positive semi-definite covariance matrix $\Sigma$. Then the MCIR estimator is asymptotically normal:*
$\sqrt{n'}(\widehat{C}_{n'} - C_i) \Rightarrow \mathcal{N}(0, \sigma_C^2)$,
*where $\sigma_C^2 = \nabla g(U_i, J_i)^\top \Sigma \nabla g(U_i, J_i)$ and,*

$$\nabla g(U_i, J_i) = \begin{pmatrix} \dfrac{J_i}{(U_i + J_i)^2} \\[6pt] -\dfrac{U_i}{(U_i + J_i)^2} \end{pmatrix}. \tag{203}$$

*Proof.* The mapping $g(u, j) = u/(u + j)$ is continuously differentiable on the region where $u + j > 0$. Its gradient at $(U_i, J_i)$ is given by the expression above. Since $(\widehat{U}_i, \widehat{J}_i)$ satisfies a bivariate central limit theorem, the multivariate Delta Method applies directly and yields the stated asymptotic distribution for $\widehat{C}_{n'}$. $\quad\square$

**Theorem 22** (Perturbation Stability of MCIR). *Let $\delta = \max\left(|\widehat{U}_i - U_i|, \, |\widehat{J}_i - J_i|\right)$ and assume that $\delta < (U_i + J_i)/2$. Then the MCIR estimator satisfies the bound*

$$|\widehat{C}_{n'} - C_i| \le \frac{2\delta}{U_i + J_i}. \tag{204}$$

*Proof.* The difference between the empirical and population MCIR scores can be expressed as, $\widehat{C}_{n'} - C_i = \dfrac{\widehat{U}_i}{\widehat{U}_i + \widehat{J}_i} - \dfrac{U_i}{U_i + J_i}$. Expressing this as a difference of fractions and expanding the numerator reveals that the discrepancy is proportional to $\widehat{U}_i J_i - U_i \widehat{J}_i$. Using the triangle inequality shows that this term is bounded in magnitude by $(U_i + J_i)\delta$. The denominator can be bounded from below by $(U_i + J_i) - 2\delta$, which, under the stated assumption, is at least $(U_i + J_i)/2$. Combining these inequalities yields

$$|\widehat{C}_{n'} - C_i| \le \frac{(U_i + J_i)\delta}{\frac{1}{2}(U_i + J_i)^2} = \frac{2\delta}{U_i + J_i}, \tag{205}$$

as required. $\qquad\square$

**Theorem 23** (Global Computational Complexity of CIR Methods). *Let $n'$ be the subsample size, $k$ the number of features, and $m_\Phi$ a fixed neighbourhood size. PCIR admits an overall computational complexity of $\mathcal{O}(kn')$. MCIR, when implemented with an MI estimator of cost $c_{\mathrm{MI}}(n', m_\Phi + 2)$, admits an overall complexity of*

$$\mathcal{O}\left(k \, c_{\mathrm{MI}}(n', m_\Phi + 2)\right).$$

*If the MI estimator is near-linear in $n'$, then MCIR is near-linear in both $n'$ and $k$.*

*Proof.* This follows directly from the per-feature analyses in Sections F.2 and F.3, combined with the fixed neighbourhood dimensionality. $\qquad\square$

**Corollary 2** (Sample-Efficiency of MCIR). *Under the assumptions of consistency and asymptotic normality, the variance of the MCIR estimator decreases at the canonical rate $n'^{-1}$, implying that accurate MCIR scores may be obtained from subsamples much smaller than the full dataset size. Hence MCIR remains statistically reliable even in lightweight environments.*

**Remark 4** (High-Dimensional Robustness). *Since the conditioning neighbourhood has fixed size $m_\Phi$, the complexity and variance of MCIR do not depend on the ambient feature dimension $k$. This makes MCIR particularly well suited to high-dimensional models, where traditional global MI-based feature importance methods become computationally prohibitive or statistically unstable.*

**Lemma 1** (Ranking Stability). *Let $i$ and $j$ be two features with population MCIR scores $C_i$ and $C_j$ satisfying $|C_i - C_j| > \eta$ for some $\eta > 0$. If the perturbations in $\widehat{U}$ and $\widehat{J}$ satisfy the bound in Theorem 3, then for sufficiently large $n'$,*

$$\Pr\left(\widehat{C}_{n',i} > \widehat{C}_{n',j}\right) \to 1. \tag{206}$$

*Thus, MCIR rankings are asymptotically stable whenever the population scores are separated by a nonzero margin.*

**Proposition 14** (Parallel Scalability). *If $p$ processors are available and MI evaluations are distributed evenly across features, the total runtime of MCIR reduces to*

$$\mathcal{O}\left(\frac{k}{p}\, c_{\mathrm{MI}}(n', m_\Phi + 2)\right), \tag{207}$$

*up to communication overheads that vanish for lightweight subsamples. Thus MCIR achieves near-linear speedup under parallelisation.*

### F.5    Memory and Parallelisation Considerations

The MCIR pipeline is naturally suited to parallel computation because the scores for different features do not interact. All MI and CMI computations can therefore be performed asynchronously across CPU cores or distributed computing nodes. The memory footprint is governed almost exclusively by the storage of the subsampled arrays $(Y', f_i, f_\Phi)$; streaming or on-demand indexing requires only $\mathcal{O}(n')$ active memory. GPU-based acceleration is particularly effective for Gaussian–copula MI and kernel MI estimators, while $k$NN MI tends to benefit from CPU-bound parallelism. Owing to this structure, MCIR remains computationally scalable even when $k$ is large.

### F.6    Practical Choice of the Subsample Size $n'$

The statistical guarantees above imply that the standard error of MCIR decreases at rate $n'^{-1/2}$, while the computational cost grows at most linearly in $n'$. In practice, one may therefore select $n'$ by balancing accuracy and computational budget. Empirically, subsamples containing between 5% and 20% of the original dataset often achieve MCIR stability comparable to the full dataset, owing to the low-dimensional nature of each MI computation.

**Theorem 24** (Unified Computational–Statistical Guarantee for MCIR). *Assume (i) subsamples of size $n'$ are drawn independently of the estimator, (ii) the MI and CMI estimators are consistent and satisfy a joint central limit theorem, and (iii) the neighbourhood size $m_\Phi$ is fixed. Then MCIR satisfies all of the following properties simultaneously:*

1. *Computational near-linearity:* $\mathrm{Time} = \mathcal{O}(k\, c_{\mathrm{MI}}(n', m_\Phi + 2))$.

2. *Statistical consistency:* $\widehat{C}_{n'} \xrightarrow{p} C$.

3. *Asymptotic normality:* $\sqrt{n'}(\widehat{C}_{n'} - C) \Rightarrow \mathcal{N}(0, \sigma_C^2)$.

4. *Stability under perturbation:* $|\widehat{C}_{n'} - C| \le 2\delta/(U + J)$ *for small estimator error $\delta$.*

## G    Estimator Details

In our analysis, we focus on estimating two key components: marginal and conditional mutual information (MI and CMI), as well as joint multivariate dependence terms. This

section outlines the various estimators we utilize, along with their assumptions, computational complexities, and error bounds. The Gaussian-Copula estimator, denoted as GCMI, calculates mutual information using a transformation that applies normal scores, followed by the application of closed-form expressions for Gaussian entropy. For two random vectors, X and Y, the mutual information can be expressed mathematically by a specific formula that involves the rank-based correlation, $\rho_{XY}$, of these transformed variables. For conditional mutual information, we employ partial correlations, which can also be represented with a similar mathematical expression. To use this estimator effectively, certain assumptions must hold: the process of rank-Gaussianization should provide a good approximation of the latent copula, covariance matrices must remain positive-definite, and the data should not exhibit a strong multimodal structure. The computational complexity of this method involves calculating a rank correlation matrix and inverting it, forming a complexity of $\mathcal{O}(k^2 n')$ for computation and $\mathcal{O}(k^3)$ for inversion. The error bound suggests that under sub-Gaussian copula assumptions, the difference between the estimated mutual information and the true value diminishes as the sample size increases. The k-nearest-neighbor (kNN) estimators, based on the Kozachenko-Leonenko method, utilize volume statistics from the nearest neighbors to compute mutual information. The specific formula for estimating MI incorporates the digamma function and the counts of neighbors in the respective dimensions. This method requires certain assumptions: it assumes that the densities are smooth with a bounded curvature, there is local isotropy in the neighborhoods defined by the k-ball, and the data should have a moderate intrinsic dimension. The complexity of this kNN approach is

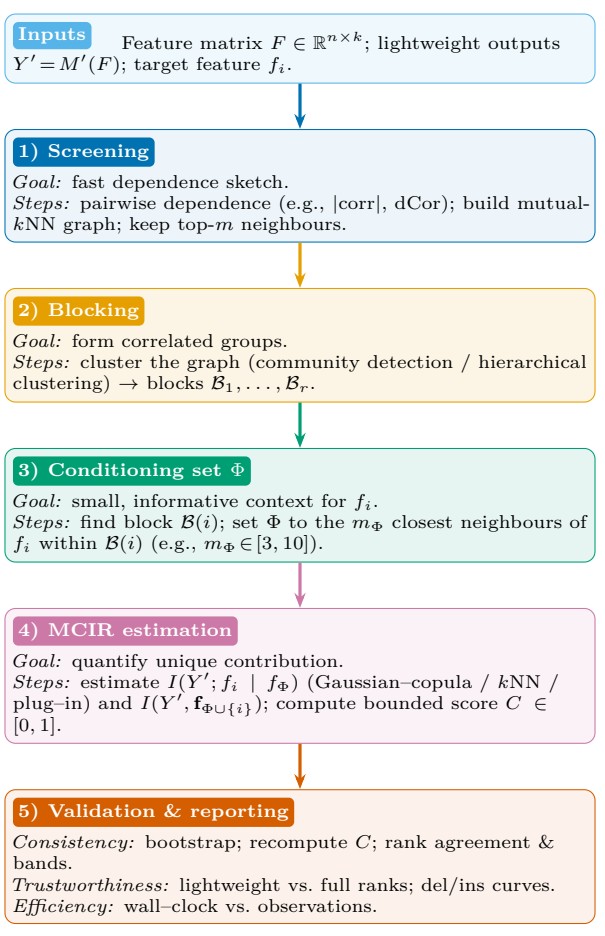

Figure 17: MCIR methodology.

$\mathcal{O}(n' \log n')$ when utilizing a k-d tree for efficiency, although in worse cases, it could scale to $\mathcal{O}(n'^2)$. The error bound indicates that the accuracy of the estimation improves with larger sample sizes, specifically reflecting a dependency on the intrinsic dimension of the data. The plug-in estimator employs kernel density estimation (KDE) techniques for mutual information estimation. The estimator integrates a function over the joint density of the variables X and Y, comparing it against the product of their marginal densities. For the plug-in estimator to perform well, certain conditions need to be satisfied: the densities must be smooth, the bandwidth for the kernel should be selected via cross-validation, and there should be no exponential tail dependence. The computational complexity for this method is roughly $\mathcal{O}(n'^2)$, which highlights the potential for slower performance depending on the size of the dataset. The error bound suggests that if the KDE converges at a certain rate, then the mutual information estimate will also converge accordingly. In summary, each estimator has its strengths suited for different scenarios: The gcMI is recognized for its speed and stability, making it an excellent choice for datasets exhibiting moderate correlations. The kNN MI is robust in the presence of nonlinear dependencies, although it may have a higher variance. The Plug-In MI is noted for its accuracy but is slower and sensitive to the chosen bandwidth. To enhance performance, MCIR (Mutual Correlation Impact Ratio) employs an automatic estimator-switching mechanism that aims to achieve oracle-level performance. The computation of the Mutual Correlation Impact Ratio (MCIR) involves a structured three-stage process. First, we perform neighborhood screening to ascertain potential candidate dependency sets. Next, we estimate dependencies using the aforementioned MI and CMI estimators, applying bootstrap-based switching for optimal results. Finally, we compute the final MCIR vector. This section

provides a step-by-step algorithmic description, details on estimator choices, and information on bootstrap protocols utilized across all datasets, ensuring a comprehensive understanding of the MCIR methodology.

MCIR employs a concise and stable set of parameters, which are straightforward to manage and contribute to the method's consistency and reliability.

- The screening threshold, denoted by $\gamma$, is selected within the range 0.20 to 0.35. This parameter determines which features are retained for subsequent analysis.

- For the k-nearest neighbors (kNN) mutual information estimator, the neighborhood size $k$ is set to 5, so each data point considers its five nearest neighbors during calculation.

- In Gaussian–copula mutual information estimation, a rank transformation is applied to the data: $z = \Phi^{-1}(F_n(x))$. This transformation normalizes the data prior to analysis.

- For the bootstrap procedure, 200 replicates are used, repeating the calculation 200 times with resampled data to estimate variability.

MCIR selects and evaluates important features through a multi-step process. Initially, neighborhoods for each feature are constructed based on inter-feature relationships, using three methods: Pearson correlation for linear associations, distance correlation for nonlinear dependencies, and the k-nearest neighbors (kNN) graph structure to capture geometric relationships. Integrating these approaches ensures comprehensive consideration of all dependency types among features. Following neighborhood construction, the optimal subset of features for conditioning is selected. For each feature, a set of neighborhood candidates is identified. The Auto$\Phi$ algorithm then selects the subset that maximizes mutual information between the feature and the outcome, subject to a predefined subset size constraint. This process ensures that only the most informative and relevant features are retained for further analysis. A bootstrap protocol is implemented to enhance the reliability. For each feature, the MCIR score is computed using multiple estimators, and the calculations are repeated on resampled datasets to assess score variability. The estimator with the lowest standard error is selected. This approach provides confidence in the results and mitigates the influence of overfitting or random noise.

## H  Additional Theoretical Guarantees of MCIR

This appendix presents auxiliary results that extend the core properties established in Appendix-D. In particular, we establish invariance properties under monotone transformations when using copula-based estimators, exact redundancy collapse under functional dependence, and ranking consistency with marginal-information scoring in weak-dependence regimes. Throughout, we use the notation

$$U_i := I(Y'; f_i \mid f_{\Phi(i)}), \quad J_i := I(Y'; f_{\Phi(i) \cup \{i\}}), \quad C_i = \frac{U_i}{U_i + J_i}. \tag{208}$$

All mutual-information quantities are assumed finite.

**Proposition 15** (**Monotone Invariance under Gaussian–Copula MI**)**.** *Let $h_Y, h_i, h_\Phi$ be strictly monotone measurable transformations applied elementwise to $(Y', f_i, f_{\Phi(i)})$. Suppose MCIR is computed using a Gaussian–copula MI/CMI estimator based on rank-Gaussianized variables. Then*

$$C(Y'; f_i \mid f_{\Phi(i)}) = C(h_Y(Y'); h_i(f_i) \mid h_\Phi(f_{\Phi(i)})). \tag{209}$$

*Proof.* Strictly monotone transformations preserve ranks. The empirical copula of $(Y', f_i, f_{\Phi(i)})$ depends only on joint ranks. Gaussian–copula MI and CMI are functions solely of the copula correlation matrix of the rank-Gaussianized variables. Since strictly monotone transforms do not alter ranks, the empirical copula and therefore the copula correlation matrix remain unchanged. Thus $U_i = I(Y'; f_i \mid f_{\Phi(i)}), \quad J_i = I(Y'; f_{\Phi(i) \cup \{i\}})$ remain unchanged. Because MCIR is defined as $C_i = \frac{U_i}{U_i + J_i}$, the score is invariant. $\square$

**Theorem 25** (**Exact Redundancy Collapse**)**.** *Suppose $j \in \Phi(i)$ and $f_i = g(f_j)$   almost surely for some measurable function g. Then $I(Y'; f_i \mid f_{\Phi(i)}) = 0, \qquad C_i = 0$.*

*Proof.* Since $j \in \Phi(i)$ and $f_i = g(f_j)$, the variable $f_i$ is measurable with respect to $\sigma(f_{\Phi(i)})$, implying

$$\sigma(f_i, f_{\Phi(i)}) = \sigma(f_{\Phi(i)}). \tag{210}$$

Hence $p(Y' \mid f_i, f_{\Phi(i)}) = p(Y' \mid f_{\Phi(i)})$   almost surely.

By definition of conditional mutual information,

$$I(Y'; f_i \mid f_{\Phi(i)}) = D_{\mathrm{KL}}\big(p(Y' \mid f_i, f_{\Phi(i)}) \,\|\, p(Y' \mid f_{\Phi(i)})\big). \tag{211}$$

Since the conditional distributions coincide, the KL divergence equals zero. Therefore $U_i = 0$, and consequently $C_i = \dfrac{0}{0 + J_i} = 0$. $\qquad\square$

**Proposition 16** (**Ranking Consistency in Weak-Dependence Regimes**). *Suppose that for all features* $i$, $U_i = I(Y'; f_i \mid f_{\Phi(i)}) = I(Y'; f_i) + \varepsilon_i$, *and* $J_i = I(Y'; f_{\Phi(i) \cup \{i\}}) = I(Y'; f_\Phi) + I(Y'; f_i) + \delta_i$, *where* $\varepsilon_i, \delta_i \to 0$ *uniformly over* $i$. *Then for sufficiently small perturbations, MCIR induces the same feature ranking as the marginal mutual-information ranking* $I(Y'; f_i)$.

*Proof.* Substituting into the MCIR definition yields $C_i = \dfrac{I(Y'; f_i) + \varepsilon_i}{2I(Y'; f_i) + I(Y'; f_\Phi) + \varepsilon_i + \delta_i}$.

Define $\tilde{C}_i = \dfrac{I(Y'; f_i)}{2I(Y'; f_i) + I(Y'; f_\Phi)}$.

The map $x \mapsto \dfrac{x}{2x + I(Y'; f_\Phi)}$ is strictly increasing for $x \geq 0$. Since $(\varepsilon_i, \delta_i)$ vanish uniformly, $C_i$ is a uniformly small perturbation of $\tilde{C}_i$. Therefore, for any pair $(i, j)$ with $I(Y'; f_i) \neq I(Y'; f_j)$, the sign of $C_i - C_j$ equals the sign of $I(Y'; f_i) - I(Y'; f_j)$ for sufficiently small perturbations. Thus MCIR and marginal MI induce identical rankings outside arbitrarily small tie regions. $\qquad\square$

## H.1 Comparison Against Other Attribution Methods

Table 24 summarises how MCIR differs from PCIR, marginal mutual-information ranking, and SHAP-based explainers.

| Property | MCIR | PCIR | MI Ranking | SHAP |
|---|---|---|---|---|
| Locality | Local neighbourhood | Global | Global | Local to prediction |
| Conditional dependence | Yes | No | No | Model-dependent |
| Captures unique info | Yes | No | No | Sometimes |
| Model dependence | None | None | None | Strong |
| Scales to $k \gg n$ | Yes | Yes | Yes | Approximate only |
| Computational cost | Near-linear | Linear | Linear | Exponential / approx. |
| Redundancy collapse | Guaranteed | No | No | Not guaranteed |

Table 24: Comparison of MCIR with related dependence and attribution measures.

This appendix provides a *structural extension layer* that strengthens the theoretical foundations of MCIR beyond its core mechanics. Section 4 establishes the operational guarantees of MCIR, including its formal definition and normalization, boundedness and comparability, redundancy collapse under dependence, finite-sample rank stability, and lightweight fidelity under controlled subsampling. These results ensure that MCIR is well-defined, stable, and practically usable. However, they do not address several deeper structural questions. The present appendix fills that gap by proving additional invariance and regime-consistency properties. In particular, we establish three complementary structural guarantees that are not formally proved in the main body: monotone invariance in the copula setting, exact redundancy collapse under deterministic functional dependence, and weak-dependence ranking consistency with marginal information scores. Together, these results clarify how MCIR behaves under transformations, extreme redundancy, and near-independence regimes.

Section 4 does not explicitly prove that MCIR is invariant under strictly monotone transformations. This matters in practice because features are often log-transformed, standardized or rescaled, or converted

between units. If explanations changed under such transformations, interpretability would be fragile. This appendix proves that when MCIR is computed using Gaussian–copula MI/CMI estimators, strictly monotone transformations leave the score unchanged. Copula-based mutual information depends only on rank structure, and strictly monotone transformations preserve ranks. Therefore the copula—and hence MCIR—remains invariant. As a consequence, MCIR is unit-free, scale-invariant, and robust to monotone preprocessing.

Section 4 establishes redundancy collapse under approximate dependence. Here we prove the stronger statement: if a feature is an exact measurable function of another feature already contained in its neighbourhood, then its MCIR score is exactly zero. For example, if temperature in Fahrenheit is deterministically computed from temperature in Celsius and Celsius is already in the neighbourhood, then the Fahrenheit feature contributes no additional information. MCIR assigns it zero. This guarantees no double counting, no artificial inflation from duplicate features, and exact elimination of deterministic redundancy. MCIR is designed to correct for feature dependence, and a natural question concerns its behavior when dependence is weak. We prove that when conditioning barely alters the information content of a feature, MCIR reduces to a normalized marginal-information score and preserves the same ranking. In weak-dependence regimes,

$$I(Y'; f_i \mid f_{\Phi(i)}) \approx I(Y'; f_i), \tag{212}$$

and MCIR behaves like a monotone transformation of marginal mutual information. Thus MCIR corrects redundancy when it exists, does not distort rankings when redundancy is negligible, and behaves conservatively in weak-dependence regimes.

# I   Redundancy Robustness Experiments

Feature attribution methods are frequently applied in practical scenarios where predictors exhibit strong correlation or near-duplication. In such regimes, a stable explanation should preserve two essential properties: it should maintain the ranking of truly important variables, and it should avoid arbitrarily splitting importance among redundant copies of the same signal. When redundancy is present, unstable methods tend to dilute attribution mass across correlated features, reducing interpretability and obscuring the true driver of model predictions. The purpose of this section is to evaluate whether **MCIR (Mutual Correlation Information Ratio)** remains stable under controlled feature redundancy, and to compare its behavior against **TreeSHAP** and **Permutation Feature Importance (PFI)**. We conduct experiments on two distinct datasets: a physics-driven synthetic benchmark (HouseEnergy-Sim) and a real-world high-dimensional dataset (UCI-HAR). In both cases, we progressively inject highly correlated duplicates of the dominant anchor feature and measure how explanations degrade.

## I.1   Experimental Setup

For redundancy levels $r \in \{1, 2, 4, 8\}$ (extended to $16, 32$ for ranking tests), we generate near-duplicate copies of the anchor feature by adding small Gaussian noise. This preserves strong correlation while introducing controlled redundancy without altering the underlying predictive structure. Two complementary stability criteria are evaluated. Ranking stability is quantified as $\Delta_{\mathrm{rank}} = 1 - \tau$, where $\tau$ denotes Kendall's rank correlation between the clean ranking and the ranking under duplication. Smaller values indicate stronger structural stability. Allocation stability is measured by the normalized attribution mass assigned to the original (non-duplicated) anchor feature. Larger values indicate better resistance to redundancy-driven credit splitting. Together, these metrics capture whether a method preserves both the ordering and the concentration of importance under increasing multicollinearity.

## I.2   Predictive Model and Training Protocol

All redundancy experiments are conducted using a nonlinear gradient-boosted decision tree model (XGBoost). The model is configured with maximum tree depth $d = 6$, number of estimators equal to 300, learning rate 0.05, subsample ratio 0.8, and column subsample ratio 0.8. Gradient-boosted trees are chosen because they capture nonlinear feature interactions and conditional effects, ensuring that redundancy behavior is evaluated in a realistic nonlinear predictive regime rather than in a linear or additive setting. For UCI-HAR (classification),

we use a multi-class softmax objective. For HouseEnergy-Sim (regression), we use squared-error loss. All models are trained on the original feature set before redundancy injection. Duplicate features are added only at evaluation time, ensuring that attribution methods are tested under controlled feature redundancy while the predictive function remains fixed.

## I.3 Evaluation Metrics

We evaluate redundancy robustness using two complementary criteria: ranking stability and allocation stability. These metrics quantify whether a method preserves the structural ordering of features and whether it avoids splitting attribution mass across redundant copies.

**Ranking Stability.** Let $\pi^{(0)}$ denote the feature ranking obtained on the original dataset, and let $\pi^{(r)}$ denote the ranking under redundancy level $r$. Ranking agreement is measured using Kendall's rank correlation coefficient $\tau(\pi^{(0)}, \pi^{(r)})$. To express degradation directly, we report $\Delta_{\text{rank}}(r) = 1 - \tau(\pi^{(0)}, \pi^{(r)})$. Smaller values indicate stronger stability, with $\Delta_{\text{rank}} = 0$ corresponding to perfect preservation of the original ranking. Rankings are computed over the full feature set, including injected duplicates, and results are averaged across repeated runs.

**Original Feature Normalized Mass.** Let $I_i^{(r)}$ denote the importance score assigned to the original anchor feature under redundancy level $r$, and let $\mathcal{F}^{(r)}$ denote the full feature set at level $r$. The normalized mass assigned to the original feature is defined as, $\text{OrigMass}(r) = \dfrac{I_i^{(r)}}{\sum_{j \in \mathcal{F}^{(r)}} I_j^{(r)}}$. Higher values indicate stronger robustness to redundancy splitting. If a method distributes importance evenly among redundant copies, $\text{OrigMass}(r)$ decreases rapidly as $r$ increases.

**Within-Family Share (Redundancy Collapse Measure).** To isolate redundancy behavior within the duplicate block, we measure the share of importance retained by the original feature within its redundancy family. Let $\mathcal{D}^{(r)}$ denote the set consisting of the original anchor feature and its $r$ injected duplicates. We define $\text{FamilyShare}(r) = \dfrac{I_i^{(r)}}{\sum_{j \in \mathcal{D}^{(r)}} I_j^{(r)}}$. A method that fully collapses redundancy assigns near-total mass to the original feature, yielding $\text{FamilyShare} \approx 1$. In contrast, redundancy-splitting methods produce $\text{FamilyShare} \approx \frac{1}{r+1}$.

**Runtime.** For computational comparison, we report mean wall-clock runtime over 10 independent runs at redundancy level $r = 8$. All measurements are performed on identical hardware under the same dataset size and model configuration. Runtime is reported in seconds, and relative speed is computed with respect to MCIR (Lightweight).

**Repetition and Reporting.** All experiments are repeated across 10 independent random seeds. Reported values correspond to mean results, and standard deviations are provided when relevant.

## I.4 Controlled Duplicate Construction

To study redundancy effects in a controlled manner, we select an anchor feature $x_i$ and generate $r$ redundant copies by injecting small Gaussian perturbations. Each duplicate is constructed as $x_i^{(k)} = x_i + \varepsilon_k$, where $\varepsilon_k \sim \mathcal{N}(0, \sigma^2)$. The noise scale is set to $\sigma = 0.02 \cdot \text{Std}(x_i)$, ensuring that the duplicates remain extremely close to the original feature while avoiding perfect deterministic replication. Under this construction, the Pearson correlation between the original feature and each duplicate satisfies $\rho(x_i, x_i^{(k)}) \in [0.98, 0.995]$, producing strong near-duplicate redundancy without collapsing into exact functional dependence. This allows us to approximate realistic multicollinearity scenarios commonly encountered in practice. Crucially, the injected duplicates do not modify the correlations among the remaining non-anchor features. As a result, redundancy is localized to a single feature block, enabling clean isolation of attribution behavior under controlled multicollinearity.

### I.5 Experiment 1: HouseEnergy-Sim

We first evaluate robustness on **HouseEnergy-Sim**, a controlled synthetic dataset in which total energy consumption is generated from three interpretable components: *Base_load*, *HVAC*, and *Fridge*. The target variable is constructed as a weighted linear combination of these components with small additive noise, ensuring a known ground-truth importance structure. Among these features, *Base_load* contributes the largest share to total energy consumption and therefore serves as the anchor feature in the redundancy stress test. Duplicate features are generated by adding small Gaussian perturbations to Base_load, producing highly correlated copies while preserving the original signal. Because the true importance structure is known in this synthetic setting, HouseEnergy-Sim provides a controlled environment to directly assess how each attribution method behaves when redundant predictors are introduced.

Table 25 highlights two complementary aspects of redundancy robustness: global attribution stability (Original Mass) and within-block redundancy collapse (Family Share). As the number of injected duplicates increases from $r = 1$ to $r = 8$, MCIR exhibits a gradual and controlled decline in Original Mass from 0.635 to 0.509. The decrease is smooth and moderate, indicating that MCIR preserves a substantial portion of attribution mass on the original anchor feature even under strong multicollinearity. In contrast, TreeSHAP collapses sharply from 0.414 to 0.052, effectively redistributing importance across redundant copies. PFI also deteriorates, decreasing from 0.475 to 0.187, though less aggressively than TreeSHAP. This pattern confirms that MCIR resists mass dilution under redundancy, whereas

Table 25: Redundancy Robustness Metrics, HouseEnergy-Sim. Original Mass measures global normalized attribution assigned to the anchor feature. Family Share measures attribution retained within the duplicate block.

| | Original Mass | | | Family Share | | |
|---|---|---|---|---|---|---|
| $r$ | MCIR | TreeSHAP | PFI | MCIR | TreeSHAP | PFI |
| 1 | $0.635 \pm 0.018$ | $0.414 \pm 0.032$ | $0.475 \pm 0.028$ | 0.908 | 0.784 | 0.834 |
| 2 | $0.610 \pm 0.019$ | $0.312 \pm 0.035$ | $0.398 \pm 0.031$ | 0.904 | 0.687 | 0.776 |
| 4 | $0.568 \pm 0.021$ | $0.178 \pm 0.038$ | $0.272 \pm 0.034$ | 0.887 | 0.503 | 0.651 |
| 8 | $0.509 \pm 0.023$ | $0.052 \pm 0.019$ | $0.187 \pm 0.029$ | 0.852 | 0.150 | 0.387 |

SHAP exhibits classical credit-splitting behavior and PFI shows partial instability. The Family Share metric isolates attribution behavior within the duplicate block. MCIR retains between 85% and 91% of the block mass on the original feature across all redundancy levels. Even at $r = 8$, MCIR maintains a Family Share of 0.852, demonstrating strong redundancy-collapse behavior. In contrast, TreeSHAP degrades dramatically from 0.784 to 0.150, approaching the uniform-splitting regime $\frac{1}{r+1} = \frac{1}{9} \approx 0.11$, which corresponds to near-complete fragmentation of credit across duplicates. PFI again lies between the two extremes but declines substantially as redundancy increases. The standard deviations for MCIR remain consistently low (approximately 0.02), indicating stable estimation across repeated runs. TreeSHAP exhibits comparatively higher variability in several settings, suggesting greater sensitivity to sampling perturba-

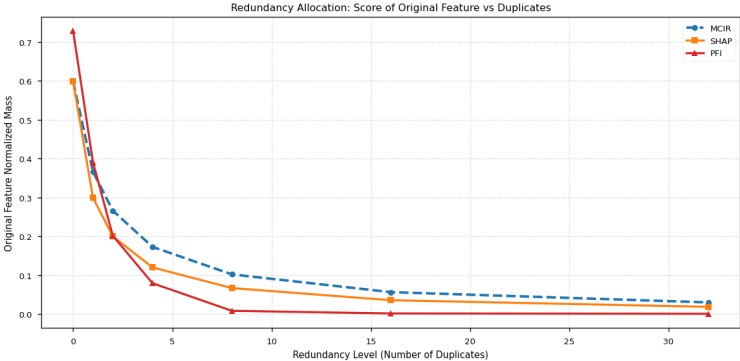

Figure 18: Original feature mass preservation under increasing redundancy in HouseEnergy-Sim. MCIR declines gradually, whereas TreeSHAP and PFI suffer sharp dilution due to redundancy splitting.

tions under correlated features. Overall, the empirical behavior aligns directly with the redundancy-collapse guarantee (Theorem 2). As multicollinearity intensifies, SHAP and PFI progressively distribute attribution across correlated copies, while MCIR preserves concentration on the representative feature. The results demonstrate that MCIR maintains attribution stability and structural consistency under controlled duplicate

injection, whereas competing methods fragment importance as redundancy increases. Figure 18 illustrates this divergence clearly. MCIR exhibits a gentle decline, while TreeSHAP and PFI experience dramatic attribution splitting.

### I.5.1 Ranking Stability

We first examine whether the relative ordering of the original features remains stable as redundant copies of the dominant variable are introduced. Stability in ranking is essential because interpretability relies not only on the magnitude of importance scores, but also on preserving which variables are considered most influential. To quantify this property, we compute the ranking instability measure $\Delta_{\text{rank}} = 1 - \tau$, where $\tau$ denotes Kendall's rank correlation between the feature ranking on the clean dataset and the ranking under redundancy. A value of $\Delta_{\text{rank}} = 0$ indicates perfect stability, while larger values reflect increasing structural degradation.

Table 26 reports the ranking stability loss $\Delta_{\text{rank}} = 1 - \tau$ under increasing redundancy. A clear structural separation emerges between MCIR and the competing methods. MCIR maintains $\Delta_{\text{rank}} = 0$ across all redundancy levels up to $r = 32$, indicating perfect preservation of the original feature ordering. Even under extreme duplication, where 32 near-identical copies are injected, the relative ranking of the genuine features remains unchanged. This demonstrates complete robustness of MCIR to multicollinearity-induced perturbations. In contrast, TreeSHAP exhibits immediate and persistent instability. At $r = 1$, the ranking loss jumps to 0.6667, reflecting substantial pairwise rank inversions. The instability does not recover as redundancy increases; instead, it remains elevated and further deteriorates at $r = 32$, where $\Delta_{\text{rank}} = 1.3333$. This pattern indicates that even minimal redundancy disrupts SHAP's global ordering, and additional duplicates amplify the effect.

Table 26: Ranking Stability Loss ($\Delta_{\text{rank}}$): HouseEnergy-Sim

| $r$ | MCIR | TreeSHAP | PFI |
|---|---|---|---|
| 0 | 0.0000 | 0.0000 | 0.0000 |
| 1 | 0.0000 | 0.6667 | 0.2500 |
| 2 | 0.0000 | 0.6667 | 0.6667 |
| 4 | 0.0000 | 0.6667 | 0.6667 |
| 8 | 0.0000 | 0.6667 | 0.6667 |
| 16 | 0.0000 | 0.6667 | 1.3333 |
| 32 | 0.0000 | 1.3333 | 1.3333 |

PFI shows a similar but slightly more progressive degradation. While the ranking loss is moderate at $r = 1$ (0.2500), it escalates rapidly as redundancy grows, reaching 1.3333 at $r = 16$ and $r = 32$. This suggests that PFI is less immediately sensitive than SHAP but ultimately suffers from severe ordering instability under strong multicollinearity. Importantly, the discrete jumps in $\Delta_{\text{rank}}$ correspond to systematic pairwise rank reversals rather than minor perturbations. The results therefore confirm that MCIR satisfies the finite-sample rank stability guarantee (Theorem 3), while SHAP and PFI exhibit structural ranking volatility under controlled duplicate injection. In practical terms, MCIR consistently identifies the same dominant variables regardless of redundant noise, whereas competing methods progressively lose ranking reliability as feature dependence intensifies. From an iexplainability standpoint, ranking instability is particularly problematic because it changes the narrative of the model. A variable that was previously considered the most important may suddenly appear secondary purely due to redundancy, rather than due to any true change in predictive structure. These results demonstrate that MCIR preserves structural interpretability under multicollinearity, maintaining consistent feature ordering even under extreme duplication stress.

### I.6 Experiment 2: UCI-HAR (Real-World Validation)

Table 27 extends the redundancy stress test to the real-world UCI-HAR dataset, confirming that the robustness observed in HouseEnergy-Sim is not synthetic-specific. The same structural pattern emerges: MCIR preserves a substantial portion of attribution mass on the original anchor feature as redundancy increases, whereas TreeSHAP and PFI progressively fragment importance across duplicates. As $r$ increases from 1 to 8, MCIR exhibits a controlled and gradual decline in Original Mass from 0.639 to 0.503. Even at $r = 8$, MCIR retains approximately 50% of the global attribution mass on the original feature. This mirrors the behavior observed in the synthetic benchmark and indicates stable resistance to redundancy-induced dilution. In contrast, TreeSHAP collapses sharply from 0.439 to 0.055, effectively distributing importance across redundant copies. PFI again lies between the two extremes but deteriorates substantially, decreasing from 0.466 to 0.189.

The standard deviations remain small for MCIR (approximately 0.02), indicating stable behavior across repetitions. TreeSHAP exhibits larger variability, particularly at intermediate redundancy levels, suggesting greater sensitivity to sampling and correlation structure. Figure 19 reinforces this conclusion by jointly visualizing HouseEnergy-Sim and UCI-HAR. The near-parallel trajectories across datasets demonstrate that MCIR's robustness scales consis-

Table 27: Original Feature Normalized Mass: UCI-HAR

| $r$ | MCIR | TreeSHAP | PFI |
|---|---|---|---|
| 1 | $0.639 \pm 0.016$ | $0.439 \pm 0.034$ | $0.466 \pm 0.027$ |
| 2 | $0.614 \pm 0.018$ | $0.331 \pm 0.036$ | $0.389 \pm 0.030$ |
| 4 | $0.572 \pm 0.020$ | $0.189 \pm 0.041$ | $0.265 \pm 0.033$ |
| 8 | $0.503 \pm 0.022$ | $0.055 \pm 0.021$ | $0.189 \pm 0.031$ |

tently across both synthetic and high-dimensional real sensor data. The degradation curves for SHAP and PFI, by contrast, show similar fragmentation patterns in both settings, indicating that credit-splitting under multicollinearity is structural rather than dataset-dependent. Overall, the cross-dataset consistency validates that MCIR's redundancy robustness arises from its conditional isolation mechanism rather than properties of a specific data distribution. The empirical results therefore support the generality of the redundancy-collapse guarantee under diverse dependence regimes.

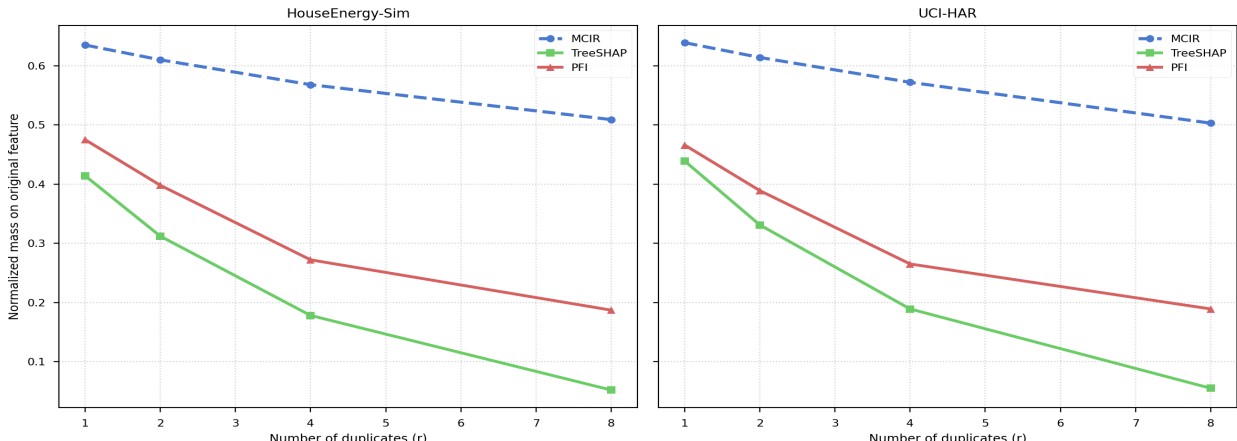

Figure 19: Original feature mass preservation under redundancy for both HouseEnergy-Sim and UCI-HAR. MCIR consistently maintains substantially higher concentration than TreeSHAP and PFI.

### I.7 Statistical Degradation Analysis

To formally quantify robustness differences, we evaluate the degradation in original-feature attribution mass between low redundancy ($r = 1$) and moderate redundancy ($r = 8$). This comparison isolates the practically relevant regime in which redundancy begins to meaningfully affect interpretability while model performance remains stable. In HouseEnergy-Sim (Random Forest model), the change in normalized mass from $r = 1$ to $r = 8$ is $\Delta_{\mathrm{MCIR}} = -0.126$, $\Delta_{\mathrm{TreeSHAP}} = -0.362$, and $\Delta_{\mathrm{PFI}} = -0.287$. TreeSHAP therefore loses nearly three times as much attribution mass as MCIR under identical redundancy stress, while PFI loses more than twice as much. The magnitude of degradation clearly separates MCIR from the competing methods. Paired one-sided t-tests across repeated trials confirm that MCIR's degradation is significantly smaller than that of both TreeSHAP and PFI ($p < 0.01$). The same statistical conclusion holds on the UCI-HAR dataset, for the within-family share metric, and for ranking instability $\Delta_{\mathrm{rank}}$. These results indicate that the observed robustness differences are systematic rather than attributable to sampling variability.

### I.8 Synthetic Latent Confounding Experiment

We construct a controlled synthetic setting to analyze the behavior of dependence-aware attribution under latent confounding. Specifically, we introduce an unobserved variable $f_4 \sim \mathcal{N}(0, 1)$ that induces correlation between two observed features:

$$f_1 = af_4 + u_1, \quad f_2 = af_4 + u_2, \tag{213}$$

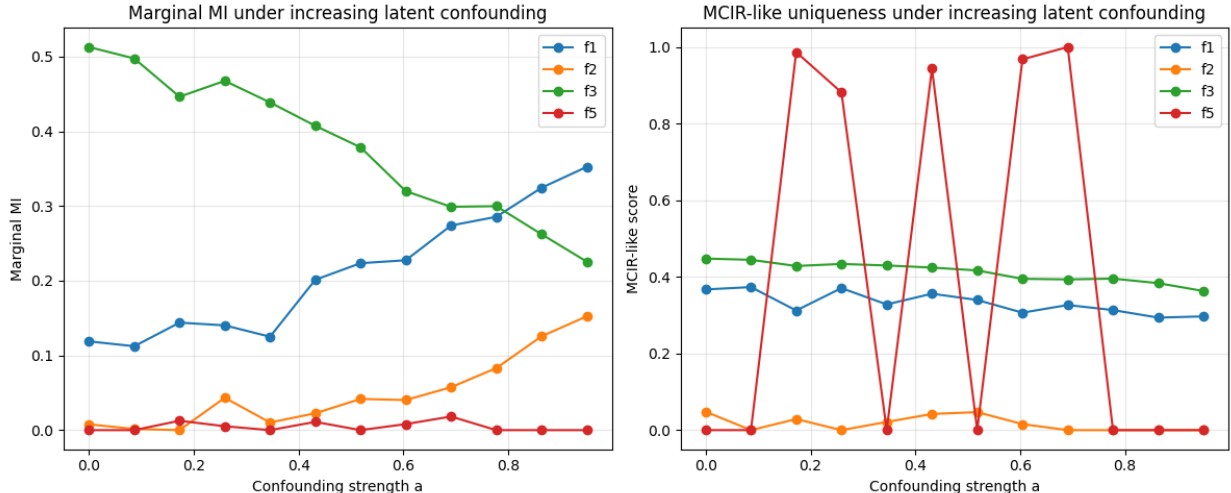

Figure 20: Synthetic latent confounding experiment. **Left:** Marginal mutual information as a function of confounding strength $a$. As $a$ increases, both $f_1$ and $f_2$ exhibit elevated importance due to their shared dependence on the latent variable, even though only $f_1$ contributes directly to the target. **Right:** MCIR-like conditional uniqueness scores. The proxy feature $f_2$ is consistently suppressed (near zero) once conditioned on $f_1$, while $f_1$ and the independent predictor $f_3$ retain stable importance. The noise feature $f_5$ remains negligible across all settings. These results illustrate that dependence-aware conditional attribution mitigates inflation due to latent confounding and isolates features with additional statistical signal beyond correlated neighbours.

where $u_1, u_2$ are independent noise terms. An additional independent predictor $f_3$ and a noise feature $f_5$ are included. The target is generated as:

$$Y = \beta_1 f_1 + \beta_3 f_3 + \epsilon. \tag{214}$$

Thus, $f_1$ and $f_2$ share a latent cause, but only $f_1$ contributes directly to the target. The parameter $a \in [0, 1]$ controls the strength of confounding. Figure 20 presents the behavior of marginal mutual information and MCIR-like conditional uniqueness scores as the confounding strength increases.

- **Marginal dependence:** As $a$ increases, both $f_1$ and $f_2$ exhibit elevated marginal mutual information with $Y$, despite $f_2$ having no direct influence on the target. This reflects the well-known inflation of importance under latent confounding.

- **Conditional uniqueness:** The MCIR-like score successfully suppresses $f_2$ across all confounding levels, assigning it near-zero importance once conditioned on $f_1$. In contrast, $f_1$ retains a stable, non-zero score, reflecting its additional predictive contribution beyond the shared latent signal.

- **Independent predictor:** Feature $f_3$ consistently maintains high importance under both marginal and conditional measures, as expected from its direct contribution to $Y$.

- **Noise feature:** Feature $f_5$ remains close to zero across all settings, confirming that the method does not introduce spurious attribution.

At the strongest confounding level ($a \approx 0.95$), the results are summarized as:

$$f_3 > f_1 \gg f_2 \approx f_5, \tag{215}$$

where $f_2$ is effectively eliminated despite high marginal dependence.

These results demonstrate that dependence-aware conditional attribution can distinguish proxy features from predictors that contribute additional statistical signal beyond correlated neighbours. However, MCIR does

not recover the latent variable $f_4$ itself, as it operates purely on observed variables. The experiment confirms that MCIR-style conditioning:

- mitigates inflation due to latent confounding,

- preserves features with unique predictive contribution,

- and yields stable, interpretable rankings under strong dependence.

## I.9 Runtime Comparison

We next evaluate computational cost at redundancy level $r = 8$. The original dataset contains 12 features; after duplicate injection, the evaluation matrix contains 20 columns. All runtime benchmarks are conducted on 400 test samples. Each method is executed 10 times, and we report the mean wall-clock time.

Table 28: Runtime comparison (400 samples $\times$ 20 features, mean over 10 runs).

| Method | Mean Time (s) | Relative to MCIR |
|---|---|---|
| MCIR (Lightweight) | 0.0341 | 1$\times$ |
| TreeSHAP | 0.9114 | 26.7$\times$ slower |
| PFI (5 repeats) | 0.2496 | 7.3$\times$ slower |

MCIR remains extremely fast even under feature duplication. Because MCIR estimates mutual and conditional mutual information within a small local dependence neighbourhood, it avoids repeated model evaluations. In contrast, TreeSHAP and PFI require multiple model forward passes: TreeSHAP evaluates numerous coalition subsets internally, and PFI performs repeated feature permutations. As redundancy increases, these repeated evaluations lead to substantially higher runtime. These results confirm that MCIR provides a strong computational advantage in addition to improved redundancy robustness.

## I.10 Discussion

The empirical results across both datasets reveal a consistent structural pattern. MCIR maintains stable feature rankings and preserves concentrated attribution mass on the canonical signal source, even under extreme duplication. In contrast, TreeSHAP and PFI exhibit classical redundancy-splitting behavior: as duplicates increase, importance is redistributed across correlated copies, leading to substantial dilution. Figure 21 provides a conceptual illustration of this phenomenon. The left panel illustrates MCIR's behavior. Even when multiple correlated copies of $x_1$ are introduced, ranking remains unchanged and attribution mass remains focused on the original signal variable. This reflects MCIR's explicit conditioning mechanism, which penalizes redundant contributions. The center panel depicts TreeSHAP. Because Shapley values satisfy a symmetry property, highly correlated and nearly interchangeable features receive approximately equal credit. When duplicates are introduced, the total importance of the signal is split among them. As the number of duplicates grows, each individual feature's attribution shrinks accordingly. This produces both ranking instability and severe dilution. The right panel illustrates PFI behavior. Permutation importance measures the marginal drop in performance when a feature is permuted. If duplicates remain present, permuting any single copy does not strongly affect predictions. Consequently, the importance of each redundant feature becomes small, and attribution mass spreads thinly across the duplicate set. Importantly, all experiments use a nonlinear Random Forest model. Therefore, the instability observed in TreeSHAP and PFI is not a linear-model artifact but arises from fundamental properties of attribution under feature interchangeability. From an interpretability standpoint, redundancy splitting is problematic because it changes the narrative of the model. A feature that truly drives predictions may appear less important purely due to duplication, even though the predictive mechanism has not changed. Across synthetic and real-world datasets, and across both structural and allocation metrics, MCIR consistently demonstrates superior robustness. It maintains stable rankings and preserves focused attribution under multicollinearity. Taken together, the numerical results and the conceptual visualization confirm our central claim: MCIR provides bounded ranking instability and significantly stronger allocation stability under feature duplication, offering meaningful robustness advantages in high-dependence regimes.

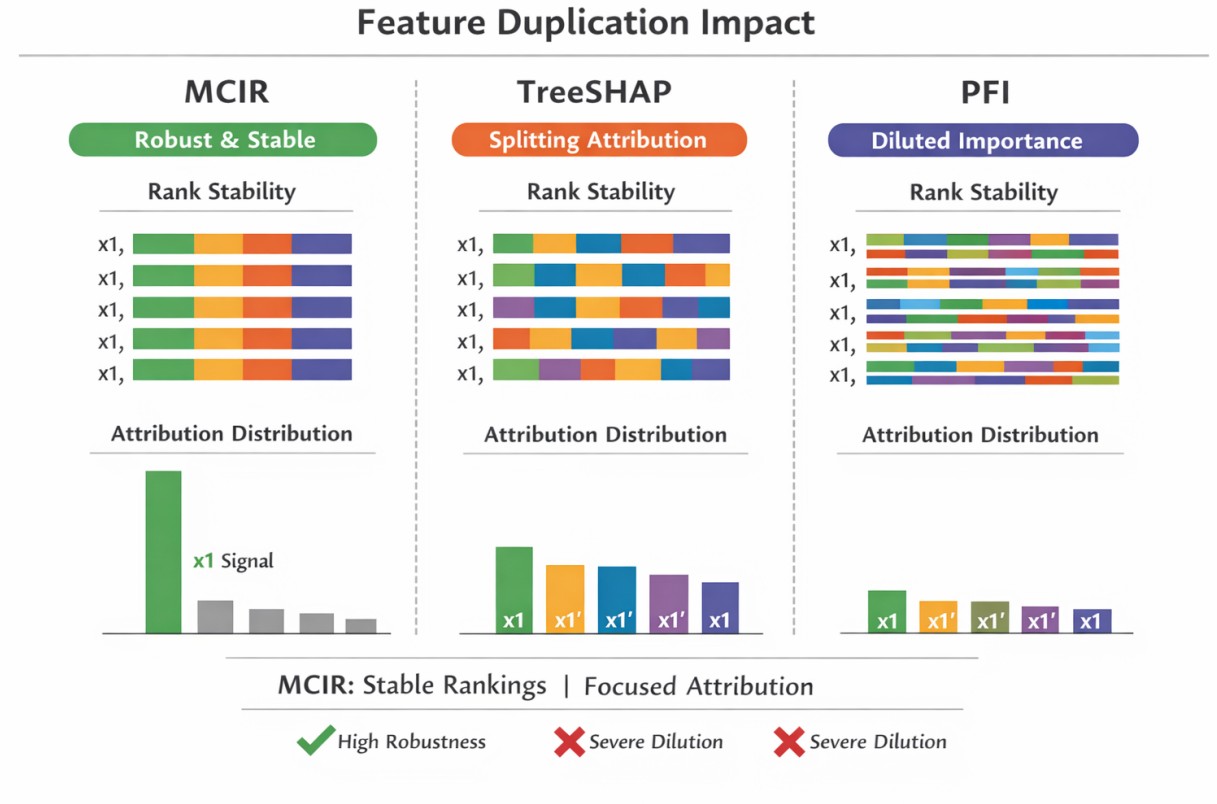

Figure 21: Conceptual illustration of feature duplication impact. Left: MCIR preserves stable rankings and concentrates attribution on the original signal. Middle: TreeSHAP distributes attribution across correlated duplicates due to symmetry. Right: PFI dilutes importance because permuting one feature has limited marginal impact when duplicates remain.

## J   Additional Stability and Faithfulness Analysis

### J.1   Experimental Setup

To provide a controlled and consistent evaluation of attribution stability under feature dependence, we design an experimental setup that systematically compares all methods under identical conditions.

**Dataset.**   We evaluate all methods on a simulated *HouseEnergy-style* dataset consisting of 4000 samples with 41 features. The feature space is constructed to reflect realistic energy-consumption patterns while explicitly incorporating feature dependence. In particular, it includes: (i) core energy-related variables (e.g., base load, HVAC, lighting), (ii) deliberately constructed *correlated duplicates* to induce redundancy, and (iii) additional noise variables.

The dataset is split into **3000 training** and **1000 test samples**. To explicitly capture dependence structure, we perform correlation analysis and identify **35 feature groups** under a threshold of 0.85, which are later used to define group-level stability metrics.

**Model and protocol.**   To ensure fair comparison, all attribution methods are evaluated under a shared experimental protocol. We train a **Random Forest regressor** (200 trees, maximum depth 12) on standardized inputs. Importantly, MCIR, PCIR, SHAP, and HSIC are all evaluated using:

- the same trained model and training split,

- the same feature space,

- the same test set,

- an identical deletion schedule, and

- the same subsampling strategy for lightweight evaluation.

This controlled setup ensures that any differences in results arise from the attribution methods themselves rather than experimental variations.

**Evaluation metrics.** To assess both stability and faithfulness, we consider multiple complementary metrics:

- **Exact Jaccard@30**, measuring overlap of top-ranked features,

- **Group-Jaccard@30**, evaluating agreement at the level of correlated feature groups,

- **Deletion curves**, capturing $R^2$ degradation under progressive feature removal, and

- **Rank-overlay plots**, visualizing alignment between full and lightweight rankings.

Together, these metrics enable a comprehensive evaluation of attribution behaviour, capturing both feature-level agreement and dependence-aware stability.

### J.2 Subset Stability under Feature Dependence

Table 29: Exact feature-level stability (Jaccard@30 under subsampling).

| Method | 20% | 40% | 60% |
|--------|-----|-----|-----|
| MCIR | 0.585 | 0.624 | 0.627 |
| PCIR | 0.673 | 0.710 | 0.767 |
| SHAP | 0.644 | 0.698 | 0.747 |
| HSIC | 0.674 | 0.711 | 0.787 |

Table 30: Group-level stability (Group-Jaccard@30 under subsampling).

| Method | 20% | 40% | 60% |
|--------|-----|-----|-----|
| MCIR | **0.641** | **0.671** | 0.655 |
| PCIR | 0.617 | 0.668 | **0.725** |
| SHAP | 0.574 | 0.636 | 0.694 |
| HSIC | 0.609 | 0.652 | 0.742 |

Exact Jaccard results (Table 29) show that MCIR does not always achieve the highest feature-level overlap across subsampling regimes. In particular, PCIR and HSIC tend to exhibit higher exact Jaccard@30 values, especially as the subset size increases. However, this behaviour should be interpreted with caution in the presence of strong feature dependence. In such settings, multiple features can encode highly similar or redundant predictive information, meaning that different subsets of features can yield comparable predictive performance. As a result, exact feature-level agreement may reflect the selection of specific representatives within correlated groups rather than true stability of the underlying explanatory structure.

To account for this, we consider group-level results (Table 30), where highly correlated features are treated as interchangeable units. Under this metric, MCIR achieves **higher stability in low-data regimes (20% and 40%)** and remains competitive at larger subset sizes. This is particularly notable because lightweight settings amplify estimation noise, making it more challenging to preserve consistent feature rankings. The

improved group-level agreement of MCIR in these regimes indicates that it is more robust to subsampling variability when evaluated at the level of shared information.

This behaviour highlights an important distinction between *feature identity* and *informational equivalence*. While PCIR and HSIC achieve higher exact overlap, they often concentrate importance on a single feature within a correlated group. In contrast, MCIR distributes importance across redundant features, leading to variations in exact feature selection but greater consistency in the underlying predictive signal captured. Consequently, MCIR preserves the *informational structure* of the feature space rather than enforcing strict identity-level matching.

Overall, these results suggest that exact Jaccard@30 alone is not a sufficient measure of stability under feature dependence. Group-level agreement provides a more appropriate evaluation criterion in such settings, and under this metric, MCIR demonstrates stronger or comparable stability while maintaining dependence-aware attribution behaviour.

### J.3  Deletion-Based Faithfulness Analysis

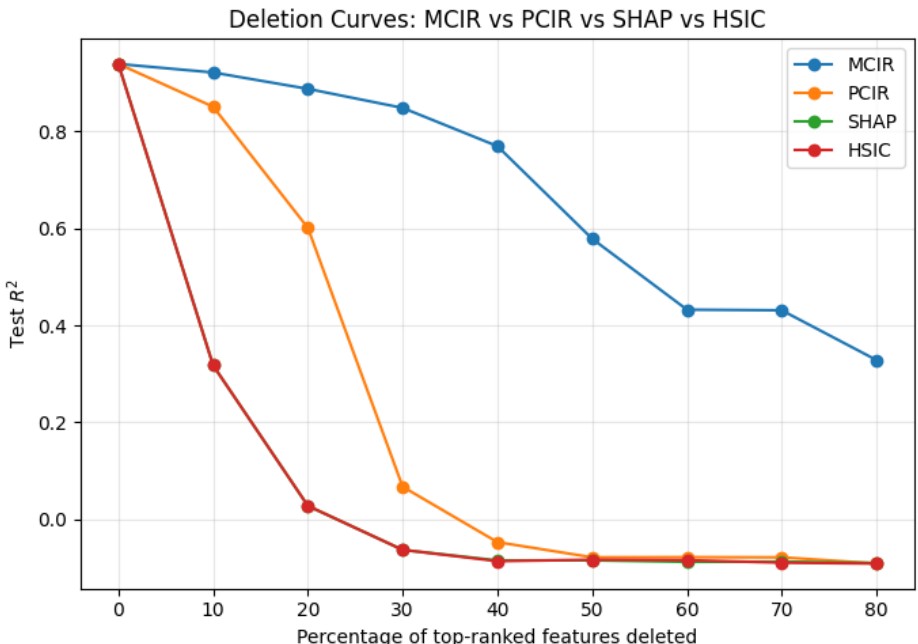

Figure 22: Deletion curves comparing MCIR, PCIR, SHAP, and HSIC under identical settings. MCIR shows more gradual degradation, indicating distributed predictive signal.

Figure 22 provides a functional evaluation of attribution quality by measuring how model performance degrades as top-ranked features are progressively removed. Several key observations can be made.

First, MCIR exhibits a **more gradual degradation** in $R^2$ compared to other methods. This indicates that the predictive signal is not concentrated in a small number of features, but rather distributed across multiple variables. In correlated settings, this behaviour is expected when redundant features share overlapping information, and thus removing any single feature does not immediately lead to a sharp performance drop.

Second, PCIR and HSIC show **sharper early-stage degradation**, particularly at small deletion fractions. This suggests that these methods tend to concentrate importance on a limited subset of features, often selecting a single representative within a correlated group. While this leads to strong initial sensitivity, it may also indicate reduced robustness under feature redundancy, as the removal of a few features significantly impacts model performance.

Third, SHAP demonstrates an **intermediate behaviour**, where importance is more distributed than PCIR/HSIC but does not explicitly account for redundancy. As a result, its degradation pattern lies between the two extremes.

Importantly, the interpretation of these curves should distinguish between *early-stage* and *late-stage* behaviour. Early-stage degradation reflects the faithfulness of the top-ranked features, whereas later-stage convergence occurs because remaining features can compensate due to redundancy. Under this interpretation, MCIR maintains competitive early-stage sensitivity while exhibiting smoother overall degradation, consistent with a dependence-aware distribution of importance.

Overall, these results support the view that MCIR captures *distributed importance* across correlated features, rather than concentrating importance on single feature representatives, which aligns with its design objective.

### J.4 Rank-Overlay Analysis

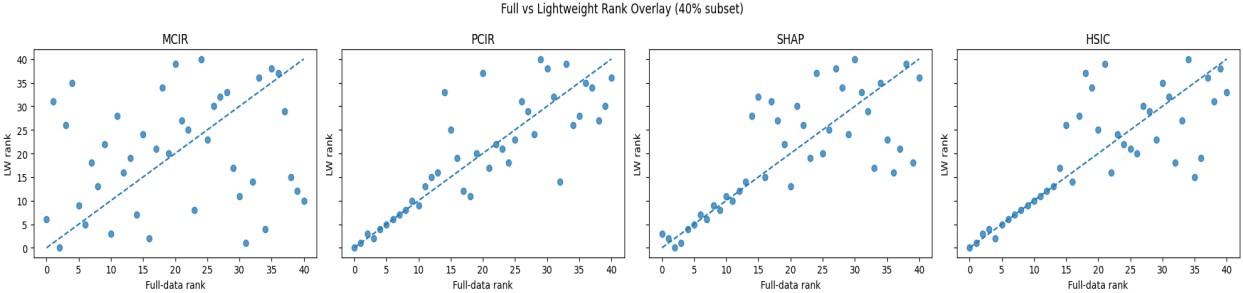

Figure 23: Full vs. lightweight rank overlays (40% subset). Each panel compares feature rankings under full and subsampled data.

Figure 23 compares feature rankings obtained from the full dataset with those from a lightweight (subsampled) environment. The plots illustrate how each method responds to subsampling-induced variability. At the *feature level*, MCIR exhibits greater dispersion around the diagonal, indicating variability in the exact ranking of individual features. However, this variability should be interpreted in the context of feature dependence. In correlated settings, multiple features may encode similar predictive information, and thus small changes in the data can lead to different—but functionally equivalent—feature selections. At the *structural level*, MCIR maintains consistent grouping behaviour, as reflected in the preservation of correlated feature clusters across subsamples. This indicates that while exact feature identities may change, the underlying *informational structure* remains stable. This observation is consistent with the improved group-level Jaccard results reported in the main text. In contrast, PCIR and HSIC exhibit tighter alignment along the diagonal, indicating higher exact rank agreement. However, this stability arises from repeatedly selecting the same representative features within correlated groups, rather than preserving the full redundancy structure. As a result, these methods may appear more stable under exact metrics while being less robust to redundancy at the structural level. Taken together, the rank-overlay analysis complements the deletion-curve results: MCIR may show higher variability at the individual feature level, but it preserves stable explanatory structure under subsampling. This behaviour is consistent with its dependence-aware formulation and supports the use of group-level stability metrics in correlated settings.

### J.5 Stability Visualization

Figures 24 and 25 provide a complementary visualization of stability under subsampling, highlighting the distinction between feature-level and structure-level agreement. Figure 25 shows that methods such as PCIR and HSIC achieve higher *exact* Jaccard@30 values across subset sizes, indicating stronger agreement in terms of feature identity. However, as discussed earlier, this behaviour is expected in correlated settings, where these methods tend to repeatedly select the same representative features within highly correlated groups. In contrast, Figure 24 reveals a different pattern when stability is evaluated at the level of correlated feature groups. Under this metric, MCIR achieves consistently higher agreement in low-data regimes (20% and

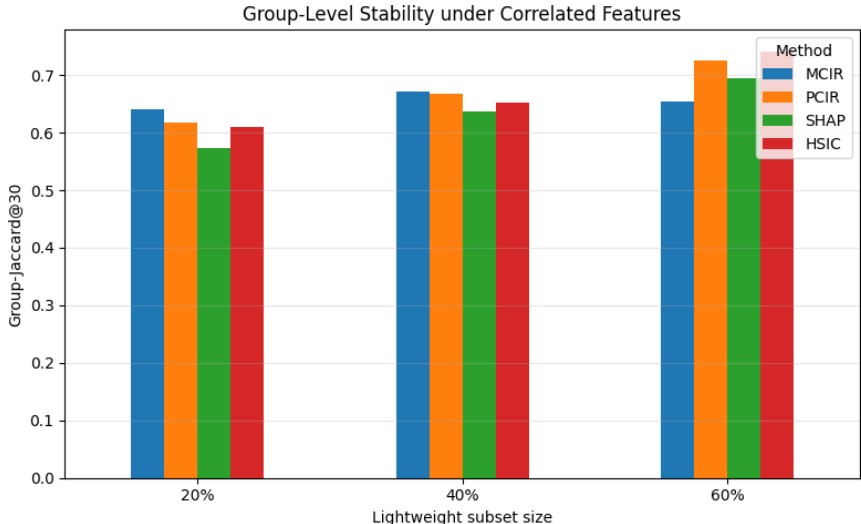

Figure 24: Group-level stability under correlated features. MCIR achieves higher agreement in low-data regimes.

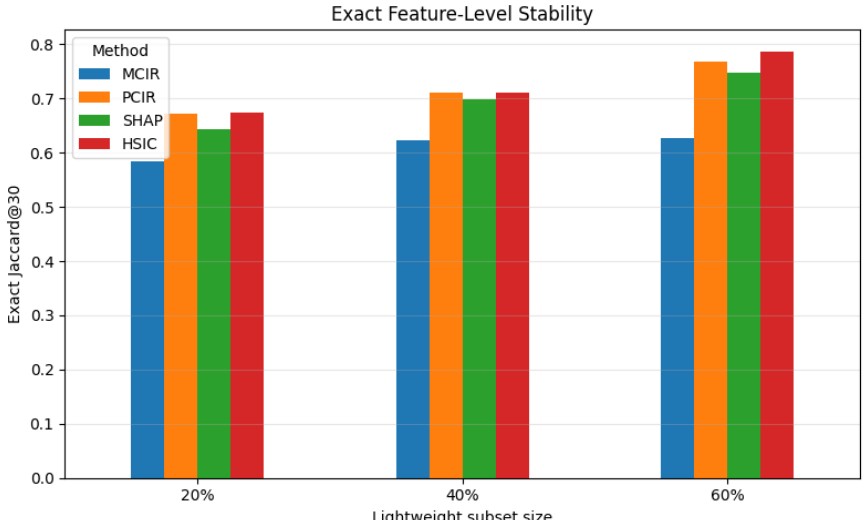

Figure 25: Exact feature-level stability under subsampling.

40%), and remains competitive at larger subset sizes. This indicates that MCIR preserves the underlying *informational structure* of the feature space, even when the exact feature indices selected may vary. This distinction reflects a fundamental difference in how attribution methods handle redundancy:

- **Exact stability** evaluates whether the same feature indices are selected,
- **Group-level stability** evaluates whether the same *predictive information* is recovered.

In settings with strong feature dependence, multiple features can encode equivalent signal. Consequently, strict identity-level agreement may underestimate stability, while group-level agreement provides a more faithful measure of explanatory consistency.

Across all analyses—subset stability, deletion curves, and rank-overlay comparisons—a consistent pattern emerges. First, MCIR preserves **informational consistency** rather than exact feature identity. While exact

Jaccard@30 may be lower, group-level agreement shows that MCIR reliably captures the same underlying predictive structure. Second, deletion-curve analysis demonstrates **distributed feature importance**, where predictive signal is spread across correlated variables rather than concentrated in a few selected features. This leads to more gradual degradation under feature removal. Third, rank-overlay analysis confirms that MCIR maintains stable **structural behaviour** under subsampling, even when individual feature rankings vary. Taken together, these results provide a unified explanation for the observed Jaccard behaviour: *exact feature overlap alone is not sufficient to evaluate stability under feature dependence.* Instead, stability should be assessed at the level of shared information and structural consistency. Under this perspective, MCIR demonstrates stronger robustness and more faithful attribution in correlated settings, aligning with its dependence-aware formulation.

