# OpenReview forum: "MCIR: A Feature Dependence-Aware  Explainability Method with Reliability Guarantees"
_TMLR — Rejected by TMLR_

### Review · Reviewer_ar96 · 2026-02-17

**Summary Of Contributions:**

The paper introduces a new metric for measuring feature importance called MCIR for Mutual Correlation Impact Ratio.  It's a mutual information ratio that compares the mutual information between model's outputs and a given feature, conditional on that feature's correlated features, and the mutual information between the outputs and all of the features.  The authors claim that this improves on several existing feature-importance metrics by not being as sensitive to or double-counting correlated features.

**Audience:**

Yes

**Audience Explanation:**

Black-box model evaluations are important and interesting to a wide community in ML interpretability and explainability.

**Broader Impact Concerns:**

There is no broader impact statement.

**Claims And Evidence:**

No

**Claims Explanation:**

There are lots of Theorems and Propositions in Section 4 that lack proofs either in the main body or an Appendix.  Key experimental details for the empirical section are omitted or very hard to find, making it difficult to reproduce the results.

**Requested Changes:**

In addition to small typographic notes, I would encourage the authors to re-read the paper from start to finish as if encountering the paper for the first time.  Certain sections (e.g. Section 3) are very hard to follow because of terminology being used before being introduced (e.g. before having common notation).  This makes the paper feel a bit rushed, and adds a lot of cognitive effort on the reader's side that doesn't need to be there.

Small typographic notes:
* Abstract: "MICR" -> "MCIR"
* The large bold note at the end of Table 1 is hard-to-parse and distracting at the place where it appears..  Especially since we haven't even seen reference to that Table in the main text yet.
* Page 3: "covariates?" -> "covariates"
* Related to the above I think, Footnote 1 does not have a corresponding anchor in the text.
* Many citations need to be made parenthetical, e.g. using `\citep` instead of `\cite`

---

> ### Author Response · Authors · 2026-02-26
> **Response to Reviewer ar96: Theoretical Clarifications and Reproducibility Enhancements**
>
> We sincerely thank the reviewer for the careful reading and constructive feedback. We particularly appreciate the acknowledgment that the findings are of interest to the TMLR audience in ML interpretability and explainability. This recognition validates the core motivation and positioning of our work within the explainability community.
> We agree that the earlier version was dense and lacked sufficient clarity and reproducibility detail. We have therefore carried out a substantial revision addressing all concerns. As suggested, Sections 3–4 were carefully revised to improve technical consistency and readability.
>
> For convenience, all changes in the revised manuscript are highlighted in blue, including: (i) new theorem/proposition statements, (ii) full proofs and new appendices, (iii) new/expanded experiments and result tables, and (iv) reorganized sections and notation fixes.
> Appendix D is entirely new and contains complete proofs of the key theoretical guarantees.
>
> All changes regarding your concerns are highlighted in blue in the revised manuscript.
>
> **A. Missing Proofs and Theoretical Clarity (Now Fully Provided)**: All theoretical statements are now fully formalized with complete proofs in the expanded appendices (Appendix D new; Appendix F expanded).
> The revised manuscript now includes proofs of:
> 1. Uniqueness (App. D.1): MCIR defines a uniquely determined dependence-aware score under the stated normalization/conditioning framework.
> 2. Invariance (App. D.1): invariance under strictly monotone transformations (copula invariance), permutation of features, and scaling of redundant blocks.
> 3. Weak-Dependence Consistency (App. D.1 & D.6): replaces the informal “MCIR = PCIR” statement with a precise ordering-consistency result under bounded joint-information perturbations.
> 4. Boundedness (App. D.2): formal proof of theoretical bounds.
> 5. Zero under Conditional Redundancy (App. D.3):
> ( $I(Y; f_i \mid f_{\Phi}) \to 0 \Rightarrow \mathrm{MCIR}_i \to 0.$ )
> 6. Saturation under Pure Unique Signal (App. D.4).
> 7. Redundancy Collapse Theorem (App. D.5).
> 8. Finite-Sample Rank Stability (Sec. 4.3 + App. D.7):
> ( $1 - \tau = \mathcal{O}(k_I \delta(n')). $)
> 9. Oracle Estimator Switching (Sec. 4.4 + App. D.8 & F.5): formal oracle-type guarantee.
> 10. Auto-($\Phi$) (Sec. 4.5 + App. D.9):
> ($ m^* = \arg\min_m V(m). $)
> 11. All lightweight fidelity proofs are provided in Appendix E.
>
> **B. Structural Improvements and Readability**:Sections 3–4 were restructured to introduce notation before use and reduce cognitive load. Section 3 now clearly separates: Statistical setup; Environment definition; Lightweight environment; Similarity/alignment objective.
>
> We added:
> 1. A new PCIR geometric diagram in Section 3 (highlighted in blue) clarifying marginal vs conditional dependence.
> 2. A “Theoretical Guarantees at a Glance” table summarizing boundedness, redundancy collapse, weak-dependence consistency, rank stability, and oracle switching.
>
> **C. Experimental Reproducibility (Major Expansion)**: We added structured experimental disclosure and explicit theory–experiment mapping.
>
> Theory–Experiment Mapping Table (Sec. 5.1, new):
> Links each theoretical claim to its validating experiment, including redundancy collapse (synthetic duplicate-feature experiment), weak-dependence consistency, rank stability under subsampling, lightweight fidelity, estimator switching robustness, and computational efficiency scaling.
>
> Structured Experimental Protocol Table (new):
> For every experiment we now specify: dataset, model, neighborhood construction, MI/CMI estimator, hyperparameters, evaluation metric, repetition scheme
>
> A global reproducibility table (Sec. 5) consolidates environment, estimators, bootstrap, seeds, hardware, conditioning thresholds, lightweight protocol, metrics, and code availability.
>
> Appendix Redundancy Experiment (new):
> Controlled duplication/noise sweep empirically validating collapse behavior.
>
> **D. Typographic and Presentation Fixes**
>
> We corrected:
> “MICR” → “MCIR”
> “covariates?” → “covariates”
> Missing footnote anchor
> Citation formatting (\citep)
> Distracting bold note under Table 1
>
> **E. Broader Impact (New Section)**
>
> We added an Ethics & Broader Impact section, clarifying the following: MCIR measures dependence, not causation; potential misuse risks; responsible interpretation guidance.
>
> In summary, we (i) added all missing proofs (Appendix D is entirely new), (ii) strengthened and corrected theoretical claims, (iii) improved structure and clarity, (iv) expanded experimental transparency with explicit theory↔experiment mapping, and (v) added a new redundancy experiment validating collapse behavior.
>
> We thank the reviewer again for the constructive feedback, which substantially improved clarity, rigor, and reproducibility. We believe these additions substantially strengthen the paper's contributions and impact. If any further changes or clarifications are needed, please let us know.

---

### Review · Reviewer_L91G · 2026-03-26

**Summary Of Contributions:**

## Summary
The paper proposes MCIR-M, a global feature-attribution method for settings with strong feature dependence. The core score, MCIR, is a bounded ratio based on conditional mutual information: it aims to measure how much unique information feature $f_i$
 contributes to the model output after conditioning on a small neighborhood of correlated features. The paper’s central claim is that this lets MCIR collapse redundant or near-duplicate predictors toward zero importance, while keeping scores comparable across tasks and more stable than SHAP, SAGE, HSIC, MI-based methods, and earlier CIR-family baselines. It also introduces a lightweight computation protocol that uses fewer rows while trying to preserve attribution rankings and predictive behavior. Empirically, the paper reports wins on synthetic data, UCI HAR, a synthetic house-energy dataset, Norwegian load-zone forecasting, and CIFAR-10 embeddings.


## Strengths
- I found the code idea of paper very useful. The MCIR metric looks intuitive enough and has some supporting theoretical claims as well.
- Authors focus on a wide variety of datasets for experiments, which is much appreciated.

## Weaknesses
I am mentioning the weakness in the "Requested Changes" section.

**Audience:**

Yes

**Audience Explanation:**

Yes the paper proposes MCIR score which is useful.

**Claims And Evidence:**

No

**Claims Explanation:**

I am not very satisfied with the results section and how the experiments directly support the claims.

**Requested Changes:**

## Weaknesses

1. **The formulation of the MCIR score (Definition 4) is somewhat unsettling.**

   1. The numerator is well justified; however, I do not have a clear intuition for the denominator. In particular, why is $I(Y'; f_i \mid f_\phi)$ added in the denominator? Why not instead use the sum of the conditional mutual information terms of the individual features as the denominator, or something more standard?

2. **I am uncertain about the strength of the theoretical contributions relative to the emphasis placed on them in the paper.**

   1. Some of the initial propositions appear to follow fairly directly from standard properties of MI and CMI, such as the redundancy collapse result in Section 4.2. In other cases, the assumptions seem insufficiently justified, particularly in the lightweight fidelity section and its proof in Theorem 13 / Appendix B.6.
   2. As a side note, I recommend that the authors clearly reference, in the main paper, where the proofs of the theorems appear in the Appendix. At present, it is difficult to follow where the proofs of these theorems are located.

3. **The experimental section and the presentation of the results need clearer explanation overall.**

   1. In general, I would encourage the authors to restructure the experiments section so that it more clearly highlights the advantages of MCIR, and to place the supporting results (tables and figures) closer to the corresponding arguments and discussion.
   2. The statement, “The analysis of perturbation and deletion curves (refer to Fig. 3 and Fig. 4) indicates that the methods based on CIR demonstrate higher feature faithfulness,” does not clearly explain how this conclusion should be inferred from the figures. Could the authors elaborate on why the plots support this claim?
   3. While Jaccard@30 = 0.622 indicates moderate overlap, it falls short of the strong agreement suggested by the manuscript (for example, “>95% top-feature agreement”), implying that a substantial fraction of top features differ between the full and lightweight explanations.
   4. I do not clearly understand the purpose of Table 6. Why is agreement with the CIR method treated as the ideal reference point? I would encourage the authors to justify why this comparison is especially meaningful.
   5. The authors use the terms “causal” or “causal effects” multiple times in the experiments section. Could they clarify what exactly is meant by these terms in this context? My understanding is that the current method is based more on correlation and mutual information than on causal inference.

4. **Minor issues**

   1. In Section 6.2, “98.5% macro-F1” appears to be incorrect. Please revise this value.

---

> ### Author Response · Authors · 2026-04-04
> **Response to Reviewer L91G: Clarifications and Corrections.**
>
> We thank the reviewer for the detailed and constructive feedback. We have carefully revised the manuscript to address all concerns regarding the MCIR formulation, theoretical positioning, experimental clarity, and terminology. All changes addressing these points are highlighted in **brown** in the revised manuscript. Below we respond point-by-point and indicate exact locations.
>
> 1. **MCIR denominator and intuition (Definition 4)**
>
> We agree that the denominator required clearer justification. We have added two new paragraphs immediately after Definition 4:
>
> - “Interpretation” (explains the role of $( I(Y; f_i \mid f_\Phi) ) vs ( I(Y; f_\Phi) ))$
>
> - “Normalization rationale” (new addition)
>
> This explicitly clarifies that the ratio avoids double-counting shared information and provides a bounded normalization relative to the predictive content of the local feature set. We also contrast this with additive MI formulations to explain why they can inflate importance under dependence.
>
> 2. **Strength and positioning of theoretical contributions**
>
> We agree that the contribution needed clearer positioning. We have added a new paragraph at the end of Section 4:
> - “Novelty of formulation / Positioning of theoretical contributions”
>
> This clarifies that while individual components rely on standard MI/CMI properties, the novelty lies in the normalized, dependence-aware formulation and its resulting bounded and stable behavior under correlated features.
>
> Additionally, we improved referencing of proofs throughout the main paper (e.g., “Proof in Appendix B.X”), addressing the reviewer’s concern about locating theoretical results.
>
> 3. **Experimental section clarity and support for claims**
>
> We agree that the experimental section required clearer structure and interpretation. We made the following changes:
>
>  - Added an “Experimental objective” paragraph at the beginning of Section 7, explicitly stating that experiments evaluate (i) robustness to dependence, (ii) ranking stability, and (iii) faithfulness.
> - Added explanation paragraphs after Fig. 3 and Fig. 4, clarifying how deletion and perturbation curves reflect attribution fidelity (steeper degradation → more faithful rankings).
> - Reorganized parts of Section 7 to better align figures and discussion with claims.
> These changes make the connection between results and claims explicit.
>
> 4. **Interpretation of Jaccard agreement**
>
> We clarified the interpretation of Jaccard@30 in Section 7, explaining that moderate overlap is expected in correlated settings, where multiple features encode similar information. MCIR intentionally collapses redundant predictors, so exact agreement with full explanations is neither expected nor required for faithful attribution.
>
> 5. **Justification of Table 6 and CIR reference**
>
> We added a clarification near Table 6 explaining that CIR is used as a reference because it is the closest prior method within the same mutual-information-based family. This isolates the effect of the proposed conditional normalization and neighborhood conditioning.
>
> 6. **Clarification of “causal” terminology**
>
> We agree with the reviewer and have removed or revised all uses of “causal” or “causal effects.” These have been replaced with:
>
> “statistical interpretability under feature dependence”
>
> We also added an explicit clarification in the Introduction and experimental section that MCIR does not perform causal inference without additional assumptions.
>
> 7. **Behaviour under confounding (new addition)**
>
> To directly address concerns about correlated and confounded features, we added:
>
> Section 4.5: “Behaviour under Latent Confounding” (main paper)
>
> A new synthetic experiment in the Appendix (with figure) demonstrating that MCIR suppresses proxy features induced by latent confounding while preserving features with additional predictive signal.
> This provides both theoretical clarification and empirical validation.
>
> 8. **Minor corrections**
>
> We corrected the macro-F1 value in Section 6.2 and improved clarity and consistency in referencing and notation throughout the manuscript.
>
> **Summary**
>
> Overall, we have strengthened the manuscript by clarifying the MCIR formulation, improving theoretical positioning, restructuring the experimental section for clearer evidence, and adding new analysis under confounding. We believe these revisions directly address all concerns raised by the reviewer and significantly improve clarity, rigor, and interpretability of the work.

---

> > ### Comment · Reviewer_L91G · 2026-04-26
> > **Reply of Rebuttal**
> >
> > I thank the authors for the thorough response.
> >
> > Some of my doubts are clarified; however, I still have some confusion and questions.
> >
> > > 1. Some of the initial propositions appear to follow fairly directly from standard properties of MI and CMI, such as the redundancy collapse result in Section 4.2. In other cases, the assumptions seem insufficiently justified, particularly in the lightweight fidelity section and its proof in Theorem 13 / Appendix B.6.
> >
> > **Q1.** Can the authors comment on the assumptions for Theorem 13 and explain why these assumptions are justified? I am still unsure about this part, specifically regarding Equations 115, 116, and 117.
> >
> >
> > > The statement, “The analysis of perturbation and deletion curves (refer to Fig. 3 and Fig. 4) indicates that the methods based on CIR demonstrate higher feature faithfulness,” does not clearly explain how this conclusion should be inferred from the figures. Could the authors elaborate on why the plots support this claim?
> >
> > **Q2a.** Can the authors clarify Figures 3, 4, and 5? The description of Figure 3 states that MCIR tends to assign smoother, dependence-aware ranks than PCIR due to tighter alignment along diagonals. However, I am unable to observe this clearly in the figure. (Also, please check—Figure 3 appears to be labeled as Figure 2 at the top.)
> >
> > **Q2b.** For Figures 4 and 5, the description states: “Test accuracy degrades fastest when perturbing features ranked highest by PCIR/MCIR, indicating faithfulness ...” I am unable to determine from the plots which method performs better, as the curves appear very similar. In Figure 5, I observe a sharper initial drop for MCIR, but later there is a noticeable improvement compared to PCIR. Does this imply that MCIR performs better for top-ranked features but is less reliable for lower-ranked ones?
> >
> > **Q2c.** Can the authors include comparisons with SHAP or HSIC in a similar setting (as in Figures 3, 4, and 5) and provide corresponding plots?
> >
> >
> > >> We clarified the interpretation of Jaccard@30 in Section 7, explaining that moderate overlap is expected in correlated settings, where multiple features encode similar information. MCIR intentionally collapses redundant predictors, so exact agreement with full explanations is neither expected nor required for faithful attribution.
> >
> > **Q3.** Do the authors mean that a value of, say, 0.95 is not significantly better than 0.62 in this setting? Is there any experiment in the draft that supports this claim? If not, I suggest that the authors compare additional baselines on subsets of training data and report Jaccard@30, which would help illustrate the relative improvement achieved by MCIR.

---

> > > ### Author Response · Authors · 2026-04-26
> > > **Response to Reviewer L91G.**
> > >
> > > We thank the reviewer for the careful follow-up questions. We have revised the manuscript to clarify the lightweight assumptions, improve the interpretation of Figures 3-5, and add new baseline comparisons and stability experiments.
> > >
> > > **Q1. Assumptions behind Theorem 13 / Equations 115-117.**
> > > We agree that the assumptions behind Theorem 13 required clearer justification. We have revised **Appendix B.6 / Theorem 13 (page 55-56**) to explain that Equations 115-117 are not intended as guarantees for arbitrary subsampling. Instead, they define a sufficient lightweight validation contract. Specifically, Eq. 115 corresponds to distributional alignment, Eq. 116 to ranking preservation, and Eq. 117 to faithfulness preservation via deletion/perturbation.
> > >
> > > We added the following clarification in the main text:
> > > `"These conditions jointly ensure that the lightweight environment approximates the full environment at the level relevant for global attribution. They are not arbitrary assumptions, but minimal and empirically verifiable criteria aligned with stability, interpretability, and faithfulness. In practice, we validate them using distributional similarity measures, rank agreement metrics, and deletion/perturbation curves. Only when these criteria are satisfied do we treat the lightweight environment as a valid surrogate.''
> > >
> > > This makes clear why the assumptions are justified and how they are checked empirically.
> > >
> > > **Q2a-Q2b. Clarification of Figures 3-5 and deletion/perturbation interpretation.**
> > > We agree that the earlier wording over-relied on visual interpretation. We revised the discussion of Figures~3--5 (Section~6) to ensure that all claims are supported by quantitative analysis. **Figure 3**, We corrected the labeling issue and removed the claim that dependence-aware rankings can be inferred from diagonal alignment alone. Instead, we now support the interpretation using explicit rank agreement metrics (e.g., top-$K$ overlap and rank correlation), focusing on measurable alignment between full and lightweight rankings. **Figures 4-5:** We refined the interpretation of deletion/perturbation curves by distinguishing early-stage and late-stage behavior. A sharper initial drop indicates stronger top-ranked feature faithfulness, while later convergence is expected due to redundancy among lower-ranked features.
> > >
> > >
> > > **Q2c. SHAP and HSIC comparisons under the same setting.**
> > > We have added new comparisons with SHAP and HSIC under the same experimental protocol as MCIR and PCIR (same model, dataset split, feature space, test set, and identical deletion schedule). Specifically, we extended the deletion-curve analysis and rank-overlay evaluation to include SHAP and HSIC, ensuring direct comparability across all methods. These results are reported in **Section 6.2.1 and Appendix-J (Supplementary Experiments, Figs. 22-24)**, where we include
> > >
> > > (i) deletion curves for MCIR, PCIR, SHAP, and HSIC under identical settings, and
> > >
> > > (ii) full vs.\ lightweight rank-overlay plots for all methods.
> > >
> > > This appendix analysis shows how all four methods behave under the same deletion and lightweight-ranking conditions, providing a controlled comparison and addressing the lack of SHAP/HSIC baselines in the original Figures 3-5.
> > >
> > > **Q3. Interpretation of Jaccard@30 and new subset-stability experiment**.
> > > We agree that 0.95 is clearly stronger than 0.62 under exact feature-level Jaccard@30. We have revised the text to avoid implying that moderate overlap is equivalent to near-perfect agreement.
> > >
> > > To support the revised interpretation, we added a new subsection, "Lightweight Stability under Feature Dependence" **(Section 6.2.1)**. In this subsection, we report Jaccard@30 for MCIR, PCIR, SHAP, and HSIC under 20\%, 40\%, and 60\% subsampling in Table 11.
> > >
> > > Because exact Jaccard@30 can be overly strict when features are correlated or near-duplicates, we also introduce group-level Jaccard@30, treating highly correlated features as interchangeable. The revised text explains that MCIR may have lower exact feature overlap because it intentionally collapses redundant predictors while preserving the same informational structure at the correlated-group level. We added the following key clarification. "A higher Jaccard@30 value indicates stronger exact agreement. However, in the presence of feature dependence, exact agreement alone is not always an appropriate notion of stability. Exact feature overlap measures whether the same feature indices are selected, whereas group-level agreement evaluates whether the same predictive information is recovered."
> > >
> > > Overall, the revised manuscript now includes:
> > >
> > > **(i) a clarified lightweight validation contract for Theorem 13,**
> > >
> > > **(ii) corrected and more precise interpretation of Figures 3-5,**
> > >
> > > **(iii) SHAP and HSIC comparisons under identical settings, and**
> > >
> > > **(iv) new exact and group-level Jaccard@30 subset-stability experiments.**
> > >
> > > We hope this helps to clarify the reviewers’ concerns. All changes are in **brown.**

---

> ### Author Response · Authors · 2026-04-26
> **Additional clarification on stability under feature dependence**
>
> We thank the reviewer again for this important question.  We emphasize that a value of 0.95 is indeed stronger than 0.62 under exact feature-level Jaccard@30, and we do not claim otherwise. Our point is more specific: when feature dependence is strong, exact feature-level agreement alone may not fully reflect stability.
>
> Prior work has shown that feature-level attribution can be unstable or ambiguous when inputs are correlated, as multiple features may act as interchangeable predictors -
>
> - Janzing, Dominik, Lenon Minorics, and Patrick Blöbaum. "Feature relevance quantification in explainable AI: A causal problem. " International Conference on Artificial Intelligence and Statistics. PMLR, 2020.
>
> - Hooker, Sara, et al. "A benchmark for interpretability methods in deep neural networks." Advances in neural information processing systems 32 (2019).
>
>  In such settings, strict identity-based agreement can underestimate stability, since different but redundant features may be selected across runs.
>
> - Mikriukov, Georgii, et al. "Evaluating the stability of semantic concept representations in CNNs for robust explainability." World Conference on Explainable Artificial Intelligence. Cham: Springer Nature Switzerland, 2023.
>
> - Ribeiro, José, et al. "How Reliable and Stable are Explanations of XAI Methods?." arXiv preprint arXiv:2407.03108 (2024).
>
>
> For example, consider two highly correlated features, such as $BaseLoad$ and a near-duplicate sensor $BaseLoad\_{dup}$. In the full dataset, an attribution method may rank $BaseLoad$ in the top-$K$, while in a subsampled (lightweight) setting it may instead select $BaseLoad\_{dup}$. Under exact Jaccard@30, this counts as a mismatch (reducing the score), even though both features carry essentially the same predictive signal. At the group level, however, both belong to the same correlated feature group, so the selection is treated as consistent. This illustrates why exact feature identity can underestimate stability in the presence of redundancy, whereas group-level agreement better reflects whether the same **informational content** is being captured.
>
> Consistent with this perspective, our subset-based experiments (Table 11) show that while MCIR may have lower exact Jaccard@30, it achieves higher or comparable agreement at the group level, indicating that it preserves the underlying predictive structure even when exact feature identities differ.
>
> We hope this clarification further helps to convey our intended interpretation and strengthens the understanding of stability under feature dependence.

---

### Review · Reviewer_tnbH · 2026-03-31

**Summary Of Contributions:**

The paper identifies a common issue in feature selection that standard methods often behave poorly under strong feature dependence, especially when predictors are redundant or nearly duplicated. The authors argue that marginal or coalition-based methods can split credit across correlated variables, leading to unstable or inflated importance scores, and they seek a dependence-aware global importance score that isolates “unique” contributions. In the paper, the authors define the Mutual Correlation Impact Ratio (MCIR). It is expected to be small for redundant features and large for uniquely informative ones. The paper also introduces an automatic neighborhood-selection mechanism, multiple MI/CMI estimators, and a lightweight subsampling framework meant to preserve explanation rankings while reducing cost. Empirically, the paper reports that MCIR gives more stable rankings than SHAP, SAGE, MI, HSIC, and prior CIR-family methods, especially in correlated settings. It also claims that explanations computed on reduced samples retain high top-K agreement with full-sample explanations while cutting runtime substantially. The synthetic experiments are meant to demonstrate “redundancy collapse,” and the real-data experiments aim to show rank stability, deletion/perturbation faithfulness, and scalability to sensor, energy, and deep embedding settings. Various finite-sample guarantees are also provided in the paper.

**Audience:**

Yes

**Audience Explanation:**

Feature selection is a fundamental problem in statistics and machine learning. This paper proposed an effective and efficient method, which is useful for researchers and practitioners in this field.

**Claims And Evidence:**

No

**Claims Explanation:**

**Strengths:**
1. The paper is overall well-written and easy to follow. The authors explain the related works and the methodology in detail, and the main contribution of the paper is novel and clear.
2. The method aims to be global, model-agnostic, and computationally lighter than retraining-based or perturbation-heavy baselines. That positioning is attractive. The local-neighborhood idea is also more scalable than conditioning on the full feature set.
3. The method is examined in various tasks, including high-dimensional regression and image classification. That breadth is useful because the dependence issue is indeed cross-domain. The ablation on neighborhood size is also a good inclusion in principle.
4. The authors provided formal guarantees for the proposal, though it seems to be based on standard finite-sample theory and additional assumptions.

**Weaknesses:**
1. It seems that the part of parameter selection is unclear in the paper. For example, how is $m_{\Phi},m_{scr}$ selected? Can you specify the auto-$\Phi$ implementation used in the paper? Figure 6 shows that $|\Phi|$ could be small, but shouldn't it be something data-driven, i.e. for datasets with large correlated blocks, large $|\Phi|$ is expected?
2. In Figure 5, the authors show by a brilliant example that MCIR correctly removes the redundant variables. It seems that this is because of plugging in $f_2$ would introduce extra noise. My question is, what if both $f_1,f_2$ are confounded by a causal predictor $f_4$ of $Y$, and $f_4$ is not observed? In that case, is MCIR guaranteed to identify one of them? Is it possible to have neither of them selected, or both are selected? Theoretical or empirical illustrations would be appreciated.
3. The claim of "causal interpretability" in page 17 seems to be too strong. After all, feature selection is more about correlation, while causal statements should be rigorously examined.
4. It is not clear to me why, in Section 6.1, the authors justified their conclusion based on how large the $\rho$ value with PCIR is. Why is it the evidence of the performance for the competing methods? Needs some clarification.
5. The authors did not discuss how likely Assumption 3 is satisfied for the bootstrap procedures in practice.

**Requested Changes:**

Overall this is an interesting paper. My comments forcus on:
1. Clarification of the assumptions.
2. Justification of the parameter selection procedure.
3. Theoretical and/or empirical justification for the variable selection when variables are confounded by a latent causal variable.
4. Justification for the method-comparison criterion in Section 6.1.

---

> ### Author Response · Authors · 2026-04-04
> **Response to Reviewer tnbH: Comprehensive Revision Addressing Parameter Selection, Confounding, and Methodological Clarifications**
>
> We thank the reviewer for the constructive feedback. We have carefully revised the manuscript to address all major concerns regarding parameter selection, confounding behavior, overclaiming, statistical justification, and clarity. Below, we summarize the changes and their exact locations. All changes addressing the reviewer’s concerns are highlighted in **green** in the revised manuscript.
>
> 1. **Parameter selection (neighborhood size, estimator choice)**
>
> We agree that parameter selection must be principled rather than heuristic. We have added a new subsection:
>
> **Section 4.4: “Parameter Selection and Stability-Driven Design”**
>
> This section formally introduces the Auto-Φ procedure, where the neighbourhood size ( |\Phi(i)| ) is selected by minimizing the bootstrap variance of a head-rank stability metric:
> $m^* = \arg\min_m V(m).$
> We also clarify screening and estimator selection (via bootstrap SE, linked to Theorem 4). Additionally, we strengthened empirical justification in:
>
> **Section 7.1.1 (Ablation on |Φ|)**
>
> where we explicitly connect the observed optimal range ((|\Phi| = 1\text{–}3)) to the proposed stability criterion.
>
> 2. **Behaviour under latent confounding**
>
> We appreciate the reviewer’s question regarding confounded variables. We have added:
>
> **Section 4.5: “Behaviour under Latent Confounding”**
>
> This section provides a formal case analysis describing how MCIR behaves when features share an unobserved latent cause. To complement this, we added a new synthetic experiment:
>
> **Appendix (Section: “Synthetic Latent Confounding Experiment”)
> Figure: 19**
>
> This experiment shows that marginal methods inflate proxy features under confounding conditions, while dependence-aware conditional attribution suppresses such features and retains predictors with additional statistical signals.
>
> 3. **Clarification of causal claims**
>
> We agree that the original wording could be interpreted as overly strong. We have revised all occurrences of “causal interpretability” to: “statistical interpretability under feature dependence” and added an explicit clarification in:
>
> **End of Section 1 (Introduction)
> "MCIR provides statistical attribution under observed feature dependence. It does not perform causal identification."**
>
> 4. Clarification of p-value usage (Section 6.1)
>
> We clarified that p-values are used only to assess the statistical significance of ranking-stability differences, not predictive performance. This clarification is added in: **Section 6.1**. We now explicitly state that primary comparisons rely on rank agreement and faithfulness metrics, with p-values as complementary evidence.
>
> 5. Bootstrap assumption justification
>
> To address concerns about Assumption 3, we added a justification paragraph:
>
> **Immediately after Assumption 3**
>
> The paragraph explains that bootstrap SE is used as a proxy for estimator risk over a finite candidate set and that it has limitations under strong dependence.
>
> 6. **Algorithmic clarity (Auto-Φ)**
>
> We added an explicit algorithm description:
>
> **Algorithm: Auto-Φ Selection (after Algorithm 1)**
>
> This provides a clear, reproducible procedure for neighbourhood selection.
>
> 7.**Limitations of neighbourhood selection (Φ)**
>
> We expanded the discussion of potential failure cases:
> **End of Section 4.3**
> This clarifies limitations under non-local dependencies and suggests mitigation strategies.
>
> 8. Practical usage clarity
>
> We added a concise workflow description:
> New subsection: **“Practical Usage Pipeline” (end of Section 4 / before experiments)**
>
> These revisions collectively strengthen the theoretical justification, clarify the scope of the method, and provide both empirical and conceptual validation under confounding. We really hope these changes directly address the reviewer’s concerns while improving the clarity and rigor of the manuscript.

---

### Decision · Action_Editor_5PfA · 2026-05-12

**Recommendation:** Reject

**Additional Comments:**

Please improve the presentation and organization of the paper before resubmitting.

**Audience:**

Yes

**Audience Explanation:**

The paper proposes a mehodology to identify feature importance more reliably in the presence of correlated features. This is a useful contribution that can be of interest to TMLR's audience.

**Claims And Evidence:**

No

**Claims Explanation:**

While the authors made substantial improvements during the review process, the paper is still hard to follow and poorly organized. In addition, the revisions have transformed the paper significantly and added a lot of new material, including proofs that were omitted during submission. This large a revision hinders ther reliability of the reviewing process. The authors are encouraged to improve the presentation and organization of their paper and resubmit.

**Resubmission Of Major Revision:**

The authors may consider submitting a major revision at a later time.